# Accelerated Convergence of Stochastic Heavy Ball Method under Anisotropic Gradient Noise

**Rui Pan**[1]*, **Yuxing Liu**[2]*, **Xiaoyu Wang**[1], **Tong Zhang**[3]

[1]The Hong Kong University of Science and Technology
[2]Fudan University
[3]University of Illinois Urbana-Champaign
`rpan@connect.ust.hk, yuxingliu20@fudan.edu.cn, maxywang@ust.hk,`
`tozhang@illinois.edu`

## Abstract

Heavy-ball momentum with decaying learning rates is widely used with SGD for optimizing deep learning models. In contrast to its empirical popularity, the understanding of its theoretical property is still quite limited, especially under the standard anisotropic gradient noise condition for quadratic regression problems. Although it is widely conjectured that heavy-ball momentum method can provide accelerated convergence and should work well in large batch settings, there is no rigorous theoretical analysis. In this paper, we fill this theoretical gap by establishing a non-asymptotic convergence bound for stochastic heavy-ball methods with step decay scheduler on quadratic objectives, under the anisotropic gradient noise condition. As a direct implication, we show that heavy-ball momentum can provide $\tilde{\mathcal{O}}(\sqrt{\kappa})$ accelerated convergence of the bias term of SGD while still achieving near-optimal convergence rate with respect to the stochastic variance term. The combined effect implies an overall convergence rate within log factors from the statistical minimax rate. This means SGD with heavy-ball momentum is useful in the large-batch settings such as distributed machine learning or federated learning, where a smaller number of iterations can significantly reduce the number of communication rounds, leading to acceleration in practice.

## 1 Introduction

Optimization techniques that can efficiently train large foundation models (Devlin et al., 2019; Brown et al., 2020; Touvron et al., 2023a;b; Ouyang et al., 2022) are rapidly gaining importance. Mathematically, most of those optimization problems can be formulated as minimizing a finite sum

$$\min_{\mathbf{w}} f(\mathbf{w}) \triangleq \frac{1}{N} \sum_{i=1}^{N} f_i(\mathbf{w}),$$

where numerical methods are normally applied to find the minimum of the above form. Among all those methods, stochastic gradient descent (SGD) (Robbins and Monro, 1951) and its variants can be regarded as one of the most widely used algorithms.

For instance, heavy-ball (HB) methods (Polyak, 1964), commonly referred as heavy-ball momentum, are one of those popular variants. Empirically, it was extremely helpful for accelerating the training of convolutional neural networks (Szegedy et al., 2015; Simonyan and Zisserman, 2015; He et al., 2015; Huang et al., 2017; Sandler et al., 2018). Theoretically, it has been shown to provide optimal acceleration for gradient descent (GD) on quadratic objectives (Nemirovski, 1995).

Nonetheless, when it comes to SGD in theory, things become much different. Despite its huge success in practice, most theoretical results of stochastic heavy ball (SHB) were negative, showing that the

---

*Equal contribution.

convergence rates of heavy-ball methods are no better than vanilla SGD (Devolder et al., 2013; Yuan et al., 2016; Loizou and Richtárik, 2017; Kidambi et al., 2018; Jain et al., 2018a; Li et al., 2022). The existence of these gaps between GD and SGD, between practice and theory, is rather intriguing, which may make one wonder: *Can stochastic heavy ball provide $\tilde{\Theta}(\sqrt{\kappa})$ accelerated convergence when the noise is small, such as under large-batch settings?*

To answer this question, the first step is to find the missing pieces in those negative results. One key observation is that all those negative results assumed constant learning rates, while in practice, decaying learning rates are usually used instead. Those decaying learning rates, often referred as learning rate schedules, were demonstrated to be critical for improving the performance of a trained model in real-world tasks (Loshchilov and Hutter, 2017; Howard and Ruder, 2018). Furthermore, if one only considers the vanilla SGD algorithm, the theoretical property of most schedules have already been well inspected (Shamir and Zhang, 2013; Jain et al., 2019; Ge et al., 2019; Harvey et al., 2019; Pan et al., 2021; Wu et al., 2022a). Briefly speaking, one can view learning rate schedules as a variance reduction technique, which helps alleviate the instability and deviation caused by stochastic gradient noise.

Since it has been pointed out by (Polyak, 1987) that variance reduction is the key to improving stochastic heavy ball's convergence rate, it is then natural to ask: *Are there proper learning rate schedules that can help us achieve accelerated convergence for SHB under large-batch settings?*

Our paper gives a positive answer to this question. As a first step, we restrict ourselves to quadratic objectives. Although these problem instances are considered one of the simplest settings in optimization, they provide important insights for understanding a model's behavior when the parameter is close to a local optimum. Furthermore, past literature on Neural Tangent Kernel (NTK) (Arora et al., 2019; Jacot et al., 2018) suggests that the gradient dynamics of sufficiently wide neural networks resemble NTKs and can have their objectives approximated by quadratic objectives given specific loss functions.

Motivated by the empirical anisotropic behavior of SGD noises near minima of modern neural networks (Sagun et al., 2018; Chaudhari and Soatto, 2018; Zhu et al., 2019) and theoretical formalization of this noise property in least square regression (Jain et al., 2018b;a; Pan et al., 2021), we conduct our analysis based on the assumption of anisotropic gradient noise, which is formally defined later as Assumption 3 in Section 3. Notice that the very same condition has already been adopted or suggested by many past literatures (Dieuleveut et al., 2017; Jastrzębski et al., 2017; Zhang et al., 2018; Zhu et al., 2019; Pan et al., 2021).

## 1.1 Our Contributions

1. We introduce novel theoretical techniques for analyzing stochastic heavy ball with multistage schedules, providing several key properties for the involved $2 \times 2$ update matrix $\mathbf{T}_i$. Specifically, we show that $\|\mathbf{T}_{T-1}\mathbf{T}_{T-2}...\mathbf{T}_0\|$ can be upper bounded by $\|\mathbf{T}_{T-1}^T\|$ under certain conditions. This allows SHB with changing learning rates to exhibit similar theoretical properties as vanilla SGD: for each eigenvalue, SHB first exponentially decreases the loss with large learning rates, then retains the reduced loss with small learning rates.

2. As a direct result of this technical innovation, we present a non-asymptotic last iterate convergence rate for stochastic heavy ball with step decay learning rate schedule on quadratic objectives, under the standard anisotropic gradient noise assumption. To the best of our knowledge, this is the first non-asymptotic result for SHB on quadratics that clearly expresses the relationship among iteration number $T$, condition number $\kappa$ and convergence rate with step-decay schedules.

3. Our results show that stochastic heavy ball can achieve near-optimal accelerated convergence under large-batch settings, while still retaining near-optimal convergence rate $\tilde{O}(d\sigma^2/T)$ in variance (up to log factors away from the statistical minimax rate).

## 2 Related Work

**Large batch training:** Large-batch training is a realistic setting of its own practical interest. In several recent efforts of accelerating large model training, it has been observed that large batch sizes

are beneficial for accelerating the training process (You et al., 2017; 2018; 2019; Izsak et al., 2021; Pan et al., 2022; Wettig et al., 2022). On top of that, in distributed machine learning (Verbraeken et al., 2020) and federated learning (Kairouz et al., 2021), one can normally support an outrageous size of large batches by adding machines/devices to the cluster/network, but unable to afford a large number of iterations due to the heavy cost of communication (Zheng et al., 2019; Qian et al., 2021). This makes acceleration techniques even more tempting under those settings.

**SGD + learning rate schedules:** In contrast, the research in SGD with learning rate schedules focused on more general settings without assuming constraints on the batch size. In (Ge et al., 2019), the convergence rate of step decay was proved to be nearly optimal on strongly convex linear square regression problems. (Pan et al., 2021) further pushed these limits to optimal for some special problem instances and offered a tighter upper bound, along with a lower bound for step decay. Concurrently, (Wu et al., 2022a) extended the analysis of (Ge et al., 2019) to a dimension-free version under overparamterized settings, with tighter lower and upper bounds provided for step decay schedules. In (Loizou et al., 2021), the convergence rate of Polyak step size on strongly convex objectives was investigated. Nevertheless, all the bounds in above works require SGD to have at least $\tilde{\Omega}(\kappa \log c)$ iterations to reduce the excess risk by any factor of $c$. There are also works with looser bounds but focus on more general objectives. Since we restrict ourselves to quadratics, we just list some of them here for reference: (Ghadimi and Lan, 2013; Hazan and Kale, 2014; Xu et al., 2016; Yuan et al., 2019; Vaswani et al., 2019; Kulunchakov and Mairal, 2019; Davis et al., 2023; Wolf, 2021).

**SGD + HB + constant learning rates:** Opposed to the positive results of near optimality for SGD, most results of stochastic HB with constant learning rates were negative, showing that its convergence rate cannot be improved unless extra techniques like iterate averaging are applied. In (Loizou and Richtárik, 2017; 2020), a linear convergence rate of SGD momentum on quadratic objectives for L2 convergence $\mathbb{E}[\|\mathbf{w}_T - \mathbf{w}_*\|^2]$ and loss $\mathbb{E}[f(\mathbf{w}_T) - f(\mathbf{w}_*)]$ was established, which requires at least $\tilde{\Omega}(\kappa \log c)$ iterations. A better bound for L1 convergence $\|\mathbb{E}[\mathbf{w}_T - \mathbf{w}_*]\|^2$ and $\mathbf{B}$ norm $\|\mathbb{E}[\mathbf{w}_T - \mathbf{w}_*]\|_{\mathbf{B}}^2$ was also proposed, but whether they are relevant to loss convergence is unclear. Here $\mathbf{B}$ is a positive definite matrix related to the problem instance and samples. In (Kidambi et al., 2018), momentum was proved to be no better than vanilla SGD on worst-case linear regression problems. In (Jain et al., 2018a), both SGD and momentum are shown to require at least $\Omega(\kappa)$ single-sample stochastic first-order oracle calls to reduce excess risk by any factor of $c$, thus extra assumptions must be made to the noise. A modified momentum method using iterate averaging was then proposed on least square regression problems and achieves $\tilde{\mathcal{O}}(\sqrt{\kappa\tilde{\kappa}})$ iteration complexity with an extra noise assumption. Here $\tilde{\kappa} \leq \kappa$ is the statistical condition number. In (Gitman et al., 2019), a last iterate rate of SGD momentum on quadratic objectives was presented, but the convergence rate is asymptotic. Non-asymptotic linear distributional convergence was shown in (Can et al., 2019), where SHB with constant learning rates achieves accelerated linear rates $\Omega(\exp(-T/\sqrt{\kappa}))$ in terms of Wasserstein Distances between distributions. However, this does not imply linear convergence in excess risks, where the variance is still a non-convergent constant term. In (Mai and Johansson, 2020), a class of weakly convex objectives were studied and a convergence rate of $\mathcal{O}(\kappa/\sqrt{T})$ was established for gradient L2 norm. In (Wang et al., 2021), HB on GD is analyzed and shown to yield non-trivial speedup on quadratic objectives and two overparameterized models. However, the analysis was done in GD instead of SGD. In (Bollapragada et al., 2022), SHB was shown to have a linear convergence rate $1 - 1/\sqrt{\kappa}$ with standard constant stepsize and large enough batch size on finite-sum quadratic problems. Their analysis, however, was based on an extra assumption on the sample method. (Tang et al., 2023) proved SHB converges to a neighborhood of the global minimum faster than SGD on quadratic target functions using constant stepsize. In (Yuan et al., 2021), a modified decentralized SGD momentum algorithm was proposed for large-batch deep training. Although it achieves $\tilde{\mathcal{O}}(1/T)$ convergence rate on a $L$-smooth and $\mu$-strongly convex objectives, it still requires at least $\tilde{\Omega}(\kappa)$ number of iterations to converge, which is no better than SGD. Wang et al. (2023) also provided cases where SHB fails to surpass SGD in small and medium batch size settings, suggesting that momentum cannot help reduce variance. There are also other variants of momentum such as Nesterov momentum (Nesterov, 2003; Liu and Belkin, 2019; Aybat et al., 2019), or modified heavy ball, but since we only consider the common version of heavy ball momentum here, we omit them in our context.

**SGD + HB + learning rate schedules:**   As for SHB with learning rate schedules, only a limited amount of research has been conducted so far. In (Liu et al., 2020), the convergence property of SHB with multistage learning rate schedule on $L$-smooth objectives was investigated. However, the inverse relationship between the stage length and learning rate size was implicitly assumed, thus its convergence rate is actually $\mathcal{O}(1/\log_\alpha T)$ for some constant $\alpha > 1$. In (Jin et al., 2022), a convergence rate was derived for general smooth objectives. But the relationship between the convergence rate and $T$ is still unclear, and the results were comparing SGD and SHB by their upper bounds. In (Li et al., 2022), a worst-case lower bound of $\Omega(\ln T/\sqrt{T})$ was found for SHB with certain choices of step sizes and momentum factors on Lipschitz and convex functions. A FTRL-based SGD momentum method was then proposed to improve SHB and achieve $\mathcal{O}(1/\sqrt{T})$ convergence rate for unconstrained convex objectives. Furthermore, in (Wang and Johansson, 2021), a $\mathcal{O}(1/\sqrt{T})$ bound was derived on general smooth non-convex objectives, whose analysis supports a more general class of non-monotonic and cyclic learning rate schedules. All these results only proved that SHB is no worse than SGD, or were comparing two methods by their upper bounds instead of lower bound against upper bound. Only until recently has SHB been shown to be superior over SGD in some settings. In (Zeng et al., 2023), a modified adaptive heavy-ball momentum method was applied to solve linear systems and achieved better performance than a direct application of SGD. In (Sebbouh et al., 2021), SHB was shown to have a convergence rate arbitrarily close to $o(1/\sqrt{T})$ on smooth convex objectives. However, the analysis stopped at this asymptotic bound and did not provide any practical implications of this result.

In contrast to all the aforementioned works, we provide positive results in theory to back up SHB's superior empirical performance, showing that SHB can yield accelerated convergence on quadratic objectives by equipping with large batch sizes and step decay learning rate schedules.

## 3   MAIN THEORY

### 3.1   PROBLEM SETUP

In this paper, we analyze quadratic objectives with the following form,

$$\min_{\mathbf{w}} f(\mathbf{w}) \triangleq \mathbb{E}_\xi\left[f(\mathbf{w},\xi)\right], \text{ where } f(\mathbf{w},\xi) = \frac{1}{2}\mathbf{w}^\top \mathbf{H}(\xi)\mathbf{w} - \mathbf{b}(\xi)^\top \mathbf{w}, \qquad (3.1)$$

where $\xi$ denotes the data sample. By setting gradient to $\mathbf{0}$, the optimum of $f(\mathbf{w})$ is obtained at

$$\mathbf{w}_* = \mathbf{H}^{-1}\mathbf{b}, \text{ where } \mathbf{H} = \mathbb{E}_\xi\left[\mathbf{H}(\xi)\right], \quad \mathbf{b} = \mathbb{E}_\xi\left[\mathbf{b}(\xi)\right]. \qquad (3.2)$$

In addition, we denote the smallest/largest eigenvalue and condition number of the Hessian $\mathbf{H}$ to be

$$\mu \triangleq \lambda_{\min}(\mathbf{H}), \quad L \triangleq \lambda_{\max}(\mathbf{H}), \quad \kappa \triangleq L/\mu, \qquad (3.3)$$

where eigenvalues from largest to smallest are denoted as

$$L = \lambda_1 \geq \lambda_2 \geq \cdots \geq \lambda_d = \mu > 0.$$

We consider the standard stochastic approximation framework (Kushner and Clark, 2012) and denote the gradient noise to be

$$\mathbf{n}_t \triangleq \nabla f(\mathbf{w}_t) - \nabla_{\mathbf{w}} f(\mathbf{w}_t, \xi). \qquad (3.4)$$

Throughout the paper, the following assumptions are adopted.

**Assumption 1.** *(Independent gradient noise)*

$$\{\mathbf{n}_t\} \text{ are pairwise independent.} \qquad (3.5)$$

**Assumption 2.** *(Unbiased gradient noise)*

$$\mathbb{E}\left[\mathbf{n}_t\right] = \mathbf{0}. \qquad (3.6)$$

**Assumption 3.** *(Anisotropic gradient noise)*

$$\mathbb{E}\left[\mathbf{n}_t\mathbf{n}_t^\top\right] \preceq \sigma^2\mathbf{H}. \qquad (3.7)$$

The anisotropic gradient noise assumption has been adopted by several past literatures (Dieuleveut et al., 2017; Pan et al., 2021), along with evidence supported in (Zhu et al., 2019; Sagun et al., 2018; Zhang et al., 2018; Jastrzębski et al., 2017; Wu et al., 2022b), which suggest that gradient noise covariance is normally close to the Hessian in neural networks training.

Let $\mathcal{B}_t$ be the minibatch of samples at iteration $t$. For simplicity, we only consider the setting where all minibatches share the same batch size

$$|\mathcal{B}_t| \equiv M, \text{ for } \forall t = 0, 1, \ldots, T - 1. \tag{3.8}$$

It follows that the number of samples is $N = MT$.

**Remark 1.** *One may also employ the common assumptions on strongly convex least square regressions as (Bach and Moulines, 2013; Jain et al., 2018a; Ge et al., 2019):*

$$\min_w f(\mathbf{w}), \quad \text{where } f(\mathbf{w}) \stackrel{def}{=} \frac{1}{2} \mathbb{E}_{(\mathbf{x},y) \sim \mathcal{D}} \left[ (y - \langle \mathbf{x}, \mathbf{w} \rangle)^2 \right], \text{ and}$$

$$\textit{(1) } y = \mathbf{w}_*^T \mathbf{x} + \epsilon, \text{ where } \mathbb{E}_{(\mathbf{x},y) \sim \mathcal{D}} \left[ \epsilon^2 \mathbf{x} \mathbf{x}^\top \right] \preceq \tilde{\sigma}^2 \mathbf{H}, \tag{3.9}$$

$$\textit{(2) } \mathbb{E} \left[ \|\mathbf{x}\|^2 \mathbf{x} \mathbf{x}^\top \right] \preceq R^2 \mathbf{H}$$

*which can also be translated into our settings under the compact set constraint $\mathbf{w} \in \Lambda$, as suggested in (Jain et al., 2018a).*

## 3.2 SUBOPTIMALITY OF SGD

We begin with the vanilla version of SGD,

$$\mathbf{w}_{t+1} = \mathbf{w}_t - \frac{\eta_t}{|\mathcal{B}_t|} \sum_{\xi \in \mathcal{B}_t} \nabla_\mathbf{w} f(\mathbf{w}_t, \xi), \tag{3.10}$$

whose theoretical property is well understood on quadratic objectives (Bach and Moulines, 2013; Jain et al., 2018b; Ge et al., 2019; Pan et al., 2021). Here $\eta_t$ means the learning rate at iteration $t$. It is known that SGD requires at least $\Omega(\kappa)$ iterations under the setting of batch size $M = 1$ (Jain et al., 2018a), nevertheless, whether this lower bound still holds for large batch settings is not rigorously claimed yet. Here we provide Theorem 1 to make things clearer.

**Theorem 1.** *There exist quadratic objectives $f(\mathbf{w})$ and initialization $\mathbf{w}_0$, no matter how large the batch size is or what learning rate scheduler is used, as long as $\eta_t \leq 2/L$ for $\forall t = 0, 1, \ldots, T - 1$, running SGD for $T$ iterations will result in*

$$\mathbb{E} \left[ f(\mathbf{w}_T) - f(\mathbf{w}_*) \right] \geq \frac{f(\mathbf{w}_0) - f(\mathbf{w}_*)}{2} \cdot \exp\left( -\frac{8T}{\kappa} \right)$$

The proof is available in Appendix B. The existence of those counterexamples suggests that in the worst case, SGD requires at least $T \geq \kappa/8 \cdot \ln(c/2) = \Omega(\kappa \log c)$ iterations to reduce the excess risk by a factor of $c \geq 2$, while in practice, $\kappa$ can be quite large near the converged point (Sagun et al., 2017; Arjevani and Field, 2020; Yao et al., 2020).

## 3.3 ACCELERATION WITH STOCHASTIC HEAVY BALL

To overcome this limitation, heavy-ball momentum (Polyak, 1964) is normally adopted by engineers to speed up SGD, equipped with various types of learning rate schedulers

$$\mathbf{v}_{t+1} = \beta \mathbf{v}_t + \frac{\eta_t}{|\mathcal{B}_t|} \sum_{\xi \in \mathcal{B}_t} \nabla_\mathbf{w} f(\mathbf{w}_t, \xi)$$

$$\mathbf{w}_{t+1} = \mathbf{w}_t - \mathbf{v}_{t+1}. \tag{3.11}$$

Despite its huge success in practice, the theoretical understanding of this method is still limited, especially for quadratic objectives. Furthermore, although it was widely recognized that stochastic heavy ball should provide acceleration in large batch settings, positive theoretical results so far are still insufficient to clearly account for that. We attempt to fill this gap.

In this section, we will show that SHB equipped with proper learning rate schedules can indeed speed up large batch training. The whole analysis is done in a general multistage learning rate scheduler framework, as shown in Algorithm 1. Specifically, in this framework, learning rates are divided into $n$ stages, with each stages' learning rates and number of iterations being $\eta'_\ell$ and $k_\ell \triangleq K$ respectively, i.e.

$$t_\ell^{(\text{start})} \triangleq K(\ell - 1), \quad t_\ell^{(\text{end})} \triangleq K\ell - 1$$
$$\eta_t \equiv \eta'_\ell, \quad \text{for } \forall t = t_\ell^{(\text{start})}, t_\ell^{(\text{start})} + 1, \ldots, t_\ell^{(\text{end})}. \tag{3.12}$$

---

**Algorithm 1** Multistage Stochastic Heavy Ball with minibatch

---

**Input:** Number of stages $n$, learning rates $\{\eta'_\ell\}_{\ell=1}^n$, momentum $\beta$, stage lengths $K$, minibatch size $M$, initialization $\mathbf{w}_0 \in \mathbb{R}^d$ and $\mathbf{v}_0 = \mathbf{0}$.

1: $t \leftarrow 0$ ▷ Iteration counter
2: **for** $\ell = 1, 2, \ldots, n$ **do**
3:     $\eta_t \leftarrow \eta'_\ell$
4:     **for** $i = 1, 2, \ldots, K$ **do**
5:         Sample a minibatch $\mathcal{B}$ uniformly from the training data
6:         $\mathbf{g}_t \leftarrow \frac{1}{M} \sum_{\xi \in \mathcal{B}} \nabla_\mathbf{w} f(\mathbf{w}, \xi)$ ▷ Mean gradient over a minibatch
7:         $\mathbf{v}_{t+1} \leftarrow \beta\mathbf{v}_t + \eta_t\mathbf{g}_t$
8:         $\mathbf{w}_{t+1} \leftarrow \mathbf{w}_t - \mathbf{v}_{t+1}$
9:         $t \leftarrow t + 1$
10: **return** $\mathbf{w}_t$ ▷ Last iterate

---

Given the above step decay scheduler, the following theorem states the convergence rate for SHB on quadratic objectives. To the best of our knowledge, this is the first non-asymptotic result that explicitly expresses the relationship between $T$ and the convergence rate of mutlistage SHB on quadratic objectives.

**Theorem 2.** *Given a quadratic objective $f(\mathbf{w})$ and a step decay learning rate scheduler with $\beta = (1 - 1/\sqrt{\kappa})^2$ with $\kappa \geq 4$, and $n \equiv T/K$ with settings that*

*1. decay factor $C$*

$$1 < C \leq T\sqrt{\kappa}. \tag{3.13}$$

*2. stepsize $\eta'_\ell$*

$$\eta'_\ell = \frac{1}{L} \cdot \frac{1}{C^{\ell-1}} \tag{3.14}$$

*3. stage length $K$*

$$K = \frac{T}{\log_C\left(T\sqrt{\kappa}\right)} \tag{3.15}$$

*4. total iteration number $T$*

$$\frac{T}{\ln\left(2^{14}T^8\right) \cdot \ln\left(2^6T^4\right) \cdot \log_C(T^2)} \geq 2C\sqrt{\kappa}, \tag{3.16}$$

*then such scheduler exists, and the output of Algorithm 1 satisfies*

$$\mathbb{E}[f(\mathbf{w}_T) - f(\mathbf{w}_*)] \leq \mathbb{E}[f(\mathbf{w}_0) - f(\mathbf{w}_*)] \cdot \exp\left(15\ln 2 + 2\ln T + 2\ln\kappa - \frac{2T}{\sqrt{\kappa}\log_C\left(T\sqrt{\kappa}\right)}\right)$$
$$+ \frac{4096C^2 d\sigma^2}{MT} \ln^2\left(2^6T^4\right) \cdot \log_C^2\left(T\sqrt{\kappa}\right).$$

Or equivalently, the result can be simplified to the following corollary.

**Corollary 3.** *Given a quadratic objective $f(\mathbf{w})$ and a step decay learning rate scheduler and momentum defined in Theorem 2, with $T \geq \tilde{\Omega}(\sqrt{\kappa})$ and $\kappa \geq 4$, the output of Algorithm 1 satisfies*

$$\mathbb{E}\left[f(\mathbf{w}_T) - f(\mathbf{w}_*)\right] \leq \mathbb{E}\left[f(\mathbf{w}_0) - f(\mathbf{w}_*)\right] \cdot \exp\left(-\tilde{\Omega}\left(\frac{T}{\sqrt{\kappa}}\right)\right) + \tilde{\mathcal{O}}\left(\frac{d\sigma^2}{MT}\right),$$

*where $\tilde{\mathcal{O}}(\cdot)$ and $\tilde{\Omega}(\cdot)$ are used to hide the log factors.*

Notice that the bias term $[f(\mathbf{w}_0) - f(\mathbf{w}_*)] \cdot \exp(-\tilde{\Omega}(T/\sqrt{\kappa}))$ is exponentially decreasing after $T = \tilde{\mathcal{O}}(\sqrt{\kappa})$ iterations, while the variance term can be bounded by $\tilde{\mathcal{O}}(1/T)$. This implies that under the large batch setting, if the batch size is large enough to counteract the extra constant in the variance term, accelerated convergence will be possible as compared to the iteration number of $\tilde{O}(\kappa)$ required by SGD. It is worth noting that this $\tilde{\Theta}(\sqrt{\kappa})$ acceleration is only log factors away from the optimal acceleration (Nemirovski, 1995) of Heavy Ball (Polyak, 1964) and Nesterov Accelerated Gradient (Nesterov, 1983) in deterministic case.

The proof outline can be split into two major steps. The first step is bias-variance decomposition, which decomposes the expected excess risk $\mathbb{E}[f(\mathbf{w}_T)] - f(\mathbf{w}_*)$ into two terms: bias and variance, where bias measures the deterministic convergence error and variance measures the effect of the gradient noise. This step adapts the well-known bias-variance decomposition technique of SGD (Bach and Moulines, 2013; Jain et al., 2018b; Ge et al., 2019; Pan et al., 2021) to SHB. Inside the adapted decomposition, a critical "contraction" term $\|\mathbf{T}_{T-1}\mathbf{T}_{T-2}...\mathbf{T}_0\|$ is introduced in both bias and variance, where each matrix $\mathbf{T}_t \in \mathbb{R}^{2\times2}$ depends on step size $\eta_t$ and differs only by a diagonal matrix $\boldsymbol{\Delta}_t \triangleq \mathbf{T}_t - \mathbf{T}_0 = \text{diag}(\delta_t, 0)$.

The second major step is to bound the matrix product tightly. Notice that this term has a form of $\prod_{t=0}^{T-1}(1 - \eta_t\lambda_j)$ for SGD and is much easier to analyze. For the general form of $\|\mathbf{T}_{T-1}\mathbf{T}_{T-2}...\mathbf{T}_0\| = \|(\mathbf{T}_0 + \boldsymbol{\Delta}_{T-1})(\mathbf{T}_0 + \boldsymbol{\Delta}_{T-2})...\mathbf{T}_0\|$, the major difficulty arises from the *non-commutative* matrix products of different $\mathbf{T}_t$'s and $\boldsymbol{\Delta}_t$'s. To overcome this obstacle, a novel technique is proposed in our paper, which is based on the special structure of $\mathbf{T}_t$. The key observation is that product with form $(\mathbf{T}_{s_1}\boldsymbol{\Delta}_{s_1'}\mathbf{T}_{s_2}\boldsymbol{\Delta}_{s_2'}\ldots\mathbf{T}_{s_n}\boldsymbol{\Delta}_{s_n'}) \in \mathbb{R}^{2\times2}$ retains two important properties: 1) The first column is always nonnegative and second column is always nonpositive; 2) The absolute value of each entry is a monotonical increasing function of $\delta_1, \ldots, \delta_{T-1}$. Hence the sum of the exponential number of terms in the binomial-like expansion also retains those two properties, which leads to a bound $\|\mathbf{T}_{T-1}\mathbf{T}_{T-2}...\mathbf{T}_0\| \leq \|\mathbf{T}_{T-1}^T\|$ tight under certain conditions. This key technique, as rigorously stated in Lemma 8 in Appendix, combined with subtle analysis of $\mathbf{T}_t$ and learning rate schedule techniques in (Ge et al., 2019; Pan et al., 2021), gives birth to Theorem 2. The full detail of the proof is provided Appendix C.

## 4 EXPERIMENTS

To verify our theoretical findings, two sets of experiments are conducted. The first one is ridge regression, which has a quadratic loss objective and is closer our theoretical settings. The second one is image classification on CIFAR-10 (Krizhevsky et al., 2009) with ResNet18 (He et al., 2015), DenseNet121 (Huang et al., 2017) and MobilenetV2 (Sandler et al., 2018), which is more of a practical interest regarding our theory's potential applications.

### 4.1 RIDGE REGRESSION

In ridge regression, we consider the following setting

$$f(\mathbf{w}) = \frac{1}{n}\|\mathbf{X}\mathbf{w} - \mathbf{Y}\|_2^2 + \alpha\|\mathbf{w}\|_2^2, \tag{4.1}$$

whose optimum has an analytic form

$$\mathbf{w}_* = \left(\mathbf{X}^\top\mathbf{X} + n\alpha\mathbf{I}\right)^{-1}\mathbf{X}^\top\mathbf{Y}.$$

Therefore the optimum loss $f(\mathbf{w}_*)$ can be directly computed. We use `a4a`[1] dataset (Chang and Lin, 2011; Dua and Graff, 2017) to realize this setting, which contains $n = 4,781$ samples and $d = 123$ features.

In all of our experiments, we set the number of epochs to 100, so the total amount of data is $N = 478,100$. Besides, we set different batch sizes $M \in \{2048, 512, 128\}$, and initialize $\mathbf{w}_0$ from a uniform distribution $(-1, 1)^d$. The partial batch at the end of each epoch is not truncated, which means the total number of iterations $T = \lceil N/M \rceil$.

Regarding hyperparameter choices for each scheduler & method, we do grid searches according to Table 3 in Appendix A and report the best loss for each random seed. For all schedulers, we set $\eta_0 \in \{10^0, 10^{-1}, 10^{-2}, 10^{-3}\}$. As for the choice of momentum factor $\beta$, we set $\beta = 0.9$ for stochastic heavy ball methods.

Table 1: Training loss statistics of ridge regression in `a4a` dataset over 5 runs.

| Methods/Schedules | $(f(\mathbf{w}) - f(\mathbf{w}_*)) \times 10^{-2}$ | | | |
|---|---|---|---|---|
| | Batch size $M = 512$ | $M = 128$ | $M = 32$ | $M = 8$ |
| SGD + constant $\eta_t$ | 2.10±0.46 | 1.17±0.81 | 1.27±0.27 | 0.94±0.83 |
| SGD + step decay | 2.44±0.45 | 0.64±0.04 | 0.11±0.01 | **0.04±0.04** |
| SHB + constant $\eta_t$ | 0.86±0.55 | 0.55±0.26 | 1.03±0.35 | 0.97±0.58 |
| SHB + step decay | **0.13±0.03** | **0.01±0.00** | **0.03±0.02** | 0.06±0.05 |

As shown in Table 1, one can observe that SHB are generally much better than SGD under large batch settings, and the step decay schedule always helps. The role of learning rate schedule and heavy-ball momentum is especially evident under the setting of $M = 512$, where SHB is able to greatly reduce the loss with a much smaller bias, but still has a large loss due to the existence of variance. This variance term is then further handled by step decay schedule and leads to a fast convergence. As the batch size decreases, the variance term becomes dominant, which explains the closing gap between SGD and SHB.

## 4.2 Image Classification on CIFAR-10

In image classification, our key focus is still verifying the superiority of SHB over SGD, so no heavy tuning was done for $\beta$. We follow the common practice of $\beta = 0.9$ for our algorithm in Theorem 2. To simulate the practical settings of distributed learning and federated learning, we restrict the number of iterations to be a few thousands (Kairouz et al., 2021), which roughly translated into $\#Epoch = 10$ for batch size $M = 128$ and $\#Epoch = 100$ for batch size $M = 2048$. On top of that, for batch size $M = 2048$, we replicate 16 nodes with micro batch size 128 on each node, hence the performance on distributed learning can be further simulated.

In this experiment, CIFAR-10 (Krizhevsky et al., 2009) dataset is adopted, which contains $50,000$ training samples and $10,000$ test samples. We use $5,000$ randomly chosen samples in the training set to form a validation set, then conduct grid searches by training on the remaining $45,000$ samples and selecting the hyperparameter with the best validation accuracy. The selected hyperparameter is then used for training the whole $50,000$ samples and testing on the test set. The final test results are thereby summarized in Table 2. For grid searches, we choose learning rate $\eta_0 \in \{1, 0.1, 0.01, 0.001\}$, with decay rate $\gamma \in \{1/2, 1/5, 1/10\}$ and number of intervals $n \in \{3, 4, 5, 6\}$.

One can observe in Table 2 and Figure 1 that under the large batch setting, SHB provides huge acceleration over SGD and achieves a significant performance improvement. This offers empirical evidence for our theory and suggests its practical value: *Heavy Ball Momentum can provide true acceleration for SGD under large-batch settings.*

---

[1]The dataset is accessible in `https://www.csie.ntu.edu.tw/~cjlin/libsvmtools/datasets/binary.html#a4a`.

Table 2: CIFAR-10: training losses and test accuracy of different methods over 5 trials.

| Setting | Method | Resnet18 | | DenseNet121 | | MobilenetV2 | |
| --- | --- | --- | --- | --- | --- | --- | --- |
| | | Crossent. Loss | Acc(%) | Crossent. Loss | Acc(%) | Crossent. Loss | Acc(%) |
| $M = 128$ | SGD | 0.46±0.01 | 81.19±0.93 | 0.22±0.01 | 88.58±0.23 | 0.45±0.00 | 82.90±0.37 |
| ($\#Epoch = 10$) | SHB | **0.38±0.08** | **85.16±2.30** | **0.18±0.00** | **88.63±0.27** | **0.35±0.01** | **86.23±0.23** |
| $M = 128 \times 16$ | SGD | 0.33±0.01 | 83.82±0.42 | 0.01±0.00 | 89.28±0.23 | 0.32±0.02 | 84.37±0.77 |
| ($\#Epoch = 100$) | SHB | **0.01±0.00** | **89.78±0.23** | **0.00±0.00** | **92.46±0.15** | **0.07±0.01** | **89.57±0.18** |

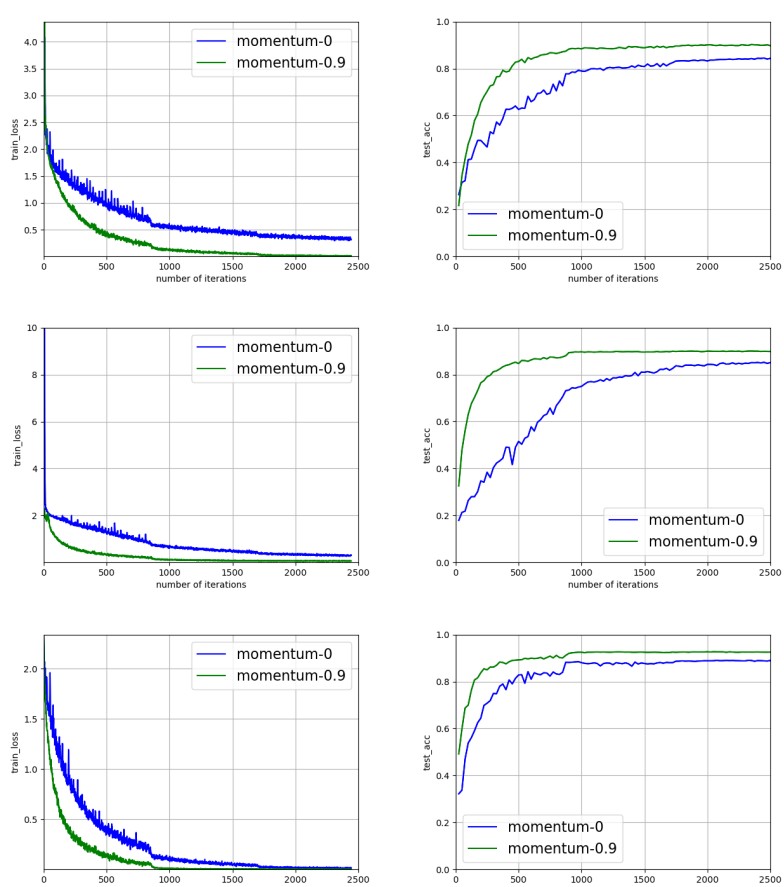

Figure 1: CIFAR-10 training statistics of batch size $M = 128 \times 16$ and $\#Epoch = 100$ on Resnet18, DenseNet121 and MobilenetV2 (from top to bottom). **Left**: Training loss; **Right**: Test accuracy.

## 5 CONCLUSION

In this paper, we present a non-asymptotic convergence rate for Stochastic Heavy Ball with step decay learning rate schedules on quadratic objectives. The proposed result demonstrates SHB's superiority over SGD under large-batch settings. To the best of our knowledge, this is the first time that the convergence rate of SHB is explicitly expressed in terms of iteration number $T$ given decaying learning rates on quadratic objectives. Theoretically, our analysis provides techniques general enough to analyze any multi-stage schedulers with SHB on quadratics. Empirically, we demonstrate the practical benefits of heavy-ball momentum for accelerating large-batch training, which matches our theoretical prediction and explains heavy-ball momentum's effectiveness in practice to a certain degree.

## ACKNOWLEDGMENTS

Rui Pan acknowledges support from the Hong Kong PhD Fellowship Scheme (HKPFS).

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

## A  EXPERIMENTAL DETAILS

The detailed learning rate schedule forms and grid search hyperparameters for ridge regression in Section 4.1 are listed in Table 3.

Table 3: Grid-search hyperparameter choices for ridge regression.

| Scheduler | Form | Hyperparameter choices |
|---|---|---|
| Constant | $\eta_t = \eta_0$ | - |
| Step decay | $\eta_t = \eta_0 \cdot \gamma^\ell$, if $t \in [\ell, \ell+1) \cdot \frac{T}{n}$ | $n \in \{2, 3, 4, 5\}$ $\gamma \in \{\frac{1}{2}, \frac{1}{4}, \frac{1}{8}\}$ |

## B  PROOF FOR SECTION 3.2: SUBOPTIMALITY OF SGD

**Theorem 1.** *There exist quadratic objectives $f(\mathbf{w})$ and initialization $\mathbf{w}_0$, no matter how large the batch size is or what learning rate scheduler is used, as long as $\eta_t \leq 2/L$ for $\forall t = 0, 1, \ldots, T-1$, running SGD for $T$ iterations will result in*

$$\mathbb{E}\left[f(\mathbf{w}_T) - f(\mathbf{w}_*)\right] \geq \frac{f(\mathbf{w}_0) - f(\mathbf{w}_*)}{2} \cdot \exp\left(-\frac{8T}{\kappa}\right)$$

*Proof.* Consider the case of $\kappa \geq 4$, $\mathbf{w}_0 = c_0 \cdot \mathbf{I}$, $d \geq \kappa + 1$, $\mathbf{H}(\xi) \equiv \mathbf{H} = \text{diag}(L, \mu, \mu, \ldots, \mu)$ and $\mathbf{b}(\xi) \equiv \mathbf{0}$, then according to SGD's update formula in Eqn. (3.10)

$$\mathbf{w}_{t+1} = \mathbf{w}_t - \frac{\eta_t}{|\mathcal{B}_t|} \sum_{\xi \in \mathcal{B}_t} \nabla_{\mathbf{w}} f(\mathbf{w}_t, \xi) = (\mathbf{I} - \eta_t \mathbf{H})\mathbf{w}_t,$$

we have the update of $j$-th $(j \geq 2)$ entry being

$$w_{t+1,j} = (1 - \eta_t \lambda_j)w_{t,j} = (1 - \eta_t \mu)w_{t,j} \geq \left(1 - \frac{2\mu}{L}\right) w_{t,j} = \left(1 - \frac{2}{\kappa}\right) w_{t,j} \geq \exp\left(-\frac{4}{\kappa}\right) w_{t,j},$$

where the first inequality comes from $\eta_t \leq 2/L$ and the second inequality is entailed by $1 - x \geq \exp(-2x)$ for $x = 2/\kappa \in [0, 1/2]$, since for $\forall x \in [0, 1/2]$

$$g(x) \triangleq \ln(1 - x) - \ln(\exp(-2x)), \quad g(0) = 0, \quad \frac{\partial g(x)}{\partial x} = -\frac{1}{1 - x} + 2 \geq 0$$

$$\Rightarrow \quad g(x) \geq g(0) = 0 \quad \Rightarrow \ln(1 - x) \geq \ln(\exp(-2x)) \quad \Rightarrow 1 - x \geq \exp(-2x).$$

It follows,

$$f(\mathbf{w}_T) - f(\mathbf{w}_*) = f(\mathbf{w}_T) = \frac{1}{2}\mathbf{w}_T^\top \mathbf{H}\mathbf{w}_T = \frac{1}{2}\left(Lw_{T,1}^2 + \mu \sum_{j=2}^{d} w_{T,j}^2\right)$$

$$\geq \frac{\mu}{2}\sum_{j=2}^{d} w_{T,j}^2 = \frac{\mu}{2}\sum_{j=2}^{d} w_{0,j}^2 \prod_{t=0}^{T-1}(1 - \eta_t \lambda_j)^2$$

$$\geq \frac{\mu}{2}\sum_{j=2}^{d} w_{0,j}^2 \exp\left(-\frac{8T}{\kappa}\right)$$

$$= \frac{\mu}{2}(d-1) \cdot c_0^2 \cdot \exp\left(-\frac{8T}{\kappa}\right) = \mu(d-1) \cdot \frac{f(\mathbf{w}_0) - f(\mathbf{w}_*)}{L + \mu(d-1)} \cdot \exp\left(-\frac{8T}{\kappa}\right)$$

$$= \frac{f(\mathbf{w}_0) - f(\mathbf{w}_*)}{\frac{L}{\mu(d-1)} + 1} \cdot \exp\left(-\frac{8T}{\kappa}\right)$$

$$\geq \frac{f(\mathbf{w}_0) - f(\mathbf{w}_*)}{2} \cdot \exp\left(-\frac{8T}{\kappa}\right).$$

□

# C    PROOF FOR SECTION 3.3: ACCELERATION WITH STOCHASTIC HEAVY BALL

We provide the broad stroke of our proofs here. The ultimate goal is bounding $\mathbb{E}[f(\mathbf{w}_T) - f(\mathbf{w}_*)]$ in terms of $T, \kappa$ and other fixed model parameters. To achieve this goal, we conduct the whole analysis in a layer-by-layer fashion. In each layer, we translate the terms in the previous layer into the terms of the current layer, specifically

- Appendix C.1: $\mathbb{E}[f(\mathbf{w}) - f(\mathbf{w}_*)] \Rightarrow \left\|\mathbf{T}_{t,j}^k\right\|$ for some special $2 \times 2$ matrix $\mathbf{T}_{t,j}$ via bias-variance decomposition.
- Appendix C.2: $\|\mathbf{T}_{t+1,j}\mathbf{T}_{t+2,j}...\mathbf{T}_{t+k,j}\| \Rightarrow \rho(\mathbf{T}_{t+k,j})^k$ via analyzing the property of $\left\|\mathbf{T}_{t,j}^k\right\|$ and $\|\mathbf{T}_{t+1,j}\mathbf{T}_{t+2,j}...\mathbf{T}_{t+k,j}\|$.
- Appendix C.3: $\rho(\mathbf{T}_{t,j})^k \Rightarrow \{\eta_t, \beta, \lambda_j\}$ via analyzing the property of $\rho(\mathbf{T}_{t,j})$.
- Appendix C.4 $\{\eta_t, \beta, \lambda_j\} \Rightarrow \{\eta'_\ell, \beta, \lambda_j\} \Rightarrow \{T, \kappa, \dots\}$ via specializing the scheduler with multistage scheduler step decay.

## C.1    BIAS VARIANCE DECOMPOSITION: $\mathbb{E}[f(\mathbf{w}) - f(\mathbf{w}_*)] \Rightarrow \left\|\mathbf{T}_{t,j}^k\right\|$

The section presents the lemmas that decompose the loss into bias and variance, expressing them in terms of the norm of product of a series of special $2 \times 2$ matrices $\|\prod_t \mathbf{T}_{t,j}\|$'s, specifically $\|\mathbf{T}_{T-1,j}\mathbf{T}_{T-2,j}\dots\mathbf{T}_{\tau+1,j}\|$ for some $\tau = 0, 1, \dots, T-1$. For simplicity, we only consider the case of batch size $M = 1$. The case of larger batches is equivalent to replacing noise term $\sigma^2$ with $\sigma^2/M$.

**Lemma 1.** *(Quadratic excess risk is **H**-norm) Given a quadratic objective $f(\mathbf{w})$, we have*

$$f(\mathbf{w}) - f(\mathbf{w}_*) = \frac{1}{2}\|\mathbf{w} - \mathbf{w}_*\|_{\mathbf{H}}^2 \triangleq \frac{1}{2}(\mathbf{w} - \mathbf{w}_*)^\top \mathbf{H}(\mathbf{w} - \mathbf{w}_*) \tag{C.1}$$

*Proof.* It holds that

$$
\begin{aligned}
f(\mathbf{w}) - f(\mathbf{w}_*) &\overset{(3.1)}{=} \mathbb{E}_\xi\left[\frac{1}{2}\mathbf{w}^\top \mathbf{H}(\xi)\mathbf{w} - \mathbf{b}(\xi)^\top \mathbf{w}\right] - \mathbb{E}_\xi\left[\frac{1}{2}\mathbf{w}_*^\top \mathbf{H}(\xi)\mathbf{w}_* - \mathbf{b}(\xi)^\top \mathbf{w}_*\right] \\
&= \left[\frac{1}{2}\mathbf{w}^\top \mathbb{E}[\mathbf{H}(\xi)]\mathbf{w} - \mathbb{E}[\mathbf{b}(\xi)]^\top \mathbf{w}\right] - \left[\frac{1}{2}\mathbf{w}_*^\top \mathbb{E}[\mathbf{H}(\xi)]\mathbf{w}_* - \mathbb{E}[\mathbf{b}(\xi)]^\top \mathbf{w}_*\right] \\
&\overset{(3.2)}{=} \left[\frac{1}{2}\mathbf{w}^\top \mathbf{H}\mathbf{w} - \mathbf{b}^\top \mathbf{w}\right] - \left[\frac{1}{2}\mathbf{w}_*^\top \mathbf{H}\mathbf{w}_* - \mathbf{b}^\top \mathbf{w}_*\right] \\
&\overset{(3.2)}{=} \left[\frac{1}{2}\mathbf{w}^\top \mathbf{H}\mathbf{w} - (\mathbf{H}\mathbf{w}_*)^\top \mathbf{w}\right] - \left[\frac{1}{2}\mathbf{w}_*^\top \mathbf{H}\mathbf{w}_* - (\mathbf{H}\mathbf{w}_*)^\top \mathbf{w}_*\right] \\
&= \frac{1}{2}\left[\mathbf{w}^\top \mathbf{H}\mathbf{w} - \mathbf{w}^\top \mathbf{H}\mathbf{w}_* - \mathbf{w}_*^\top \mathbf{H}\mathbf{w} + \mathbf{w}_*^\top \mathbf{H}\mathbf{w}_*\right] \\
&= \frac{1}{2}(\mathbf{w} - \mathbf{w}_*)^\top \mathbf{H}(\mathbf{w} - \mathbf{w}_*).
\end{aligned}
$$

Here the fifth equality comes from $\mathbf{H} = \mathbf{H}^\top$ being a symmetric matrix, and $\mathbf{w}_*^\top \mathbf{H}\mathbf{w} = (\mathbf{w}_*^\top \mathbf{H}\mathbf{w})^\top = \mathbf{w}^\top \mathbf{H}\mathbf{w}_*$ being a scalar. □

**Lemma 2.** *(Bias-variance decomposition) Given a quadratic function $f(\mathbf{w})$, if batch size $M = 1$, after running $T$ iterations of Algorithm 1, we have the last iterate $\mathbf{w}_T$ satisfying*

$$
\begin{aligned}
&\mathbb{E}\left[\|\mathbf{w}_T - \mathbf{w}_*\|_{\mathbf{H}}^2\right] \\
&= \mathbb{E}\left[\begin{bmatrix}\mathbf{w}_0 - \mathbf{w}_* \\ \mathbf{w}_{-1} - \mathbf{w}_*\end{bmatrix}^\top \mathbf{M}_0^\top \mathbf{M}_1^\top \dots \mathbf{M}_{T-1}^\top \begin{bmatrix}\mathbf{H} & \mathbf{O} \\ \mathbf{O} & \mathbf{O}\end{bmatrix} \mathbf{M}_{T-1}\mathbf{M}_{T-2}\dots\mathbf{M}_0 \begin{bmatrix}\mathbf{w}_0 - \mathbf{w}_* \\ \mathbf{w}_{-1} - \mathbf{w}_*\end{bmatrix}\right] \\
&\quad + \sum_{\tau=0}^{T-1}\mathbb{E}\left[\eta_\tau^2 \begin{bmatrix}\mathbf{n}_\tau \\ \mathbf{0}\end{bmatrix}^\top \mathbf{M}_{\tau+1}^\top \mathbf{M}_{\tau+2}^\top \dots \mathbf{M}_{T-1}^\top \begin{bmatrix}\mathbf{H} & \mathbf{O} \\ \mathbf{O} & \mathbf{O}\end{bmatrix} \mathbf{M}_{T-1}\mathbf{M}_{T-2}\dots\mathbf{M}_{\tau+1} \begin{bmatrix}\mathbf{n}_\tau \\ \mathbf{0}\end{bmatrix}\right],
\end{aligned} \tag{C.2}
$$

*where $\mathbf{w}_{-1} = \mathbf{w}_0 + \mathbf{v}_0$ and*

$$\mathbf{M}_t \triangleq \begin{bmatrix} (1+\beta)\mathbf{I} - \eta_t \mathbf{H} & -\beta\mathbf{I} \\ \mathbf{I} & \mathbf{O} \end{bmatrix}. \tag{C.3}$$

*Proof.* Denote

$$\tilde{\mathbf{w}}_t \triangleq \begin{bmatrix} \mathbf{w}_t - \mathbf{w}_* \\ \mathbf{w}_{t-1} - \mathbf{w}_* \end{bmatrix} \in \mathbb{R}^{2d}, \quad \tilde{\mathbf{n}}_t \triangleq \begin{bmatrix} \mathbf{n}_t \\ \mathbf{0} \end{bmatrix} \in \mathbb{R}^{2d}, \quad \tilde{\mathbf{H}} \triangleq \begin{bmatrix} \mathbf{H} & \mathbf{O} \\ \mathbf{O} & \mathbf{O} \end{bmatrix} \in \mathbb{R}^{2d \times 2d} \tag{C.4}$$

as the extended parameter difference, extended gradient noise and extended Hessian matrix, then according to momentum's update formula in Eqn. (3.11), or line 7-8 in Algorithm 1 with batch size $M = 1$, we have

$$
\begin{aligned}
\mathbf{v}_{t+1} &= \beta\mathbf{v}_t + \eta_t \nabla_{\mathbf{w}} f(\mathbf{w}_t, \xi) \\
\mathbf{w}_{t+1} &= \mathbf{w}_t - \mathbf{v}_{t+1} \\
\Rightarrow \quad \mathbf{w}_{t+1} - \mathbf{w}_* &= \mathbf{w}_t - \mathbf{w}_* - \mathbf{v}_{t+1} \\
&= \mathbf{w}_t - \mathbf{w}_* - (\beta\mathbf{v}_t + \eta_t \nabla_{\mathbf{w}} f(\mathbf{w}_t, \xi)) \\
&= \mathbf{w}_t - \mathbf{w}_* - \beta(\mathbf{w}_{t-1} - \mathbf{w}_t) - \eta_t \nabla_{\mathbf{w}} f(\mathbf{w}_t, \xi) \\
&\overset{(3.4)}{=} \mathbf{w}_t - \mathbf{w}_* - \beta(\mathbf{w}_{t-1} - \mathbf{w}_t) - \eta_t (\nabla_{\mathbf{w}} f(\mathbf{w}_t) - \mathbf{n}_t) \\
&\overset{(3.2)}{=} \mathbf{w}_t - \mathbf{w}_* - \beta(\mathbf{w}_{t-1} - \mathbf{w}_t) - \eta_t (\mathbf{H}(\mathbf{w}_t - \mathbf{w}_*) - \mathbf{n}_t) \\
&= \mathbf{w}_t - \mathbf{w}_* - \beta[(\mathbf{w}_{t-1} - \mathbf{w}_*) - (\mathbf{w}_t - \mathbf{w}_*)] - \eta_t (\mathbf{H}(\mathbf{w}_t - \mathbf{w}_*) - \mathbf{n}_t) \\
&= [(1+\beta)\mathbf{I} - \eta_t \mathbf{H}](\mathbf{w}_t - \mathbf{w}_*) - \beta\mathbf{I}(\mathbf{w}_{t-1} - \mathbf{w}_*) + \eta_t \mathbf{n}_t \\
\Rightarrow \quad \tilde{\mathbf{w}}_{t+1} = \begin{bmatrix} \mathbf{w}_{t+1} - \mathbf{w}_* \\ \mathbf{w}_t - \mathbf{w}_* \end{bmatrix} &= \begin{bmatrix} (1+\beta)\mathbf{I} - \eta_t \mathbf{H} & -\beta\mathbf{I} \\ \mathbf{I} & \mathbf{O} \end{bmatrix} \begin{bmatrix} \mathbf{w}_t - \mathbf{w}_* \\ \mathbf{w}_{t-1} - \mathbf{w}_* \end{bmatrix} + \eta_t \begin{bmatrix} \mathbf{n}_t \\ \mathbf{0} \end{bmatrix} \\
&= \mathbf{M}_t \tilde{\mathbf{w}}_t + \eta_t \tilde{\mathbf{n}}_t.
\end{aligned}
$$

It follows

$$
\begin{aligned}
\tilde{\mathbf{w}}_{t+1} &= \mathbf{M}_t \tilde{\mathbf{w}}_t + \eta_t \tilde{\mathbf{n}}_t \\
&= \mathbf{M}_t \mathbf{M}_{t-1} \tilde{\mathbf{w}}_{t-1} + \eta_{t-1} \mathbf{M}_t \tilde{\mathbf{n}}_{t-1} + \eta_t \tilde{\mathbf{n}}_t \\
&= \mathbf{M}_t \mathbf{M}_{t-1} \mathbf{M}_{t-2} \tilde{\mathbf{w}}_{t-2} + \eta_{t-2} \mathbf{M}_t \mathbf{M}_{t-1} \tilde{\mathbf{n}}_{t-2} + \eta_{t-1} \mathbf{M}_t \tilde{\mathbf{n}}_{t-1} + \eta_t \tilde{\mathbf{n}}_t \\
&= \ldots \\
&= \mathbf{M}_t \mathbf{M}_{t-1} \ldots \mathbf{M}_0 \tilde{\mathbf{w}}_0 + \sum_{\tau=0}^{t} (\eta_\tau \mathbf{M}_t \mathbf{M}_{t-1} \ldots \mathbf{M}_{\tau+1} \tilde{\mathbf{n}}_\tau).
\end{aligned}
$$

where $\tilde{\mathbf{w}}_0 = \begin{bmatrix} \mathbf{w}_0 - \mathbf{w}_* \\ \mathbf{w}_{-1} - \mathbf{w}_* \end{bmatrix}$ and $\mathbf{w}_{-1} = \mathbf{w}_0 + \mathbf{v}_0$ which is associated with iteration $t = 0$.

We can decompose the above process into two parts

$$
\begin{aligned}
\tilde{\mathbf{w}}_{t+1}^{(b)} &= \mathbf{M}_t \tilde{\mathbf{w}}_t^{(b)} & \text{with } \tilde{\mathbf{w}}_0^{(b)} = \tilde{\mathbf{w}}_0 \\
\tilde{\mathbf{w}}_{t+1}^{(v)} &= \mathbf{M}_t \tilde{\mathbf{w}}_t^{(v)} + \eta_t \tilde{\mathbf{n}}_t & \text{with } \tilde{\mathbf{w}}_0^{(v)} = \mathbf{0},
\end{aligned} \tag{C.5}
$$

since

$$
\begin{aligned}
\tilde{\mathbf{w}}_{t+1}^{(b)} &= \mathbf{M}_t \mathbf{M}_{t-1} \ldots \mathbf{M}_0 \tilde{\mathbf{w}}_0^{(b)} = \mathbf{M}_t \mathbf{M}_{t-1} \ldots \mathbf{M}_0 \tilde{\mathbf{w}}_0 \\
\tilde{\mathbf{w}}_{t+1}^{(v)} &= \sum_{\tau=0}^{t} (\eta_\tau \mathbf{M}_t \mathbf{M}_{t-1} \ldots \mathbf{M}_{\tau+1} \tilde{\mathbf{n}}_\tau) \\
\Rightarrow \quad \tilde{\mathbf{w}}_{t+1} &= \tilde{\mathbf{w}}_{t+1}^{(b)} + \tilde{\mathbf{w}}_{t+1}^{(v)}.
\end{aligned} \tag{C.6}
$$

Furthermore, we have

$$
\begin{aligned}
\mathbb{E}\left[\left(\tilde{\mathbf{w}}_{t+1}^{(b)}\right)^{\top}\tilde{\mathbf{H}}\tilde{\mathbf{w}}_{t+1}^{(v)}\right] &= \left(\tilde{\mathbf{w}}_{t+1}^{(b)}\right)^{\top}\tilde{\mathbf{H}}\mathbb{E}\left[\tilde{\mathbf{w}}_{t+1}^{(v)}\right] \\
&\stackrel{(C.6)}{=} \left(\tilde{\mathbf{w}}_{t+1}^{(b)}\right)^{\top}\tilde{\mathbf{H}}\mathbb{E}\left[\sum_{\tau=0}^{t}\left(\eta_{\tau}\mathbf{M}_t\mathbf{M}_{t-1}\ldots\mathbf{M}_{\tau+1}\tilde{\mathbf{n}}_{\tau}\right)\right] \\
&= \left(\tilde{\mathbf{w}}_{t+1}^{(b)}\right)^{\top}\tilde{\mathbf{H}}\sum_{\tau=0}^{t}\left(\eta_{\tau}\mathbf{M}_t\mathbf{M}_{t-1}\ldots\mathbf{M}_{\tau+1}\mathbb{E}\left[\tilde{\mathbf{n}}_{\tau}\right]\right) \\
&\stackrel{(3.6)}{=} \mathbf{0}
\end{aligned}
\tag{C.7}
$$

and

$$
\begin{aligned}
&\mathbb{E}\left[\left(\tilde{\mathbf{w}}_{t+1}^{(v)}\right)^{\top}\tilde{\mathbf{H}}\tilde{\mathbf{w}}_{t+1}^{(v)}\right] \\
&= \mathbb{E}\left[\left(\sum_{\tau=0}^{t}\left(\eta_{\tau}\mathbf{M}_t\mathbf{M}_{t-1}\ldots\mathbf{M}_{\tau+1}\tilde{\mathbf{n}}_{\tau}\right)\right)^{\top}\tilde{\mathbf{H}}\left(\sum_{\tau'=0}^{t}\left(\eta_{\tau'}\mathbf{M}_t\mathbf{M}_{t-1}\ldots\mathbf{M}_{\tau'+1}\tilde{\mathbf{n}}_{\tau'}\right)\right)\right] \\
&= \sum_{\tau=0,\tau'=0}^{t}\mathbb{E}\left[\eta_{\tau}\eta_{\tau'}\tilde{\mathbf{n}}_{\tau}^{\top}\mathbf{M}_{\tau+1}^{\top}\mathbf{M}_{\tau+2}^{\top}\ldots\mathbf{M}_t^{\top}\tilde{\mathbf{H}}\mathbf{M}_t\mathbf{M}_{t-1}\ldots\mathbf{M}_{\tau'+1}\tilde{\mathbf{n}}_{\tau'}\right] \\
&= \sum_{\tau=0}^{t}\mathbb{E}\left[\eta_{\tau}^2\tilde{\mathbf{n}}_{\tau}^{\top}\mathbf{M}_{\tau+1}^{\top}\mathbf{M}_{\tau+2}^{\top}\ldots\mathbf{M}_t^{\top}\tilde{\mathbf{H}}\mathbf{M}_t\mathbf{M}_{t-1}\ldots\mathbf{M}_{\tau+1}\tilde{\mathbf{n}}_{\tau}\right] \\
&\quad + \sum_{\tau=0,\tau'=0,\tau\neq\tau'}^{t}\mathbb{E}\left[\eta_{\tau}\eta_{\tau'}\tilde{\mathbf{n}}_{\tau}^{\top}\mathbf{M}_{\tau+1}^{\top}\mathbf{M}_{\tau+2}^{\top}\ldots\mathbf{M}_t^{\top}\tilde{\mathbf{H}}\mathbf{M}_t\mathbf{M}_{t-1}\ldots\mathbf{M}_{\tau'+1}\tilde{\mathbf{n}}_{\tau'}\right] \\
&= \sum_{\tau=0}^{t}\mathbb{E}\left[\eta_{\tau}^2\tilde{\mathbf{n}}_{\tau}^{\top}\mathbf{M}_{\tau+1}^{\top}\mathbf{M}_{\tau+2}^{\top}\ldots\mathbf{M}_t^{\top}\tilde{\mathbf{H}}\mathbf{M}_t\mathbf{M}_{t-1}\ldots\mathbf{M}_{\tau+1}\tilde{\mathbf{n}}_{\tau}\right] + 0,
\end{aligned}
\tag{C.8}
$$

where the last equality is because

$$
\tilde{\mathbf{n}}_{\tau} \stackrel{(C.4)}{=} \begin{bmatrix}\mathbf{n}_{\tau}\\0\end{bmatrix} \text{ and } \tilde{\mathbf{n}}_{\tau'} \stackrel{(C.4)}{=} \begin{bmatrix}\mathbf{n}_{\tau'}\\0\end{bmatrix}
$$

are pairwise independent and have mean $\mathbf{0}$ given Assumption 1 and 2.

Then we obtain

$$
\begin{aligned}
&\mathbb{E}\left\|\mathbf{w}_T-\mathbf{w}_*\right\|_{\mathbf{H}}^2 \\
&\stackrel{(C.1)}{=} \left(\mathbf{w}_T-\mathbf{w}_*\right)\mathbf{H}\left(\mathbf{w}_T-\mathbf{w}_*\right) = \begin{bmatrix}\mathbf{w}_T-\mathbf{w}_*\\\mathbf{w}_{T-1}-\mathbf{w}_*\end{bmatrix}^{\top}\begin{bmatrix}\mathbf{H}&\mathbf{O}\\\mathbf{O}&\mathbf{O}\end{bmatrix}\begin{bmatrix}\mathbf{w}_T-\mathbf{w}_*\\\mathbf{w}_{T-1}-\mathbf{w}_*\end{bmatrix} \\
&\stackrel{(C.4)}{=} \mathbb{E}\left[\tilde{\mathbf{w}}_T^{\top}\tilde{\mathbf{H}}\tilde{\mathbf{w}}_T\right] \stackrel{(C.6)}{=} \mathbb{E}\left[\left(\tilde{\mathbf{w}}_T^{(b)}+\tilde{\mathbf{w}}_T^{(v)}\right)^{\top}\tilde{\mathbf{H}}\left(\tilde{\mathbf{w}}_T^{(b)}+\tilde{\mathbf{w}}_T^{(v)}\right)\right] \\
&= \mathbb{E}\left[\left(\tilde{\mathbf{w}}_T^{(b)}\right)^{\top}\tilde{\mathbf{H}}\left(\tilde{\mathbf{w}}_T^{(b)}\right)+\left(\tilde{\mathbf{w}}_T^{(v)}\right)^{\top}\tilde{\mathbf{H}}\left(\tilde{\mathbf{w}}_T^{(v)}\right)+2\left(\tilde{\mathbf{w}}_T^{(b)}\right)^{\top}\tilde{\mathbf{H}}\left(\tilde{\mathbf{w}}_T^{(v)}\right)\right] \\
&\stackrel{(C.7)}{=} \mathbb{E}\left[\left(\tilde{\mathbf{w}}_T^{(b)}\right)^{\top}\tilde{\mathbf{H}}\left(\tilde{\mathbf{w}}_T^{(b)}\right)\right]+\mathbb{E}\left[\left(\tilde{\mathbf{w}}_T^{(v)}\right)^{\top}\tilde{\mathbf{H}}\left(\tilde{\mathbf{w}}_T^{(v)}\right)\right] \\
&\stackrel{(C.6),\;(C.8)}{=} \mathbb{E}\left[\tilde{\mathbf{w}}_0\mathbf{M}_0^{\top}\mathbf{M}_1^{\top}\ldots\mathbf{M}_{T-1}^{\top}\tilde{\mathbf{H}}\mathbf{M}_{T-1}\mathbf{M}_{T-2}\ldots\mathbf{M}_0\tilde{\mathbf{w}}_0\right] \\
&\quad + \sum_{\tau=0}^{T-1}\mathbb{E}\left[\eta_{\tau}^2\tilde{\mathbf{n}}_{\tau}^{\top}\mathbf{M}_{\tau+1}^{\top}\mathbf{M}_{\tau+2}^{\top}\ldots\mathbf{M}_{T-1}^{\top}\tilde{\mathbf{H}}\mathbf{M}_{T-1}\mathbf{M}_{T-2}\ldots\mathbf{M}_{\tau+1}\tilde{\mathbf{n}}_{\tau}\right]
\end{aligned}
$$

Here the fourth equality is because $(\tilde{\mathbf{w}}_{t+1}^{(b)})^{\top}\tilde{\mathbf{H}}(\tilde{\mathbf{w}}_{t+1}^{(v)})$ is a scalar and $\tilde{\mathbf{H}}$ is symmetric. Replacing the extended terms $\tilde{\mathbf{w}}$, $\tilde{\mathbf{n}}$, $\tilde{\mathbf{H}}$ with their definitions in Eqn. (C.4), we get the desired form in Eqn. (C.2).

□

**Lemma 3.** *(Decomposing* $\mathrm{M}_t$ *into a block diagonal matrix) Given a matrix* $\mathbf{M}_t \in \mathbb{R}^{2d \times 2d}$ *defined in Eqn.* (C.3)*, we have*

$$\mathbf{M}_t = \mathbf{V}\mathbf{\Pi} \begin{bmatrix} \mathbf{T}_{t,1} & & & \\ & \mathbf{T}_{t,2} & & \\ & & \ddots & \\ & & & \mathbf{T}_{t,d} \end{bmatrix} \mathbf{\Pi}^\top \mathbf{V}^\top, \qquad (\text{C.9})$$

*where*

$$\mathbf{T}_{t,j} \triangleq \begin{bmatrix} 1 + \beta - \eta_t \lambda_j & -\beta \\ 1 & 0 \end{bmatrix} \in \mathbb{R}^{2 \times 2} \qquad (\text{C.10})$$

*and orthogonal matrices*

$$\mathbf{\Pi} \triangleq \begin{bmatrix} \mathbf{e}_1 & \mathbf{0} & \mathbf{e}_2 & \mathbf{0} & \dots & \mathbf{e}_d & \mathbf{0} \\ \mathbf{0} & \mathbf{e}_1 & \mathbf{0} & \mathbf{e}_2 & \dots & \mathbf{0} & \mathbf{e}_d \end{bmatrix} \in \mathbb{R}^{2d \times 2d}, \mathbf{V} \triangleq \begin{bmatrix} \mathbf{U} & \mathbf{O} \\ \mathbf{O} & \mathbf{U} \end{bmatrix} \in \mathbb{R}^{2d \times 2d}, \qquad (\text{C.11})$$

*given the eigendecomposition of* $\mathbf{H}$ *being*

$$\mathbf{H} = \mathbf{U}\mathbf{\Lambda}\mathbf{U}^\top \in \mathbb{R}^{d \times d}, \quad \left( \mathbf{U}^\top \mathbf{U} = \mathbf{I} \right) \qquad (\text{C.12})$$

*and standard unit vectors/standard basis being*

$$\mathbf{e}_i = [0 \dots 0 \underbrace{1}_{i-th} 0 \dots 0]^\top \in \mathbb{R}^{d \times 1} \qquad (\text{C.13})$$

*Proof.*

$$\mathbf{\Pi}^\top \mathbf{V}^\top \mathbf{M}_t \mathbf{V}\mathbf{\Pi}$$

$$\overset{(\text{C.3})}{=} \mathbf{\Pi}^\top \mathbf{V}^\top \begin{bmatrix} (1+\beta)\mathbf{I} - \eta_t \mathbf{H} & -\beta \mathbf{I} \\ \mathbf{I} & \mathbf{O} \end{bmatrix} \mathbf{V}\mathbf{\Pi}$$

$$\overset{(\text{C.11})}{=} \mathbf{\Pi}^\top \begin{bmatrix} \mathbf{U}^\top & \mathbf{O} \\ \mathbf{O} & \mathbf{U}^\top \end{bmatrix} \begin{bmatrix} (1+\beta)\mathbf{I} - \eta_t \mathbf{H} & -\beta \mathbf{I} \\ \mathbf{I} & \mathbf{O} \end{bmatrix} \begin{bmatrix} \mathbf{U} & \mathbf{O} \\ \mathbf{O} & \mathbf{U} \end{bmatrix} \mathbf{\Pi}$$

$$\overset{(\text{C.12})}{=} \mathbf{\Pi}^\top \begin{bmatrix} (1+\beta)\mathbf{I} - \eta_t \mathbf{\Lambda} & -\beta \mathbf{I} \\ \mathbf{I} & \mathbf{O} \end{bmatrix} \mathbf{\Pi}$$

$$\overset{(\text{C.11})}{=} \begin{bmatrix} \mathbf{e}_1^\top & \mathbf{0}^\top \\ \mathbf{0}^\top & \mathbf{e}_1^\top \\ \vdots & \\ \mathbf{e}_d^\top & \mathbf{0}^\top \\ \mathbf{0}^\top & \mathbf{e}_d^\top \end{bmatrix} \begin{bmatrix} (1+\beta)\mathbf{I} - \eta_t \mathbf{\Lambda} & -\beta \mathbf{I} \\ \mathbf{I} & \mathbf{O} \end{bmatrix} \begin{bmatrix} \mathbf{e}_1 & \mathbf{0} & \mathbf{e}_2 & \mathbf{0} & \dots & \mathbf{e}_d & \mathbf{0} \\ \mathbf{0} & \mathbf{e}_1 & \mathbf{0} & \mathbf{e}_2 & \dots & \mathbf{0} & \mathbf{e}_d \end{bmatrix}$$

$$\overset{(\text{C.13})}{=} \begin{bmatrix} \mathbf{e}_1^\top & \mathbf{0}^\top \\ \mathbf{0}^\top & \mathbf{e}_1^\top \\ \vdots & \\ \mathbf{e}_d^\top & \mathbf{0}^\top \\ \mathbf{0}^\top & \mathbf{e}_d^\top \end{bmatrix} \begin{bmatrix} (1+\beta - \eta_t \lambda_1)\mathbf{e}_1 & -\beta \mathbf{e}_1 & \dots & (1+\beta - \eta_t \lambda_d)\mathbf{e}_d & -\beta \mathbf{e}_d \\ \mathbf{e}_1 & \mathbf{0} & \dots & \mathbf{e}_d & \mathbf{0} \end{bmatrix}$$

$$= \begin{bmatrix} \mathbf{S}_{1,1} & \mathbf{S}_{1,2} & \dots & \mathbf{S}_{1,d} \\ \mathbf{S}_{2,1} & \mathbf{S}_{2,2} & \dots & \mathbf{S}_{2,d} \\ \vdots & \vdots & \ddots & \vdots \\ \mathbf{S}_{d,1} & \mathbf{S}_{d,2} & \dots & \mathbf{S}_{d,d} \end{bmatrix}, \text{ where } \mathbf{S}_{i,j} = \begin{bmatrix} (1+\beta - \eta_t \lambda_j)\mathbf{e}_i^\top \mathbf{e}_j & -\beta \mathbf{e}_i^\top \mathbf{e}_j \\ \mathbf{e}_i^\top \mathbf{e}_j & 0 \end{bmatrix}$$

$$\overset{(\text{C.10})}{=} \begin{bmatrix} \mathbf{T}_{t,1} & & & \\ & \mathbf{T}_{t,2} & & \\ & & \ddots & \\ & & & \mathbf{T}_{t,d} \end{bmatrix}$$

Since

$$\mathbf{\Pi}^\top \mathbf{\Pi} \overset{(C.11)}{=} \begin{bmatrix} \mathbf{e}_1^\top & \mathbf{0}^\top \\ \mathbf{0}^\top & \mathbf{e}_1^\top \\ \vdots & \\ \mathbf{e}_d^\top & \mathbf{0}^\top \\ \mathbf{0}^\top & \mathbf{e}_d^\top \end{bmatrix} \begin{bmatrix} \mathbf{e}_1 & \mathbf{0} & \mathbf{e}_2 & \mathbf{0} & \dots & \mathbf{e}_d & \mathbf{0} \\ \mathbf{0} & \mathbf{e}_1 & \mathbf{0} & \mathbf{e}_2 & \dots & \mathbf{0} & \mathbf{e}_d \end{bmatrix} \overset{(C.13)}{=} \mathbf{I}_{2d \times 2d} \tag{C.14}$$

$$\mathbf{V}^\top \mathbf{V} \overset{(C.11)}{=} \begin{bmatrix} \mathbf{U}^\top & \mathbf{O} \\ \mathbf{O} & \mathbf{U}^\top \end{bmatrix} \begin{bmatrix} \mathbf{U} & \mathbf{O} \\ \mathbf{O} & \mathbf{U} \end{bmatrix} \overset{(C.12)}{=} \mathbf{I}_{2d \times 2d}$$

are both orthogonal matrices, we thereby have

$$\mathbf{M}_t = \mathbf{V}\mathbf{\Pi} \begin{bmatrix} \mathbf{T}_{t,1} & & & \\ & \mathbf{T}_{t,2} & & \\ & & \ddots & \\ & & & \mathbf{T}_{t,d} \end{bmatrix} \mathbf{\Pi}^\top \mathbf{V}^\top.$$

$\square$

**Lemma 4.** *(Bound Variance with $\left\| \mathbf{T}_{t,j}^k \right\|$) Assuming batch size $M = 1$, we have*

$$\sum_{\tau=0}^{T-1} \mathbb{E}\left[ \eta_\tau^2 \begin{bmatrix} \mathbf{n}_\tau \\ \mathbf{0} \end{bmatrix}^\top \mathbf{M}_{\tau+1}^\top \mathbf{M}_{\tau+2}^\top \dots \mathbf{M}_{T-1}^\top \begin{bmatrix} \mathbf{H} & \mathbf{O} \\ \mathbf{O} & \mathbf{O} \end{bmatrix} \mathbf{M}_{T-1} \mathbf{M}_{T-2} \dots \mathbf{M}_{\tau+1} \begin{bmatrix} \mathbf{n}_\tau \\ \mathbf{0} \end{bmatrix} \right] \tag{C.15}$$

$$\leq \sigma^2 \sum_{j=1}^{d} \lambda_j^2 \sum_{\tau=0}^{T-1} \eta_\tau^2 \left\| \mathbf{T}_{T-1,j} \mathbf{T}_{T-2,j} \dots \mathbf{T}_{\tau+1,j} \right\|^2,$$

*where $\mathbf{M}_t$ is defined in Eqn. (C.3) and $\mathbf{T}_{t,j} \in \mathbb{R}^{2 \times 2}$ is defined in (C.10).*

*Proof.* Similarly, we define $\tilde{\mathbf{n}}$ and $\tilde{\mathbf{H}}$ as in Eqn. (C.4). Notice that $\tilde{\mathbf{H}}$ is a positive semi-definite matrix since if the eigenvalue decomposition of Hessian $\mathbf{H} = \mathbf{U}\mathbf{\Lambda}\mathbf{U}^\top$, we have

$$\tilde{\mathbf{H}} = \begin{bmatrix} \mathbf{H} & \mathbf{O} \\ \mathbf{O} & \mathbf{O} \end{bmatrix} = \begin{bmatrix} \mathbf{U} & \mathbf{O} \\ \mathbf{O} & \mathbf{U} \end{bmatrix} \begin{bmatrix} \mathbf{\Lambda} & \mathbf{O} \\ \mathbf{O} & \mathbf{O} \end{bmatrix} \begin{bmatrix} \mathbf{U} & \mathbf{O} \\ \mathbf{O} & \mathbf{U} \end{bmatrix}^\top.$$

Therefore, $\tilde{\mathbf{H}}^{1/2}$ is well-defined. Denote

$$\mathbf{A}_\tau \triangleq \tilde{\mathbf{H}}^{1/2} \mathbf{M}_{T-1} \mathbf{M}_{T-2} \dots \mathbf{M}_{\tau+1}, \tag{C.16}$$

then

$$\mathbf{A}_\tau^\top = \mathbf{M}_{\tau+1}^\top \mathbf{M}_{\tau+2}^\top \dots \mathbf{M}_{T-1}^\top \left( \tilde{\mathbf{H}}^{1/2} \right)^\top = \mathbf{M}_{\tau+1}^\top \mathbf{M}_{\tau+2}^\top \dots \mathbf{M}_{T-1}^\top \tilde{\mathbf{H}}^{1/2}, \tag{C.17}$$

where the second equality is because $\tilde{\mathbf{H}}^{1/2}$ is symmetric.

It follows

$$\sum_{\tau=0}^{T-1} \mathbb{E}\left[ \eta_\tau^2 \begin{bmatrix} \mathbf{n}_\tau \\ \mathbf{0} \end{bmatrix}^\top \mathbf{M}_{\tau+1}^\top \mathbf{M}_{\tau+2}^\top \dots \mathbf{M}_{T-1}^\top \begin{bmatrix} \mathbf{H} & \mathbf{O} \\ \mathbf{O} & \mathbf{O} \end{bmatrix} \mathbf{M}_{T-1} \mathbf{M}_{T-2} \dots \mathbf{M}_{\tau+1} \begin{bmatrix} \mathbf{n}_\tau \\ \mathbf{0} \end{bmatrix} \right]$$

$$\overset{(C.4)}{=} \sum_{\tau=0}^{T-1} \mathbb{E}\left[ \eta_\tau^2 \tilde{\mathbf{n}}_\tau^\top \mathbf{M}_{\tau+1}^\top \mathbf{M}_{\tau+2}^\top \dots \mathbf{M}_{T-1}^\top \tilde{\mathbf{H}} \mathbf{M}_{T-1} \mathbf{M}_{T-2} \dots \mathbf{M}_{\tau+1} \tilde{\mathbf{n}}_\tau \right]$$

$$\overset{(C.16)(C.17)}{=} \sum_{\tau=0}^{T-1} \mathbb{E}\left[ \eta_\tau^2 \tilde{\mathbf{n}}_\tau^\top \mathbf{A}_\tau^\top \mathbf{A}_\tau \tilde{\mathbf{n}}_\tau \right] = \sum_{\tau=0}^{T-1} \eta_\tau^2 \mathbb{E}\left[ \text{tr}\left( \tilde{\mathbf{n}}_\tau^\top \mathbf{A}_\tau^\top \mathbf{A}_\tau \tilde{\mathbf{n}}_\tau \right) \right] \qquad \triangleright \text{ Trace of a scalar is itself}$$

$$= \sum_{\tau=0}^{T-1} \eta_\tau^2 \mathbb{E}\left[ \text{tr}\left( \mathbf{A}_\tau \tilde{\mathbf{n}}_\tau \tilde{\mathbf{n}}_\tau^\top \mathbf{A}_\tau^\top \right) \right] \qquad \triangleright \text{ Cyclic property of trace}$$

$$= \sum_{\tau=0}^{T-1} \eta_\tau^2 \text{tr}\left( \mathbb{E}\left[ \mathbf{A}_\tau \tilde{\mathbf{n}}_\tau \tilde{\mathbf{n}}_\tau^\top \mathbf{A}_\tau^\top \right] \right) \qquad \triangleright \text{ Linearity of expectation}$$

$$\overset{(D.1)}{=} \sum_{\tau=0}^{T-1} \eta_\tau^2 \mathrm{tr}\left(\mathbf{A}_\tau \mathbb{E}\left[\tilde{\mathbf{n}}_\tau \tilde{\mathbf{n}}_\tau^\top\right]\mathbf{A}_\tau^\top\right) \overset{(C.4)}{=} \sum_{\tau=0}^{T-1}\eta_\tau^2 \mathrm{tr}\left(\mathbf{A}_\tau \mathbb{E}\begin{bmatrix}\mathbf{n}_\tau\mathbf{n}_\tau^\top & \mathbf{O}\\ \mathbf{O} & \mathbf{O}\end{bmatrix}\mathbf{A}_\tau^\top\right)$$

$$\leq \sum_{\tau=0}^{T-1}\eta_\tau^2 \mathrm{tr}\left(\mathbf{A}_\tau\begin{bmatrix}\sigma^2\mathbf{H} & \mathbf{O}\\ \mathbf{O} & \mathbf{O}\end{bmatrix}\mathbf{A}_\tau^\top\right) = \sigma^2 \sum_{\tau=0}^{T-1}\eta_\tau^2 \mathrm{tr}\left(\mathbf{A}_\tau\begin{bmatrix}\mathbf{H} & \mathbf{O}\\ \mathbf{O} & \mathbf{O}\end{bmatrix}\mathbf{A}_\tau^\top\right)$$

$$=\sigma^2\sum_{\tau=0}^{T-1}\eta_\tau^2 \mathrm{tr}\left(\mathbf{A}_\tau^\top\mathbf{A}_\tau\begin{bmatrix}\mathbf{H} & \mathbf{O}\\ \mathbf{O} & \mathbf{O}\end{bmatrix}\right) \qquad \triangleright \text{ Cyclic property of trace}$$

$$\overset{(C.16)}{=}\overset{(C.17)}{\sigma^2}\sum_{\tau=0}^{T-1}\eta_\tau^2 \mathrm{tr}\left(\mathbf{M}_{\tau+1}^\top\mathbf{M}_{\tau+2}^\top\ldots\mathbf{M}_{T-1}^\top\tilde{\mathbf{H}}\mathbf{M}_{T-1}\mathbf{M}_{T-2}\ldots\mathbf{M}_{\tau+1}\begin{bmatrix}\mathbf{H} & \mathbf{O}\\ \mathbf{O} & \mathbf{O}\end{bmatrix}\right)$$

$$\overset{(C.4)}{=}\sigma^2\sum_{\tau=0}^{T-1}\eta_\tau^2 \mathrm{tr}\left(\mathbf{M}_{\tau+1}^\top\mathbf{M}_{\tau+2}^\top\ldots\mathbf{M}_{T-1}^\top\tilde{\mathbf{H}}\mathbf{M}_{T-1}\mathbf{M}_{T-2}\ldots\mathbf{M}_{\tau+1}\tilde{\mathbf{H}}\right)$$

where the inequality is because

$$\mathbb{E}\begin{bmatrix}\mathbf{n}_\tau\mathbf{n}_\tau^\top & \mathbf{O}\\ \mathbf{O} & \mathbf{O}\end{bmatrix} \preceq \begin{bmatrix}\sigma^2\mathbf{H} & \mathbf{O}\\ \mathbf{O} & \mathbf{O}\end{bmatrix}$$

given $\mathbb{E}[\mathbf{n}_\tau\mathbf{n}_\tau^\top] \preceq \sigma^2\mathbf{H}$ in Assumption 3, along with basic properties of Loewner order in Lemma 13, 14 and 15.

Let

$$\tilde{\mathbf{T}}_t \triangleq \begin{bmatrix}\mathbf{T}_{t,1} & & & \\ & \mathbf{T}_{t,2} & & \\ & & \ddots & \\ & & & \mathbf{T}_{t,d}\end{bmatrix}, \tag{C.18}$$

we have the variance term being

$$\sum_{\tau=0}^{T-1}\mathbb{E}\left[\eta_\tau^2\begin{bmatrix}\mathbf{n}_\tau\\ \mathbf{0}\end{bmatrix}^\top\mathbf{M}_{\tau+1}^\top\mathbf{M}_{\tau+2}^\top\ldots\mathbf{M}_{T-1}^\top\begin{bmatrix}\mathbf{H} & \mathbf{O}\\ \mathbf{O} & \mathbf{O}\end{bmatrix}\mathbf{M}_{T-1}\mathbf{M}_{T-2}\ldots\mathbf{M}_{\tau+1}\begin{bmatrix}\mathbf{n}_\tau\\ \mathbf{0}\end{bmatrix}\right]$$

$$\leq\sigma^2\sum_{\tau=0}^{T-1}\eta_\tau^2 \mathrm{tr}\left(\mathbf{M}_{\tau+1}^\top\mathbf{M}_{\tau+2}^\top\ldots\mathbf{M}_{T-1}^\top\tilde{\mathbf{H}}\mathbf{M}_{T-1}\mathbf{M}_{T-2}\ldots\mathbf{M}_{\tau+1}\tilde{\mathbf{H}}\right)$$

$$\overset{(C.9)}{=}\sigma^2\sum_{\tau=0}^{T-1}\eta_\tau^2 \mathrm{tr}\left(\mathbf{V}\mathbf{\Pi}\tilde{\mathbf{T}}_{\tau+1}^\top\tilde{\mathbf{T}}_{\tau+2}^\top\ldots\tilde{\mathbf{T}}_{T-1}^\top\mathbf{\Pi}^\top\mathbf{V}^\top\tilde{\mathbf{H}}\mathbf{V}\mathbf{\Pi}\tilde{\mathbf{T}}_{T-1}\tilde{\mathbf{T}}_{T-2}\ldots\tilde{\mathbf{T}}_{\tau+1}\mathbf{\Pi}^\top\mathbf{V}^\top\tilde{\mathbf{H}}\right)$$

$$=\sigma^2\sum_{\tau=0}^{T-1}\eta_\tau^2 \mathrm{tr}\left(\tilde{\mathbf{T}}_{\tau+1}^\top\tilde{\mathbf{T}}_{\tau+2}^\top\ldots\tilde{\mathbf{T}}_{T-1}^\top\left(\mathbf{\Pi}^\top\mathbf{V}^\top\tilde{\mathbf{H}}\mathbf{V}\mathbf{\Pi}\right)\tilde{\mathbf{T}}_{T-1}\tilde{\mathbf{T}}_{T-2}\ldots\tilde{\mathbf{T}}_{\tau+1}\left(\mathbf{\Pi}^\top\mathbf{V}^\top\tilde{\mathbf{H}}\mathbf{V}\mathbf{\Pi}\right)\right).$$

Here the last equality comes from the cyclic property of trace. Given the definition of $\mathbf{\Pi}$, $\mathbf{V}$ and $\tilde{\mathbf{H}}$ in Eqn. (C.11) and Eqn. (C.4), we have

$$
\mathbf{\Pi}^\top \mathbf{V}^\top \tilde{\mathbf{H}} \mathbf{V} \mathbf{\Pi} = \begin{bmatrix} \mathbf{e}_1^\top & \mathbf{0}^\top \\ \mathbf{0}^\top & \mathbf{e}_1^\top \\ & \vdots \\ \mathbf{e}_d^\top & \mathbf{0}^\top \\ \mathbf{0}^\top & \mathbf{e}_d^\top \end{bmatrix} \begin{bmatrix} \mathbf{U}^\top & \mathbf{O} \\ \mathbf{O} & \mathbf{U}^\top \end{bmatrix} \begin{bmatrix} \mathbf{H} & \mathbf{O} \\ \mathbf{O} & \mathbf{O} \end{bmatrix} \begin{bmatrix} \mathbf{U} & \mathbf{O} \\ \mathbf{O} & \mathbf{U} \end{bmatrix} \begin{bmatrix} \mathbf{e}_1 & \mathbf{0} & \mathbf{e}_2 & \mathbf{0} & \ldots & \mathbf{e}_d & \mathbf{0} \\ \mathbf{0} & \mathbf{e}_1 & \mathbf{0} & \mathbf{e}_2 & \ldots & \mathbf{0} & \mathbf{e}_d \end{bmatrix}
$$

$$
\overset{(\text{C.12})}{=} \begin{bmatrix} \mathbf{e}_1^\top & \mathbf{0}^\top \\ \mathbf{0}^\top & \mathbf{e}_1^\top \\ & \vdots \\ \mathbf{e}_d^\top & \mathbf{0}^\top \\ \mathbf{0}^\top & \mathbf{e}_d^\top \end{bmatrix} \begin{bmatrix} \mathbf{\Lambda} & \mathbf{O} \\ \mathbf{O} & \mathbf{O} \end{bmatrix} \begin{bmatrix} \mathbf{e}_1 & \mathbf{0} & \mathbf{e}_2 & \mathbf{0} & \ldots & \mathbf{e}_d & \mathbf{0} \\ \mathbf{0} & \mathbf{e}_1 & \mathbf{0} & \mathbf{e}_2 & \ldots & \mathbf{0} & \mathbf{e}_d \end{bmatrix}
$$

$$
= \begin{bmatrix} \lambda_1 & & & & & \\ & 0 & & & & \\ & & \lambda_2 & & & \\ & & & 0 & & \\ & & & & \ddots & \\ & & & & & \lambda_d \\ & & & & & & 0 \end{bmatrix} = \mathbf{\Lambda} \otimes \begin{bmatrix} 1 & 0 \\ 0 & 0 \end{bmatrix},
$$

(C.19)

Here $\otimes$ is the Kronecker product. Then the variance term is simplified to

$$
\sum_{\tau=0}^{T-1} \mathbb{E} \left[ \eta_\tau^2 \begin{bmatrix} \mathbf{n}_\tau \\ \mathbf{0} \end{bmatrix}^\top \mathbf{M}_{\tau+1}^\top \mathbf{M}_{\tau+2}^\top \ldots \mathbf{M}_{T-1}^\top \begin{bmatrix} \mathbf{H} & \mathbf{O} \\ \mathbf{O} & \mathbf{O} \end{bmatrix} \mathbf{M}_{T-1} \mathbf{M}_{T-2} \ldots \mathbf{M}_{\tau+1} \begin{bmatrix} \mathbf{n}_\tau \\ \mathbf{0} \end{bmatrix} \right]
$$

$$
\leq \sigma^2 \sum_{\tau=0}^{T-1} \eta_\tau^2 \mathrm{tr} \left( \tilde{\mathbf{T}}_{\tau+1}^\top \tilde{\mathbf{T}}_{\tau+2}^\top \ldots \tilde{\mathbf{T}}_{T-1}^\top \left( \mathbf{\Lambda} \otimes \begin{bmatrix} 1 & 0 \\ 0 & 0 \end{bmatrix} \right) \tilde{\mathbf{T}}_{T-1} \tilde{\mathbf{T}}_{T-2} \ldots \tilde{\mathbf{T}}_{\tau+1} \left( \mathbf{\Lambda} \otimes \begin{bmatrix} 1 & 0 \\ 0 & 0 \end{bmatrix} \right) \right)
$$

$$
\overset{(\text{C.18})}{=} \sigma^2 \sum_{\tau=0}^{T-1} \eta_\tau^2 \sum_{j=1}^d \mathrm{tr} \left( \mathbf{T}_{\tau+1,j}^\top \mathbf{T}_{\tau+2,j}^\top \ldots \mathbf{T}_{T-1,j}^\top \begin{bmatrix} \lambda_j & 0 \\ 0 & 0 \end{bmatrix} \mathbf{T}_{T-1,j} \mathbf{T}_{T-2,j} \ldots \mathbf{T}_{\tau+1,j} \begin{bmatrix} \lambda_j & 0 \\ 0 & 0 \end{bmatrix} \right)
$$

$\triangleright$ All are block diagonal matrices

$$
= \sigma^2 \sum_{\tau=0}^{T-1} \eta_\tau^2 \sum_{j=1}^d \lambda_j^2 \mathrm{tr} \left( \mathbf{T}_{\tau+1,j}^\top \mathbf{T}_{\tau+2,j}^\top \ldots \mathbf{T}_{T-1,j}^\top \begin{bmatrix} 1 \\ 0 \end{bmatrix} \begin{bmatrix} 1 \\ 0 \end{bmatrix}^\top \mathbf{T}_{T-1,j} \mathbf{T}_{T-2,j} \ldots \mathbf{T}_{\tau+1,j} \begin{bmatrix} 1 \\ 0 \end{bmatrix} \begin{bmatrix} 1 \\ 0 \end{bmatrix}^\top \right)
$$

$$
= \sigma^2 \sum_{\tau=0}^{T-1} \eta_\tau^2 \sum_{j=1}^d \lambda_j^2 \mathrm{tr} \left( \begin{bmatrix} 1 \\ 0 \end{bmatrix}^\top \mathbf{T}_{\tau+1,j}^\top \mathbf{T}_{\tau+2,j}^\top \ldots \mathbf{T}_{T-1,j}^\top \begin{bmatrix} 1 \\ 0 \end{bmatrix} \begin{bmatrix} 1 \\ 0 \end{bmatrix}^\top \mathbf{T}_{T-1,j} \mathbf{T}_{T-2,j} \ldots \mathbf{T}_{\tau+1,j} \begin{bmatrix} 1 \\ 0 \end{bmatrix} \right)
$$

$\triangleright$ Cyclic property of trace

$$
= \sigma^2 \sum_{\tau=0}^{T-1} \eta_\tau^2 \sum_{j=1}^d \lambda_j^2 \left( \begin{bmatrix} 1 \\ 0 \end{bmatrix}^\top \mathbf{T}_{\tau+1,j}^\top \mathbf{T}_{\tau+2,j}^\top \ldots \mathbf{T}_{T-1,j}^\top \begin{bmatrix} 1 \\ 0 \end{bmatrix} \right) \left( \begin{bmatrix} 1 \\ 0 \end{bmatrix}^\top \mathbf{T}_{T-1,j} \mathbf{T}_{T-2,j} \ldots \mathbf{T}_{\tau+1,j} \begin{bmatrix} 1 \\ 0 \end{bmatrix} \right)
$$

$\triangleright$ Notice that the term inside the trace is a scalar

$$
= \sigma^2 \sum_{\tau=0}^{T-1} \eta_\tau^2 \sum_{j=1}^d \lambda_j^2 \left( \begin{bmatrix} 1 \\ 0 \end{bmatrix}^\top \mathbf{T}_{T-1,j} \mathbf{T}_{T-2,j} \ldots \mathbf{T}_{\tau+1,j} \begin{bmatrix} 1 \\ 0 \end{bmatrix} \right)^2
$$

$\triangleright$ Transpose of a scalar is itself

$$
\leq \sigma^2 \sum_{\tau=0}^{T-1} \eta_\tau^2 \sum_{j=1}^d \lambda_j^2 \left\| \mathbf{T}_{T-1,j} \mathbf{T}_{T-2,j} \ldots \mathbf{T}_{\tau+1,j} \right\|^2
$$

$$= \sigma^2 \sum_{j=1}^{d} \lambda_j^2 \sum_{\tau=0}^{T-1} \eta_\tau^2 \left\| \mathbf{T}_{T-1,j} \mathbf{T}_{T-2,j} \ldots \mathbf{T}_{\tau+1,j} \right\|^2.$$

Here the last inequality is entailed by the fact that for $\forall \mathbf{C} \in \mathbb{R}^{2\times 2}$,

$$\begin{bmatrix} 1 \\ 0 \end{bmatrix}^\top \mathbf{C} \begin{bmatrix} 1 \\ 0 \end{bmatrix} = \left( \mathbf{C} \begin{bmatrix} 1 \\ 0 \end{bmatrix} \right)_1 \leq \sqrt{\left( \mathbf{C} \begin{bmatrix} 1 \\ 0 \end{bmatrix} \right)_1^2 + \left( \mathbf{C} \begin{bmatrix} 1 \\ 0 \end{bmatrix} \right)_2^2} = \left\| \mathbf{C} \begin{bmatrix} 1 \\ 0 \end{bmatrix} \right\| \leq \|\mathbf{C}\|,$$

with $(\mathbf{x})_1, (\mathbf{x})_2$ standing for the first and second element of vector $\mathbf{x}$. $\qquad\square$

**Lemma 5.** *(Bound Bias with $\left\| \mathbf{T}_{t,j}^k \right\|$)*

$$\mathbb{E}\left[ \begin{bmatrix} \mathbf{w}_0 - \mathbf{w}_* \\ \mathbf{w}_{-1} - \mathbf{w}_* \end{bmatrix}^\top \mathbf{M}_0^\top \mathbf{M}_1^\top \ldots \mathbf{M}_{T-1}^\top \begin{bmatrix} \mathbf{H} & \mathbf{O} \\ \mathbf{O} & \mathbf{O} \end{bmatrix} \mathbf{M}_{T-1} \mathbf{M}_{T-2} \ldots \mathbf{M}_0 \begin{bmatrix} \mathbf{w}_0 - \mathbf{w}_* \\ \mathbf{w}_{-1} - \mathbf{w}_* \end{bmatrix} \right]$$
$$\leq \sum_{j=1}^{d} \lambda_j \left\| \mathbf{T}_{T-1,j} \mathbf{T}_{T-2,j} \ldots \mathbf{T}_{0,j} \right\|^2 \mathbb{E} \left\| \left( \mathbf{\Pi}^\top \mathbf{V}^\top \begin{bmatrix} \mathbf{w}_0 - \mathbf{w}_* \\ \mathbf{w}_{-1} - \mathbf{w}_* \end{bmatrix} \right)_{2j-1:2j} \right\|^2, \tag{C.20}$$

*where $\mathbf{M}_t$ is defined in Eqn. (C.3), $\mathbf{T}_{t,j} \in \mathbb{R}^{2\times 2}$ is defined in (C.10) and $\mathbf{\Pi}, \mathbf{V}$ are orthogonal matrices defined in (C.11). Here notation $\mathbf{z}_{j_1:j_2}$ means*

$$\text{For } \forall \mathbf{z} = \begin{bmatrix} z_1 \\ z_2 \\ \vdots \\ z_{d'} \end{bmatrix} \in \mathbb{R}^{d'}, 1 \leq j \leq j' \leq d', \qquad \mathbf{z}_{j:j'} \triangleq \begin{bmatrix} z_j \\ z_{j+1} \\ \vdots \\ z_{j'} \end{bmatrix} \tag{C.21}$$

*Proof.* The proof is similar to a simplified version of the variance case, so we will reuse some of its notations to shorten the proof.

$$\mathbb{E}\left[ \begin{bmatrix} \mathbf{w}_0 - \mathbf{w}_* \\ \mathbf{w}_{-1} - \mathbf{w}_* \end{bmatrix}^\top \mathbf{M}_0^\top \mathbf{M}_1^\top \ldots \mathbf{M}_{T-1}^\top \begin{bmatrix} \mathbf{H} & \mathbf{O} \\ \mathbf{O} & \mathbf{O} \end{bmatrix} \mathbf{M}_{T-1} \mathbf{M}_{T-2} \ldots \mathbf{M}_0 \begin{bmatrix} \mathbf{w}_0 - \mathbf{w}_* \\ \mathbf{w}_{-1} - \mathbf{w}_* \end{bmatrix} \right]$$
$$\overset{(\text{C.4})}{=} \mathbb{E}\left[ \tilde{\mathbf{w}}_0^\top \mathbf{M}_1^\top \mathbf{M}_2^\top \ldots \mathbf{M}_{T-1}^\top \tilde{\mathbf{H}} \mathbf{M}_{T-1} \mathbf{M}_{T-2} \ldots \mathbf{M}_1 \tilde{\mathbf{w}}_0 \right]$$
$$\overset{(\text{C.9})\,(\text{C.18})}{=} \mathbb{E}\left[ \tilde{\mathbf{w}}_0^\top \mathbf{V}\mathbf{\Pi} \tilde{\mathbf{T}}_0^\top \tilde{\mathbf{T}}_1^\top \ldots \tilde{\mathbf{T}}_{T-1}^\top \mathbf{\Pi}^\top \mathbf{V}^\top \tilde{\mathbf{H}} \mathbf{V}\mathbf{\Pi} \tilde{\mathbf{T}}_{T-1} \tilde{\mathbf{T}}_{T-2} \ldots \tilde{\mathbf{T}}_0 \mathbf{\Pi}^\top \mathbf{V}^\top \tilde{\mathbf{w}}_0 \right]$$
$$= \mathbb{E}\left[ \left( \tilde{\mathbf{w}}_0^\top \mathbf{V}\mathbf{\Pi} \right) \tilde{\mathbf{T}}_0^\top \tilde{\mathbf{T}}_1^\top \ldots \tilde{\mathbf{T}}_{T-1}^\top \left( \mathbf{\Pi}^\top \mathbf{V}^\top \tilde{\mathbf{H}} \mathbf{V}\mathbf{\Pi} \right) \tilde{\mathbf{T}}_{T-1} \tilde{\mathbf{T}}_{T-2} \ldots \tilde{\mathbf{T}}_0 \left( \mathbf{\Pi}^\top \mathbf{V}^\top \tilde{\mathbf{w}}_0 \right) \right]$$
$$\overset{(\text{C.19})}{=} \mathbb{E}\left[ \left( \tilde{\mathbf{w}}_0^\top \mathbf{V}\mathbf{\Pi} \right) \tilde{\mathbf{T}}_0^\top \tilde{\mathbf{T}}_1^\top \ldots \tilde{\mathbf{T}}_{T-1}^\top \left( \mathbf{\Lambda} \otimes \begin{bmatrix} 1 & 0 \\ 0 & 0 \end{bmatrix} \right) \tilde{\mathbf{T}}_{T-1} \tilde{\mathbf{T}}_{T-2} \ldots \tilde{\mathbf{T}}_0 \left( \mathbf{\Pi}^\top \mathbf{V}^\top \tilde{\mathbf{w}}_0 \right) \right]$$
$$\overset{(\text{C.18})\,(\text{C.21})}{=} \mathbb{E}\left[ \begin{bmatrix} \left( \mathbf{\Pi}^\top \mathbf{V}^\top \tilde{\mathbf{w}}_0 \right)_{1:2} \\ \left( \mathbf{\Pi}^\top \mathbf{V}^\top \tilde{\mathbf{w}}_0 \right)_{3:4} \\ \vdots \\ \left( \mathbf{\Pi}^\top \mathbf{V}^\top \tilde{\mathbf{w}}_0 \right)_{2d-1:2d} \end{bmatrix}^\top \begin{bmatrix} \mathbf{S}_1 & & & \\ & \mathbf{S}_2 & & \\ & & \ddots & \\ & & & \mathbf{S}_d \end{bmatrix} \begin{bmatrix} \left( \mathbf{\Pi}^\top \mathbf{V}^\top \tilde{\mathbf{w}}_0 \right)_{1:2} \\ \left( \mathbf{\Pi}^\top \mathbf{V}^\top \tilde{\mathbf{w}}_0 \right)_{3:4} \\ \vdots \\ \left( \mathbf{\Pi}^\top \mathbf{V}^\top \tilde{\mathbf{w}}_0 \right)_{2d-1:2d} \end{bmatrix} \right],$$

$$\text{where } \mathbf{S}_j = \mathbf{T}_{0,j}^\top \mathbf{T}_{1,j}^\top \ldots \mathbf{T}_{T-1,j}^\top \begin{bmatrix} \lambda_j & 0 \\ 0 & 0 \end{bmatrix} \mathbf{T}_{T-1,j} \mathbf{T}_{T-2,j} \ldots \mathbf{T}_{0,j}$$

$$= \mathbb{E}\left[ \sum_{j=1}^{d} \left( \mathbf{\Pi}^\top \mathbf{V}^\top \tilde{\mathbf{w}}_0 \right)_{2j-1:2j}^\top \mathbf{S}_j \left( \mathbf{\Pi}^\top \mathbf{V}^\top \tilde{\mathbf{w}}_0 \right)_{2j-1:2j} \right],$$

$$\text{where } \mathbf{S}_j = \mathbf{T}_{0,j}^\top \mathbf{T}_{1,j}^\top \ldots \mathbf{T}_{T-1,j}^\top \begin{bmatrix} \lambda_j & 0 \\ 0 & 0 \end{bmatrix} \mathbf{T}_{T-1,j} \mathbf{T}_{T-2,j} \ldots \mathbf{T}_{0,j}$$

$$= \mathbb{E}\left[ \sum_{j=1}^{d} \left( \mathbf{\Pi}^\top \mathbf{V}^\top \tilde{\mathbf{w}}_0 \right)_{2j-1:2j}^\top \left( \lambda_j \mathbf{s}_j \mathbf{s}_j^\top \right) \left( \mathbf{\Pi}^\top \mathbf{V}^\top \tilde{\mathbf{w}}_0 \right)_{2j-1:2j} \right],$$

$$\text{where } \mathbf{s}_j = \mathbf{T}_{0,j}^{\top}\mathbf{T}_{1,j}^{\top}\ldots\mathbf{T}_{T-1,j}^{\top}\begin{bmatrix}1\\0\end{bmatrix}$$

$$=\mathbb{E}\left[\sum_{j=1}^{d}\lambda_j\left(\left(\mathbf{\Pi}^{\top}\mathbf{V}^{\top}\tilde{\mathbf{w}}_0\right)_{2j-1:2j}^{\top}\mathbf{s}_j\right)\left(\mathbf{s}_j^{\top}\left(\mathbf{\Pi}^{\top}\mathbf{V}^{\top}\tilde{\mathbf{w}}_0\right)_{2j-1:2j}\right)\right],$$

$$\text{where } \mathbf{s}_j = \mathbf{T}_{0,j}^{\top}\mathbf{T}_{1,j}^{\top}\ldots\mathbf{T}_{T-1,j}^{\top}\begin{bmatrix}1\\0\end{bmatrix}$$

$$=\mathbb{E}\left[\sum_{j=1}^{d}\lambda_j\left(\mathbf{s}_j^{\top}\left(\mathbf{\Pi}^{\top}\mathbf{V}^{\top}\tilde{\mathbf{w}}_0\right)_{2j-1:2j}\right)^2\right], \text{ where } \mathbf{s}_j = \mathbf{T}_{0,j}^{\top}\mathbf{T}_{1,j}^{\top}\ldots\mathbf{T}_{T-1,j}^{\top}\begin{bmatrix}1\\0\end{bmatrix}$$

$$=\mathbb{E}\left[\sum_{j=1}^{d}\lambda_j\left(\begin{bmatrix}1\\0\end{bmatrix}^{\top}\mathbf{T}_{T-1,j}\mathbf{T}_{T-2,j}\ldots\mathbf{T}_{0,j}\left(\mathbf{\Pi}^{\top}\mathbf{V}^{\top}\tilde{\mathbf{w}}_0\right)_{2j-1:2j}\right)^2\right]$$

$$\leq\mathbb{E}\left[\sum_{j=1}^{d}\lambda_j\left\|\mathbf{T}_{T-1,j}\mathbf{T}_{T-2,j}\ldots\mathbf{T}_{0,j}\right\|^2\left\|\left(\mathbf{\Pi}^{\top}\mathbf{V}^{\top}\tilde{\mathbf{w}}_0\right)_{2j-1:2j}\right\|^2\right]$$

$$=\sum_{j=1}^{d}\lambda_j\left\|\mathbf{T}_{T-1,j}\mathbf{T}_{T-2,j}\ldots\mathbf{T}_{0,j}\right\|^2\mathbb{E}\left\|\left(\mathbf{\Pi}^{\top}\mathbf{V}^{\top}\tilde{\mathbf{w}}_0\right)_{2j-1:2j}\right\|^2$$

$\triangleright$ Linearity of expectation

$$\stackrel{(C.4)}{=}\sum_{j=1}^{d}\lambda_j\left\|\mathbf{T}_{T-1,j}\mathbf{T}_{T-2,j}\ldots\mathbf{T}_{0,j}\right\|^2\mathbb{E}\left\|\left(\mathbf{\Pi}^{\top}\mathbf{V}^{\top}\begin{bmatrix}\mathbf{w}_0-\mathbf{w}_*\\\mathbf{w}_{-1}-\mathbf{w}_*\end{bmatrix}\right)_{2j-1:2j}\right\|^2.$$

Here the inequality is entailed by the fact that for $\forall \mathbf{C}\in\mathbb{R}^{2\times2}, \mathbf{z}\in\mathbb{R}^2$,

$$\begin{bmatrix}1\\0\end{bmatrix}^{\top}\mathbf{C}\mathbf{z} = (\mathbf{C}\mathbf{z})_1 \leq \sqrt{(\mathbf{C}\mathbf{z})_1^2+(\mathbf{C}\mathbf{z})_2^2} = \|\mathbf{C}\mathbf{z}\| \leq \|\mathbf{C}\|\,\|\mathbf{z}\|$$

$\square$

## C.2 BOUNDING $\|\mathbf{T}_{t+k,j}...\mathbf{T}_{t+1,j}\|$ WITH $\rho(\mathbf{T}_{t+k,j})$

This section upper bounds the matrix product $\|\mathbf{T}_{t+k,j}...\mathbf{T}_{t+1,j}\|$ with the spectral radius $\rho(\mathbf{T}_{t+1,j})$. Similar results for bounding $\left\|\mathbf{T}_{t,j}^k\right\|$ have been shown in (Wang et al., 2021)(Theorem 5), but our result is more general, so we still put our proofs here. In the following proof, we use $\|\cdot\|_F$ to denote the Frobenius norm of matrices.

**Lemma 6** (Bounding $\left\|\mathbf{T}_{t,j}^k\right\|_F$ with $\rho(\mathbf{T}_{t,j})^k$). *Given momentum matrices $\mathbf{T}_{t,j}$ that are defined in Eqn. (C.10) and $\beta \geq 1/4$, for all positive integer $k \geq 1$, it holds that*

$$\left\|\mathbf{T}_{t,j}^k\right\|_F \leq \min\left(8k, \frac{8}{\sqrt{|(1+\beta-\eta_t\lambda_j)^2-4\beta|}}\right)\rho(\mathbf{T}_{t,j})^k. \tag{C.22}$$

*Proof.* According to the definition of $\mathbf{T}_{t,j}$ in Eqn. (C.10),

$$\mathbf{T}_{t,j} = \begin{bmatrix}1+\beta-\eta_t\lambda_j & -\beta\\1 & 0\end{bmatrix}.$$

We can directly analyze the product by Jordan decomposition that there exists $\mathbf{P}\in\mathbb{C}^{2\times2}$ such that

$$\mathbf{T}_{t,j} = \mathbf{P}\mathbf{J}\mathbf{P}^{-1}$$

where $\mathbf{J}$ can have the following two cases

$$\mathbf{J} = \begin{bmatrix}\gamma_1 & 0\\0 & \gamma_2\end{bmatrix} \text{ or } \mathbf{J} = \begin{bmatrix}\gamma_1 & 1\\0 & \gamma_2\end{bmatrix}$$

depending on whether $\mathbf{T}_{t,j}$ is diagnolizable. Here $\gamma_1, \gamma_2$ are the eigenvalues of $\mathbf{T}_{t,j}$ and we assume without generality that $\gamma_1 \geq \gamma_2$ if $\gamma_1, \gamma_2 \in \mathbb{R}$. And when $\gamma_1, \gamma_2 \notin \mathbb{R}$, it holds that $\gamma_1$ and $\gamma_2$ are conjugate thus $|\gamma_1| = |\gamma_2|$. Therefore $|\gamma_1|$ is the spectral radius of $\mathbf{T}_{t,j}$, i.e. $\rho(\mathbf{T}_{t,j}) = |\gamma_1|$. Then we discuss case by case.

**i) If $\gamma_1 \neq \gamma_2$:** In this case one can verify that

$$\mathbf{P} = \begin{bmatrix} \gamma_1 & \gamma_2 \\ 1 & 1 \end{bmatrix}, \quad \mathbf{J} = \begin{bmatrix} \gamma_1 & 0 \\ 0 & \gamma_2 \end{bmatrix}, \quad \mathbf{P}^{-1} = \frac{1}{\gamma_1 - \gamma_2} \begin{bmatrix} 1 & -\gamma_2 \\ -1 & \gamma_1 \end{bmatrix}$$

where $\gamma_1, \gamma_2$ are eigenvalues of $\mathbf{T}_{t,j}$. And the characteristic polynomial of $\mathbf{T}_{t,j}$

$$\det\left(\mathbf{T}_{t,j} - \gamma\mathbf{I}\right) = \begin{vmatrix} 1 + \beta - \eta_t\lambda_j - \gamma & -\beta \\ 1 & -\gamma \end{vmatrix} = \gamma^2 - (1 + \beta - \eta_t\lambda_j)\gamma + \beta = 0$$

entails $\gamma_1 + \gamma_2 = 1 + \beta - \eta_t\lambda_j, \gamma_1\gamma_2 = \beta$. Thus in this case, it holds that

$$\mathbf{T}_{t,j}^k = \mathbf{P}\mathbf{J}^k\mathbf{P}^{-1} = \begin{bmatrix} \gamma_1 & \gamma_2 \\ 1 & 1 \end{bmatrix} \cdot \begin{bmatrix} \gamma_1^k & 0 \\ 0 & \gamma_2^k \end{bmatrix} \cdot \frac{1}{\gamma_1 - \gamma_2} \begin{bmatrix} 1 & -\gamma_2 \\ -1 & \gamma_1 \end{bmatrix}$$

$$= \begin{bmatrix} \frac{\gamma_1^{k+1} - \gamma_2^{k+1}}{\gamma_1 - \gamma_2} & -\beta\frac{\gamma_1^k - \gamma_2^k}{\gamma_1 - \gamma_2} \\ \frac{\gamma_1^k - \gamma_2^k}{\gamma_1 - \gamma_2} & -\beta\frac{\gamma_1^{k-1} - \gamma_2^{k-1}}{\gamma_1 - \gamma_2} \end{bmatrix}.$$

It holds that

$$\|\mathbf{T}_{t,j}\|_F = \frac{1}{|\gamma_1 - \gamma_2|} \left\| \begin{bmatrix} \gamma_1^{k+1} - \gamma_2^{k+1} & -\beta\left(\gamma_1^k - \gamma_2^k\right) \\ \gamma_1^k - \gamma_2^k & -\beta\left(\gamma_1^{k-1} - \gamma_2^{k-1}\right) \end{bmatrix} \right\|_F$$

$$\leq \frac{1}{|\gamma_1 - \gamma_2|} \left\| \begin{bmatrix} 2\rho(\mathbf{T}_{t,j})^{k+1} & -2\beta\rho(\mathbf{T}_{t,j})^k \\ 2\rho(\mathbf{T}_{t,j})^k & -2\beta\rho(\mathbf{T}_{t,j})^{k-1} \end{bmatrix} \right\|_F$$

$$\leq \frac{1}{|\gamma_1 - \gamma_2|} \left\| \begin{bmatrix} 2\rho(\mathbf{T}_{t,j})^k & -2\beta\rho(\mathbf{T}_{t,j})^k \\ 2\rho(\mathbf{T}_{t,j})^k & -4\beta\rho(\mathbf{T}_{t,j})^k \end{bmatrix} \right\|_F \leq \frac{2\rho(\mathbf{T}_{t,j})^k}{|\gamma_1 - \gamma_2|} \left\| \begin{bmatrix} 1 & -\beta \\ 1 & -2\beta \end{bmatrix} \right\|_F$$

$$\leq \frac{8\rho(\mathbf{T}_{t,j})^k}{|\gamma_1 - \gamma_2|} = \frac{8}{\sqrt{|(1 + \beta - \eta_t\lambda_j)^2 - 4\beta|}}\rho(\mathbf{T}_{t,j})^k,$$

where the first inequality holds as $|\gamma_1| = |\gamma_2| = \rho(\mathbf{T}_{t,j})$ and the third inequality holds as $\rho(\mathbf{T}_{t,j}) \geq \gamma_1\gamma_2 = \beta \geq 1/4$ according to Lemma 16 and thus $\rho(\mathbf{T}_{t,j}) \in (1/2, 1)$. For more details about the properties of $\rho(\mathbf{T}_{t,j})$ one can refer to Appendix C.3. We can also analyze the norm from another point: it holds that for all $k$,

$$\left| \frac{\gamma_1^k - \gamma_2^k}{\gamma_1 - \gamma_2} \right| = \left| \sum_{j=0}^{k-1} \gamma_1^j \gamma_2^{k-1-j} \right| \leq \sum_{j=0}^{k-1} \left| \gamma_1^j \gamma_2^{k-1-j} \right| = \sum_{j=0}^{k-1} \rho(\mathbf{T}_{t,j})^{k-1} = k\rho(\mathbf{T}_{t,j})^{k-1}.$$

Substituting we have

$$\|\mathbf{T}_{t,j}\|_F = \left\| \begin{bmatrix} \frac{\gamma_1^{k+1} - \gamma_2^{k+1}}{\gamma_1 - \gamma_2} & -\beta\frac{\gamma_1^k - \gamma_2^k}{\gamma_1 - \gamma_2} \\ \frac{\gamma_1^k - \gamma_2^k}{\gamma_1 - \gamma_2} & -\beta\frac{\gamma_1^{k-1} - \gamma_2^{k-1}}{\gamma_1 - \gamma_2} \end{bmatrix} \right\|_F$$

$$\leq \left\| \begin{bmatrix} (k+1)\rho(\mathbf{T}_{t,j})^k & -\beta k\rho(\mathbf{T}_{t,j})^{k-1} \\ k\rho(\mathbf{T}_{t,j})^{k-1} & -\beta(k-1)\rho(\mathbf{T}_{t,j})^{k-2} \end{bmatrix} \right\|_F$$

$$\leq \left\| \begin{bmatrix} 2k\rho(\mathbf{T}_{t,j})^k & -2k\rho(\mathbf{T}_{t,j})^k \\ 2k\rho(\mathbf{T}_{t,j})^k & -4k\rho(\mathbf{T}_{t,j})^k \end{bmatrix} \right\|_F \leq 8k\rho(\mathbf{T}_{t,j})^k.$$

Thus we prove the lemma when $\gamma_1 \neq \gamma_2$.

**ii) If $\gamma_1 = \gamma_2 = \gamma$:** In this case, $\mathbf{T}_{t,j}$ can not be diagonalized, which means that

$$\mathbf{T}_{t,j} = \begin{bmatrix} \gamma & 1 \\ 0 & \gamma \end{bmatrix}$$

one can verify that

$$\mathbf{P} = \begin{bmatrix} \gamma & 1 \\ 1 & 0 \end{bmatrix}, \quad \mathbf{J}^k = \begin{bmatrix} \gamma^k & k\gamma^{k-1} \\ 0 & \gamma^k \end{bmatrix}, \quad \mathbf{P}^{-1} = \begin{bmatrix} 0 & 1 \\ 1 & -\gamma \end{bmatrix}$$

where $\gamma_1 = \gamma_2 = \gamma \in \mathbb{R}$. In this case, it holds that

$$\begin{aligned} \mathbf{T}_{t,j}^k = \mathbf{P}\mathbf{J}^k\mathbf{P}^{-1} &= \begin{bmatrix} \gamma & 1 \\ 1 & 0 \end{bmatrix} \cdot \begin{bmatrix} \gamma^k & k\gamma^{k-1} \\ 0 & \gamma^k \end{bmatrix} \cdot \begin{bmatrix} 0 & 1 \\ 1 & -\gamma \end{bmatrix} \\ &= \begin{bmatrix} (k+1)\gamma^k & -k\gamma^{k+1} \\ k\gamma^{k-1} & -(k-1)\gamma^k \end{bmatrix} \end{aligned}$$

Then it holds that

$$\left\| \mathbf{T}_{t,j}^k \right\|_F = \left\| \begin{bmatrix} (k+1)\gamma^k & -k\gamma^{k+1} \\ k\gamma^{k-1} & -(k-1)\gamma^k \end{bmatrix} \right\|_F \leq \left\| \begin{bmatrix} 2k\gamma^k & -k\gamma^k \\ 2k\gamma^k & -k\gamma^k \end{bmatrix} \right\|_F \leq 8k\gamma^k = 8k\rho(\mathbf{T}_{t,j})^k.$$

Therefore, combining the two cases, we obtain the conclusion. $\square$

**Lemma 7** (Bounding $\|(\mathbf{T}_{t,j} + \boldsymbol{\Delta}_1)(\mathbf{T}_{t,j} + \boldsymbol{\Delta}_1)...(\mathbf{T}_{t,j} + \boldsymbol{\Delta}_k)\|_F$ with matrix power). *Given matrices $\mathbf{T}_{t,j}$ defined in Eqn. (C.10) and $\boldsymbol{\Delta}_i$, $\boldsymbol{\Delta}$ defined as*

$$\boldsymbol{\Delta}_i = \begin{bmatrix} \delta_i & 0 \\ 0 & 0 \end{bmatrix}, \qquad \boldsymbol{\Delta} = \begin{bmatrix} \delta & 0 \\ 0 & 0 \end{bmatrix},$$

*where $\delta_i \geq 0$ and $\delta = \max_{1 \leq i \leq k} \delta_i$, if $(1 + \beta - \eta_t\lambda_j)^2 - 4\beta \geq 0$, it holds that*

$$\|(\mathbf{T}_{t,j} + \boldsymbol{\Delta}_1)(\mathbf{T}_{t,j} + \boldsymbol{\Delta}_2)...(\mathbf{T}_{t,j} + \boldsymbol{\Delta}_k)\|_F \leq \left\| (\mathbf{T}_{t,j} + \boldsymbol{\Delta})^k \right\|_F. \tag{C.23}$$

*Proof.* As we assume $(1 + \beta - \eta_t\lambda_j)^2 - 4\beta \geq 0$, the eigenvalues of $\mathbf{T}_{t,j}$ $\gamma_1, \gamma_2 \in \mathbb{R}$. Following the same method as the proof of Lemma 6 to apply Jordan decomposition to $\mathbf{T}_{t,j}$, the power of momentum matrix $\mathbf{T}_{t,j}$ can be written as

$$\mathbf{T}_{t,j}^k = \begin{bmatrix} \frac{\gamma_1^{k+1} - \gamma_2^{k+1}}{\gamma_1 - \gamma_2} & \frac{-\beta(\gamma_1^k - \gamma_2^k)}{\gamma_1 - \gamma_2} \\ \frac{\gamma_1^k - \gamma_2^k}{\gamma_1 - \gamma_2} & \frac{-\beta(\gamma_2^{k-1} - \gamma_1^{k-1})}{\gamma_1 - \gamma_2} \end{bmatrix}, \quad \text{if } \gamma_1 \neq \gamma_2$$

$$\mathbf{T}_{t,j}^k = \begin{bmatrix} (k+1)\gamma^k & -k\gamma^{k+1} \\ k\gamma^{k-1} & -(k-1)\gamma^k \end{bmatrix}, \quad \text{if } \gamma_1 = \gamma_2 = \gamma$$

We can observe that in this case, the first column of $\mathbf{T}_{t,j}^k$ is nonnegative and the second column is nonpositive as $\gamma_1, \gamma_2, \gamma \in \mathbb{R}$. For simplicity, in the following proof we use $\prod$ to denote a product from $i = 1$ to $i = k$ orderly from left to right, namely, $\prod_{i=1}^k \mathbf{T}_{t+i,k} = \mathbf{T}_{t+k,j}\mathbf{T}_{t+k-1,j}...\mathbf{T}_{t+1,j}$. We first consider the combination product form of $\prod_{i=1}^k (\mathbf{T}_{t,j} + \boldsymbol{\Delta}_i)$ that

$$\prod_{i=1}^k (\mathbf{T}_{t,j} + \boldsymbol{\Delta}_i) = \sum_{l_1+...+l_t+k_1+...+k_{t+1}=k} \mathbf{T}_{t,j}^{k_1}\boldsymbol{\Delta}_{11}...\boldsymbol{\Delta}_{1l_1}\mathbf{T}_{t,j}^{k_2}\boldsymbol{\Delta}_{21}...\boldsymbol{\Delta}_{2l_2}\mathbf{T}_{t,j}^{k_3}...\boldsymbol{\Delta}_{t1}...\boldsymbol{\Delta}_{tl_t}\mathbf{T}_{t,j}^{k_{t+1}},$$

which is similar to the binomial expansion but without the commutativity of $\mathbf{T}$ and $\boldsymbol{\Delta}_i$. Now we consider one arbitrary combination term $\mathbf{S}$ that

$$\mathbf{S} = \mathbf{T}_{t,j}^{k_1}\boldsymbol{\Delta}_{11}...\boldsymbol{\Delta}_{1l_1}\mathbf{T}_{t,j}^{k_2}\boldsymbol{\Delta}_{21}...\boldsymbol{\Delta}_{2l_2}\mathbf{T}_{t,j}^{k_3}...\boldsymbol{\Delta}_{t1}...\boldsymbol{\Delta}_{tl_t}\mathbf{T}_{t,j}^{k_{t+1}}.$$

We first prove by induction that $\mathbf{S}$ has the following properties:

1. the first column of $\mathbf{S}$ is nonnegative and the second column of $\mathbf{S}$ is nonpositive;

2. the absolute value of each entry of $\mathbf{S}$ is monotonically increasing with respect to $\delta_1, ..., \delta_k$.

We call $\mathbf{T}_{t,j}^{k_i}$ or $\boldsymbol{\Delta}_{ij}$ one multiple component of $\mathbf{S}$ in the following proof. And we denote $\mathbf{S}_p$ the product of the first $p$ multiple component of $\mathbf{S}$ in the following proof. The first multiple component of $\mathbf{S}$ can be $\mathbf{S}_1 = \mathbf{T}_{t,j}^{k_1}$ or $\mathbf{S}_1 = \boldsymbol{\Delta}_{11}$, which satisfies the two desired properties naturally. Then we assume that the product of the first $p$ multiple component $\mathbf{S}_p$ satisfies the two properties. We discuss $\mathbf{S}_{p+1}$ in cases that

1. if the $p$-th multiple component is $\mathbf{T}_{t,j}^i$, where $i$ represents an arbitrary integer, then the $(p+1)$-th multiple component should be $\boldsymbol{\Delta}_{i'}$, where $i'$ also represents an arbitrary integer, or the $p$-th and $(p+1)$-th component can be merged. Then it holds that

$$\mathbf{S}_{p+1} = \mathbf{S}_p \boldsymbol{\Delta}_{i'} = \begin{bmatrix} \mathbf{S}_{p,11} & \mathbf{S}_{p,12} \\ \mathbf{S}_{p,21} & \mathbf{S}_{p,22} \end{bmatrix} \begin{bmatrix} \delta_{i'} & 0 \\ 0 & 0 \end{bmatrix} = \begin{bmatrix} \delta_{i'} \mathbf{S}_{p,11} & 0 \\ \delta_{i'} \mathbf{S}_{p,21} & 0 \end{bmatrix}.$$

Thus if the two properties hold for $p$, it also holds for $p+1$ in this case.

2. if the $p$-th multiple component is $\boldsymbol{\Delta}_i$, where $i$ represents an arbitrary integer, and the $(p+1)$-th multiple component is $\boldsymbol{\Delta}_{i+1}$, then it holds that

$$\mathbf{S}_{p+1} = \mathbf{S}_p \boldsymbol{\Delta}_{i+1} = \begin{bmatrix} \mathbf{S}_{p,11} & \mathbf{S}_{p,12} \\ \mathbf{S}_{p,21} & \mathbf{S}_{p,22} \end{bmatrix} \begin{bmatrix} \delta_{i+1} & 0 \\ 0 & 0 \end{bmatrix} = \begin{bmatrix} \delta_{i+1} \mathbf{S}_{p,11} & 0 \\ \delta_{i+1} \mathbf{S}_{p,21} & 0 \end{bmatrix}.$$

Thus if the two properties hold for $p$, it also holds for $p+1$ in this case.

3. if the $p$-th multiple component is $\boldsymbol{\Delta}_i$, where $i$ represents an arbitrary integer, and the $(p+1)$-th multiple component is $\mathbf{T}_{t,j}^{i'}$, then it holds that

$$\mathbf{S}_p = \mathbf{S}_{p-1} \boldsymbol{\Delta}_i = \begin{bmatrix} \mathbf{S}_{p-1,11} & \mathbf{S}_{p-1,12} \\ \mathbf{S}_{p-1,21} & \mathbf{S}_{p-1,22} \end{bmatrix} \begin{bmatrix} \delta_i & 0 \\ 0 & 0 \end{bmatrix} = \begin{bmatrix} \delta_i \mathbf{S}_{p-1,11} & 0 \\ \delta_i \mathbf{S}_{p-1,21} & 0 \end{bmatrix},$$

which implies that $\mathbf{S}_{p,12} = \mathbf{S}_{p,22} = 0$, thus we can substitute that

$$\mathbf{S}_{p+1} = \mathbf{S}_p \mathbf{T}_{t,j}^{i'} = \begin{bmatrix} \mathbf{S}_{p,11} & 0 \\ \mathbf{S}_{p,21} & 0 \end{bmatrix} \begin{bmatrix} t_{11} & t_{12} \\ t_{21} & t_{22} \end{bmatrix} = \begin{bmatrix} \mathbf{S}_{p,11} t_{11} & \mathbf{S}_{p,11} t_{12} \\ \mathbf{S}_{p,21} t_{11} & \mathbf{S}_{p,21} t_{12} \end{bmatrix}.$$

As $t_{11}, \mathbf{S}_{p,11}, \mathbf{S}_{p,21}$ are nonnegative, $t_{12}$ is nonpositive, the two properties also hold for $\mathbf{S}_{p+1}$.

Therefore, the two properties hold for $\mathbf{S}_{p+1}$ and thus for any combination term $\mathbf{S}$ by induction. And one can verify that the properties also hold for their summation $\prod_{i=1}^k (\mathbf{T} + \boldsymbol{\Delta}_i)$. Because of the definition of frobenius norm, when the two properties hold, $\left\| \prod_{i=1}^k (\mathbf{T} + \boldsymbol{\Delta}_i) \right\|_F$ is monotonically increasing with respect to $\delta_1, ... \delta_k$ as well. Therefore, it holds that

$$\left\| \prod_{i=1}^k (\mathbf{T}_{t,j} + \boldsymbol{\Delta}_i) \right\|_F \leq \left\| \prod_{i=1}^k (\mathbf{T}_{t,j} + \boldsymbol{\Delta}) \right\|_F = \left\| (\mathbf{T}_{t,j} + \boldsymbol{\Delta})^k \right\|_F,$$

which concludes the proof. $\qquad\square$

Then combining Lemma 6 and Lemma 7, we can obtain a conclusion.

**Lemma 8.** *Given* $\beta \in [1/4, 1)$, $\mathbf{T}_{t,j}$ *defined as Eqn. (C.10), if* $\mathbf{T}_{t,j}$ *only has real eigenvalues, which is equivalent to that the discriminant of* $\mathbf{T}_{t,j}$ *satisfies that* $(1 + \beta - \eta_t \lambda_j)^2 - 4\beta \geq 0$, *it holds that*

$$\|\mathbf{T}_{t+1,j} \mathbf{T}_{t+2,j} ... \mathbf{T}_{t+k,j}\| \leq \min\left(8k, \frac{8}{\sqrt{|(1+\beta - \eta_{t+k}\lambda_j)^2 - 4\beta|}}\right) \rho(\mathbf{T}_{t+k,j})^k. \tag{C.24}$$

*Proof.* The difference of two momentum matrices $\mathbf{T}_{t,j}$ and $\mathbf{T}_{t',j}$ that $\eta_{t'} \leq \eta_t$ is

$$\mathbf{T}_{t',j} - \mathbf{T}_{t,j} = \begin{bmatrix} (\eta_{t'} - \eta_t)\lambda_j & 0 \\ 0 & 0 \end{bmatrix},$$

which has the same structure with $\boldsymbol{\Delta}_i$ in Lemma 7. Thus Lemma 8 is a natural combination of Lemma 6 and Lemma 7. One can verify that

$$\|\mathbf{T}_{t+1,j} \mathbf{T}_{t+2,j} ... \mathbf{T}_{t+k,j}\| \leq \|\mathbf{T}_{t+1,j} \mathbf{T}_{t+2,j} ... \mathbf{T}_{t+k,j}\|_F \overset{(C.23)}{\leq} \left\| \mathbf{T}_{t+k,j}^k \right\|_F$$

$$\overset{(C.22)}{\leq} \min\left(8k, \frac{8}{\sqrt{|(1+\beta - \eta_{t+k}\lambda_j)^2 - 4\beta|}}\right) \rho(\mathbf{T}_{t+k,j})^k,$$

which concludes the proof. $\qquad\square$

### C.3 KEY PROPERTIES OF $\rho(\mathbf{T})$

This section offers some useful property of spectral radius $\rho(\mathbf{T}_{t,j})$ in terms of different $\eta_t$, $\beta$ and $\lambda_j$.

**Lemma 9** (The exact form of $\rho$ and its relationship with $\{\beta, \eta, \lambda\}$). *Given momentum matrix*

$$\mathbf{T} = \begin{bmatrix} 1 + \beta - \eta\lambda & -\beta \\ 1 & 0 \end{bmatrix},$$

*if $\mathbf{T}_{t,j}$ only has real eigenvalues, which is equivalent to that the discriminant $(1 + \beta - \eta\lambda)^2 - 4\beta > 0$ the spectral radius of $\mathbf{T}$ is*

$$\rho(\mathbf{T}) = \frac{1}{2}\left[1 + \beta - \eta\lambda + \sqrt{(1 + \beta - \eta\lambda)^2 - 4\beta}\right].$$

*Else the spectral radius of $\mathbf{T}$ is*

$$\rho(\mathbf{T}) = \sqrt{\beta}.$$

*Thus under the assumption that $\eta\lambda \leq 1$, we have $\rho(\mathbf{T})$ is monotonically decreasing with respect to $\eta\lambda$.*

*Proof.* To find the spectral radius, we first derive the eigenvalues of $\mathbf{T}$, which is equivalent to solving the equation

$$\det|\mathbf{T} - \gamma\mathbf{I}| = \det\begin{vmatrix} 1 + \beta - \eta\lambda - \gamma & -\beta \\ 1 & -\gamma \end{vmatrix} = 0.$$

After rearrangement we have

$$\gamma^2 - (1 + \beta - \eta\lambda)\gamma + \beta = 0,$$

which is a quadratic equation. The discriminant is that

$$\Delta = (1 + \beta - \eta\lambda)^2 - 4\beta = \left(\left(1 - \sqrt{\beta}\right)^2 - \eta\lambda\right)\left(\left(1 + \sqrt{\beta}\right)^2 - \eta\lambda\right).$$

Under the assumption that $\eta\lambda \leq 1$, the positivity of $\Delta$ depends on the positivity of $\left(1 - \sqrt{\beta}\right)^2 - \eta\lambda$. If $\Delta \geq 0$, then the eigenvalues $\gamma_1, \gamma_2 \in \mathbb{R}$. If $\Delta < 0$, then the eigenvalues $\gamma_1, \gamma_2 \notin \mathbb{R}$. We then discuss these two cases.

**Case 1:** $\Delta \geq 0$ In this case, we have

$$\gamma_1 = \frac{1}{2}\left[1 + \beta - \eta\lambda + \sqrt{(1 + \beta - \eta\lambda)^2 - 4\beta}\right],$$
$$\gamma_2 = \frac{1}{2}\left[1 + \beta - \eta\lambda - \sqrt{(1 + \beta - \eta\lambda)^2 - 4\beta}\right].$$

Then we have $\gamma_1 \geq \gamma_2$, thus the spectral radius

$$\rho(\mathbf{T}) = \gamma_1 = \frac{1}{2}\left[1 + \beta - \eta\lambda + \sqrt{(1 + \beta - \eta\lambda)^2 - 4\beta}\right].$$

Then we justify the monotonicity in this case. We have

$$\frac{\partial\rho(\mathbf{T})}{\partial(\eta\lambda)} = \frac{1}{2}\left[-1 + \frac{-(1 + \beta - \eta\lambda)}{\sqrt{(1 + \beta - \eta\lambda)^2 - 4\beta}}\right] < 0,$$

thus $\rho(\mathbf{T})$ is monotonically decreasing with respect to $\eta\lambda$.

**Case 2:** $\Delta < 0$ In this case, we have

$$\gamma_1 = \frac{1}{2}\left[1 + \beta - \eta\lambda + \sqrt{4\beta - (1 + \beta - \eta\lambda)^2}i\right],$$
$$\gamma_2 = \frac{1}{2}\left[1 + \beta - \eta\lambda - \sqrt{4\beta - (1 + \beta - \eta\lambda)^2}i\right],$$

where $i$ is the imaginary unit. One can observe that $\gamma_1$ is the complex conjugate of $\gamma_2$. Thus we have the spectral radius is

$$\rho(\mathbf{T}_{t,j}) = |\gamma_1| = |\gamma_2| = \sqrt{\gamma_1\bar{\gamma}_1} = \sqrt{\gamma_1\gamma_2} \overset{(D.2)}{=} \sqrt{\beta}.$$

In this case, $\rho(\mathbf{T}_{t,j})$ does not depend on $\eta\lambda$. Thus we finish the proof. $\qquad\square$

**Lemma 10** ($\rho$ bounded by $\eta, \lambda, \beta$ in real-eigenvalue case). *For momentum matrix*

$$\mathbf{T} = \begin{bmatrix} 1 + \beta - \eta\lambda & -\beta \\ 1 & 0 \end{bmatrix},$$

*if $\mathbf{T}_{t,j}$ only has real eigenvalues, which is equivalent to that the discriminant $(1 + \beta - \eta\lambda)^2 - 4\beta > 0$, it holds that*

$$\rho(\mathbf{T}) \leq 1 - \frac{\eta\lambda}{2} - \frac{\eta\lambda}{4\left(1 - \sqrt{\beta}\right)}. \tag{C.25}$$

*Proof.* According to Lemma 9, the spectral radius of $\mathbf{T}$ is

$$\rho(\mathbf{T}) = \frac{1 + \beta - \eta\lambda + \sqrt{(1 + \beta - \eta\lambda)^2 - 4\beta}}{2}.$$

It holds that

$$
\begin{aligned}
\rho(\mathbf{T}) &= \frac{1}{2}\left[1 + \beta - \eta\lambda + \sqrt{(1 + \beta - \eta\lambda)^2 - 4\beta}\right] \\
&= \frac{1}{2}\left[1 + \beta - \eta\lambda + \sqrt{\left(1 + \sqrt{\beta}\right)^2 - \eta\lambda} \cdot \sqrt{\left(1 - \sqrt{\beta}\right)^2 - \eta\lambda}\right] \\
&\leq \frac{1}{2}\left[1 + \beta - \eta\lambda + \left(1 + \sqrt{\beta}\right)\sqrt{\left(1 - \sqrt{\beta}\right)^2 - \eta\lambda}\right] \\
&= \frac{1}{2}\left[1 + \beta - \eta\lambda + (1 - \beta)\sqrt{1 - \frac{\eta\lambda}{\left(1 - \sqrt{\beta}\right)^2}}\right] \\
&\overset{(D.3)}{\leq} \frac{1}{2}\left[1 + \beta - \eta\lambda + (1 - \beta)\left(1 - \frac{\eta\lambda}{2\left(1 - \sqrt{\beta}\right)^2}\right)\right] \\
&= \frac{1}{2}\left[2 - \eta\lambda - \frac{(1 - \beta)\eta\lambda}{2(1 - \sqrt{\beta})^2}\right] \leq \frac{1}{2}\left[2 - \eta\lambda - \frac{\eta\lambda}{2(1 - \sqrt{\beta})}\right] \\
&= 1 - \frac{\eta\lambda}{2} - \frac{\eta\lambda}{4\left(1 - \sqrt{\beta}\right)}.
\end{aligned}
$$

$\square$

## C.4 PROOF OF THEOREM 2

In this section, we specify the schedule to step decay and prove Theorem 2. We denote the step size of the $\ell$-th stage $\eta'_\ell$ and its corresponding momentum matrix $\mathbf{T}'_{\ell,j}$ to specify the stagewise case.

We first present a lemma to simplify our proof in the stagewise case.

**Lemma 11.** *For all stage $\ell > 1$, given matrices $\mathbf{T}'_{\ell,j}$ defined in Eqn. (C.10), if $\eta'_\ell\lambda_j > (1 - \sqrt{\beta})^2$, and the length of the stage*

$$K \geq \sqrt{\kappa}\ln(8T),$$

*it holds that*

$$\left\|\left(\mathbf{T}'_{\ell,j}\right)^K\right\| \leq 1. \tag{C.26}$$

*Proof.* In this case, the eigenvalues of $\mathbf{T}'_{\ell,j}$ are not real and thus the spectral radius $\rho(\mathbf{T}'_{\ell,j}) = \sqrt{\beta}$ as Lemma 9 suggests. Thus it holds that

$$
\begin{aligned}
\left\|\left(\mathbf{T}'_{\ell,j}\right)^K\right\| &\leq \left\|\left(\mathbf{T}'_{\ell,j}\right)^K\right\|_F \overset{(C.22)}{\leq} 8K\rho\left(\mathbf{T}'_{\ell,j}\right)^K = 8K\left(\sqrt{\beta}\right)^K \\
&\leq 8T\left(1 - \frac{1}{\sqrt{\kappa}}\right)^{\sqrt{\kappa}\ln(8T)}
\end{aligned}
$$

$$\overset{(D.4)}{\leq} 8T \cdot e^{-\ln(8T)} = 1,$$

which concludes the proof. $\qquad\square$

Then we are ready to prove the convergence of step decay schedule.

**Theorem 2.** *Given a quadratic objective $f(\mathbf{w})$ and a step decay learning rate scheduler with $\beta = \left(1 - 1/\sqrt{\kappa}\right)^2$ with $\kappa \geq 4$, and $n \equiv T/K$ with settings that*

    1. *decay factor $C$*

$$1 < C \leq T\sqrt{\kappa}. \tag{3.13}$$

    2. *stepsize $\eta'_\ell$*

$$\eta'_\ell = \frac{1}{L} \cdot \frac{1}{C^{\ell-1}} \tag{3.14}$$

    3. *stage length $K$*

$$K = \frac{T}{\log_C\left(T\sqrt{\kappa}\right)} \tag{3.15}$$

    4. *total iteration number $T$*

$$\frac{T}{\ln\left(2^{14}T^8\right) \cdot \ln\left(2^6T^4\right) \cdot \log_C(T^2)} \geq 2C\sqrt{\kappa}, \tag{3.16}$$

*then such scheduler exists, and the output of Algorithm 1 satisfies*

$$\mathbb{E}[f(\mathbf{w}_T) - f(\mathbf{w}_*)] \leq \mathbb{E}\left[f(\mathbf{w}_0) - f(\mathbf{w}_*)\right] \cdot \exp\left(15\ln 2 + 2\ln T + 2\ln\kappa - \frac{2T}{\sqrt{\kappa}\log_C\left(T\sqrt{\kappa}\right)}\right)$$
$$+ \frac{4096C^2d\sigma^2}{MT}\ln^2\left(2^6T^4\right) \cdot \log_C^2\left(T\sqrt{\kappa}\right).$$

*Proof.* From (3.16), the total iteration number $T$ satisfies that

$$T \geq 2C\sqrt{\kappa}\ln\left(2^{14}T^6\kappa\right) \cdot \ln\left(2^6T^4\right) \cdot \log_C\left(T\sqrt{\kappa}\right), \tag{C.27}$$

and we define an auxiliary constant for our proof that

$$h \equiv h(T, \kappa) = 4\ln\left(2^6T^4\right) \cdot \log_C\left(T\sqrt{\kappa}\right) \geq 1. \tag{C.28}$$

From (3.16), we know that $T \geq 2C\sqrt{\kappa}$. Then with (C.27) and (C.28), we can verify that the following requirements are satisfied.

    1. From Lemma 11:

$$K \geq \sqrt{\kappa}\ln\left(8T\right). \tag{C.29}$$

    **Verify:** As $T \geq \sqrt{\kappa}$, it holds that

$$K \overset{(3.15)}{=} \frac{T}{\log_C(T\sqrt{\kappa})} \overset{(3.16)}{\geq} 2C\sqrt{\kappa}\ln(2^{14}T^8)\ln(2^6T^4) \geq \sqrt{\kappa}\ln(8T).$$

    2. From variance case 1.1, the final stage needs the variance to be small enough that $\eta'_n L \leq h/(T\sqrt{\kappa})$:

$$K \leq \frac{T}{\log_C\left(\frac{T\sqrt{\kappa}}{h}\right)}. \tag{C.30}$$

    **Verify:** As $T \geq \sqrt{\kappa}$ and $h \geq 1$, it holds that

$$K \overset{(3.15)}{=} \frac{T}{\log_C(T\sqrt{\kappa})} \leq \frac{T}{\log_C\left(\frac{T\sqrt{\kappa}}{h}\right)}.$$

3. From variance case 1.1:

$$K \leq \frac{T}{\log_C\left(\frac{4\kappa}{3}\right) + 1}. \tag{C.31}$$

**Verify:** As $T \geq 2C\sqrt{\kappa}$, it holds that

$$K \stackrel{(3.15)}{=} \frac{T}{\log_C(T\sqrt{\kappa})} \leq \frac{T}{\log_C\left(\frac{4\kappa}{3}\right) + 1}.$$

4. From variance case 1.2:

$$K \geq \frac{4T}{h} \ln\left(2^6 T^4\right). \tag{C.32}$$

**Verify:** It holds that

$$h \stackrel{(C.28)}{=} 4 \ln\left(2^6 T^4\right) \cdot \log_C\left(T\sqrt{\kappa}\right) \stackrel{(3.15)}{=} \frac{4T}{K} \ln\left(2^6 T^4\right).$$

5. From variance case 2.1:

$$K \geq 2C\sqrt{\kappa} \ln\left(2^{12} T^6\right). \tag{C.33}$$

**Verify:** It holds that

$$K \stackrel{(3.15)}{=} \frac{T}{\log_C(T\sqrt{\kappa})} \stackrel{(3.16)}{\geq} 2C\sqrt{\kappa} \ln(2^{14} T^8) \ln(2^6 T^4) \geq 2C\sqrt{\kappa} \ln\left(2^{12} T^6\right).$$

6. From variance case 2.2:

$$K \geq \frac{\sqrt{\kappa}}{2} \ln\left(2^{14} T^6 \kappa\right). \tag{C.34}$$

**Verify:** As $T \geq \sqrt{\kappa}$, it holds that

$$K \stackrel{(3.15)}{=} \frac{T}{\log_C(T\sqrt{\kappa})} \stackrel{(3.16)}{\geq} 2C\sqrt{\kappa} \ln(2^{14} T^8) \ln(2^6 T^4) \geq \frac{\sqrt{\kappa}}{2} \ln\left(2^{14} T^6 \kappa\right).$$

Now we are ready to start our main analysis. From the former lemmas, it holds that

$$
\begin{aligned}
&2\mathbb{E}\left[f(\mathbf{w}_T) - f(\mathbf{w}_*)\right] \\
&\stackrel{(C.1)}{=} (\mathbf{w}_T - \mathbf{w}_*)^\top \mathbf{H}(\mathbf{w}_T - \mathbf{w}_*) \\
&\stackrel{(C.2)}{=} \mathbb{E}\left[\begin{bmatrix}\mathbf{w}_0 - \mathbf{w}_* \\ \mathbf{w}_{-1} - \mathbf{w}_*\end{bmatrix}^\top \mathbf{M}_0^\top \mathbf{M}_1^\top \ldots \mathbf{M}_{T-1}^\top \begin{bmatrix}\mathbf{H} & \mathbf{O} \\ \mathbf{O} & \mathbf{O}\end{bmatrix} \mathbf{M}_{T-1}\mathbf{M}_{T-2}\ldots\mathbf{M}_0 \begin{bmatrix}\mathbf{w}_0 - \mathbf{w}_* \\ \mathbf{w}_{-1} - \mathbf{w}_*\end{bmatrix}\right] \\
&\quad + \sum_{\tau=0}^{T-1} \mathbb{E}\left[\eta_\tau^2 \begin{bmatrix}\mathbf{n}_\tau \\ \mathbf{0}\end{bmatrix}^\top \mathbf{M}_{\tau+1}^\top \mathbf{M}_{\tau+2}^\top \ldots \mathbf{M}_{T-1}^\top \begin{bmatrix}\mathbf{H} & \mathbf{O} \\ \mathbf{O} & \mathbf{O}\end{bmatrix} \mathbf{M}_{T-1}\mathbf{M}_{T-2}\ldots\mathbf{M}_{\tau+1} \begin{bmatrix}\mathbf{n}_\tau \\ \mathbf{0}\end{bmatrix}\right] \\
&\stackrel{(C.15)}{\leq} \stackrel{(C.20)}{} \sum_{j=1}^{d} \lambda_j \left\|\mathbf{T}_{T-1,j}\mathbf{T}_{T-2,j}\ldots\mathbf{T}_{0,j}\right\|^2 \mathbb{E}\left\|\left(\boldsymbol{\Pi}^\top \mathbf{V}^\top \begin{bmatrix}\mathbf{w}_0 - \mathbf{w}_* \\ \mathbf{w}_{-1} - \mathbf{w}_*\end{bmatrix}\right)_{2j-1:2j}\right\|^2 \\
&\quad + \sigma^2 \sum_{j=1}^{d} \lambda_j^2 \sum_{\tau=0}^{T-1} \eta_\tau^2 \left\|\mathbf{T}_{T-1,j}\mathbf{T}_{T-2,j}\ldots\mathbf{T}_{\tau+1,j}\right\|^2.
\end{aligned}
$$

If we denote bias and variance as

$$B \triangleq \sum_{j=1}^{d} \lambda_j \left\|\mathbf{T}_{T-1,j}\mathbf{T}_{T-2,j}...\mathbf{T}_{0,j}\right\|^2 \mathbb{E}\left\|\left(\boldsymbol{\Pi}^\top \mathbf{V}^\top \begin{bmatrix}w_0 - w_* \\ w_{-1} - w_*\end{bmatrix}\right)_{2j-1:2j}\right\|^2,$$

$$V \triangleq \sigma^2 \sum_{j=1}^{d} \lambda_j^2 \sum_{\tau=0}^{T-1} \eta_\tau^2 \left\|\mathbf{T}_{T-1,j}\mathbf{T}_{T-2,j}...\mathbf{T}_{\tau+1,j}\right\|^2,$$

we have the following results.

**(1) Bounding bias term:**

It holds that

$$B = \sum_{j=1}^{d} \lambda_j \left\| \mathbf{T}_{T-1,j} \mathbf{T}_{T-2,j} ... \mathbf{T}_{0,j} \right\|^2 \mathbb{E} \left\| \left( \mathbf{\Pi}^\top \mathbf{V}^\top \begin{bmatrix} w_0 - w_* \\ w_{-1} - w_* \end{bmatrix} \right)_{2j-1:2j} \right\|^2$$

$$= \sum_{j=1}^{d} \lambda_j \left\| \mathbf{T}_{T-1,j} \mathbf{T}_{T-2,j} ... \mathbf{T}_{0,j} \right\|^2 \mathbb{E} \left\| \left( \mathbf{\Pi}^\top \mathbf{V}^\top \tilde{\mathbf{w}}_0 \right)_{2j-1:2j} \right\|^2$$

$$= \sum_{j=1}^{d} \lambda_j \left\| \left( \mathbf{T}'_{n,j} \right)^{k_n} \left( \mathbf{T}'_{n_{\ell-1},j} \right)^{k_{n_{\ell-1}}} ... \left( \mathbf{T}'_{1,j} \right)^{k_1} \right\|^2 \mathbb{E} \left\| \left( \mathbf{\Pi}^\top \mathbf{V}^\top \tilde{\mathbf{w}}_0 \right)_{2j-1:2j} \right\|^2$$

$$\overset{\text{(C.26), (C.24)}}{\leq} \sum_{j=1}^{d} \lambda_j \left\| \left( \mathbf{T}'_{1,j} \right)^{k_1} \right\|^2 \left( \frac{8}{\sqrt{(1+\beta-\eta_{T-1}\lambda_j)^2 - 4\beta}} \right)^2 \mathbb{E} \left\| \left( \mathbf{\Pi}^\top \mathbf{V}^\top \tilde{\mathbf{w}}_0 \right)_{2j-1:2j} \right\|^2$$

$$\leq \frac{256}{\eta'_1 \mu} \sum_{j=1}^{d} \lambda_j \left\| \left( \mathbf{T}'_{1,j} \right)^{k_1} \right\|^2 \mathbb{E} \left\| \left( \mathbf{\Pi}^\top \mathbf{V}^\top \tilde{\mathbf{w}}_0 \right)_{2j-1:2j} \right\|^2$$

$$\overset{\text{(C.22)}}{\leq} \frac{256}{\eta'_1 \mu} \sum_{j=1}^{d} \lambda_j \left( 8 k_1 \rho \left( \mathbf{T}'_{1,j} \right)^{k_1} \right)^2 \cdot \mathbb{E} \left\| \left( \mathbf{\Pi}^\top \mathbf{V}^\top \tilde{\mathbf{w}}_0 \right)_{2j-1:2j} \right\|^2$$

$$\overset{\text{Lem. 9}}{\leq} \frac{256}{\eta'_1 \mu} \sum_{j=1}^{d} \lambda_j \left( 8 k_1 \rho \left( \mathbf{T}'_{1,d} \right)^{k_1} \right)^2 \cdot \mathbb{E} \left\| \left( \mathbf{\Pi}^\top \mathbf{V}^\top \tilde{\mathbf{w}}_0 \right)_{2j-1:2j} \right\|^2$$

$$= \frac{256}{\eta'_1 \mu} \left( 8 k_1 \rho \left( \mathbf{T}'_{1,d} \right)^{k_1} \right)^2 \cdot \sum_{j=1}^{d} \lambda_j \mathbb{E} \left\| \left( \mathbf{\Pi}^\top \mathbf{V}^\top \tilde{\mathbf{w}}_0 \right)_{2j-1:2j} \right\|^2$$

$$= \frac{256}{\eta'_1 \mu} \left( 8 k_1 \rho \left( \mathbf{T}'_{1,d} \right)^{k_1} \right)^2 \cdot \mathbb{E} \left[ \sum_{j=1}^{d} \lambda_j \left( \mathbf{\Pi}^\top \mathbf{V}^\top \tilde{\mathbf{w}}_0 \right)_{2j-1:2j}^\top \left( \mathbf{\Pi}^\top \mathbf{V}^\top \tilde{\mathbf{w}}_0 \right)_{2j-1:2j} \right]$$

$$= \frac{256}{\eta'_1 \mu} \left( 8 k_1 \rho \left( \mathbf{T}'_{1,d} \right)^{k_1} \right)^2 \cdot \mathbb{E} \left[ \sum_{j=1}^{d} \left( \mathbf{\Pi}^\top \mathbf{V}^\top \tilde{\mathbf{w}}_0 \right)_{2j-1:2j}^\top \begin{bmatrix} \lambda_j & 0 \\ 0 & \lambda_j \end{bmatrix} \left( \mathbf{\Pi}^\top \mathbf{V}^\top \tilde{\mathbf{w}}_0 \right)_{2j-1:2j} \right]$$

$$= \frac{256}{\eta'_1 \mu} \left( 8 k_1 \rho \left( \mathbf{T}'_{1,d} \right)^{k_1} \right)^2$$

$$\cdot \mathbb{E} \left[ \begin{bmatrix} \left( \mathbf{\Pi}^\top \mathbf{V}^\top \tilde{\mathbf{w}}_0 \right)_{1:2} \\ \left( \mathbf{\Pi}^\top \mathbf{V}^\top \tilde{\mathbf{w}}_0 \right)_{3:4} \\ \vdots \\ \left( \mathbf{\Pi}^\top \mathbf{V}^\top \tilde{\mathbf{w}}_0 \right)_{2d-1:2d} \end{bmatrix}^\top \begin{bmatrix} \mathbf{S}_1 & & & \\ & \mathbf{S}_2 & & \\ & & \ddots & \\ & & & \mathbf{S}_d \end{bmatrix} \begin{bmatrix} \left( \mathbf{\Pi}^\top \mathbf{V}^\top \tilde{\mathbf{w}}_0 \right)_{1:2} \\ \left( \mathbf{\Pi}^\top \mathbf{V}^\top \tilde{\mathbf{w}}_0 \right)_{3:4} \\ \vdots \\ \left( \mathbf{\Pi}^\top \mathbf{V}^\top \tilde{\mathbf{w}}_0 \right)_{2d-1:2d} \end{bmatrix} \right],$$

with $\mathbf{S}_j = \begin{bmatrix} \lambda_j & 0 \\ 0 & \lambda_j \end{bmatrix}$

$$= \frac{256}{\eta'_1 \mu} \left( 8 k_1 \rho \left( \mathbf{T}'_{1,d} \right)^{k_1} \right)^2 \cdot \mathbb{E} \left[ \left( \mathbf{\Pi}^\top \mathbf{V}^\top \tilde{\mathbf{w}}_0 \right)^\top \left( \mathbf{\Lambda} \otimes \begin{bmatrix} 1 & 0 \\ 0 & 1 \end{bmatrix} \right) \left( \mathbf{\Pi}^\top \mathbf{V}^\top \tilde{\mathbf{w}}_0 \right) \right]$$

$$= \frac{256}{\eta'_1 \mu} \left( 8 k_1 \rho \left( \mathbf{T}'_{1,d} \right)^{k_1} \right)^2 \cdot \mathbb{E} \left[ \tilde{\mathbf{w}}_0^\top \left( \mathbf{V} \mathbf{\Pi} \left( \mathbf{\Lambda} \otimes \begin{bmatrix} 1 & 0 \\ 0 & 1 \end{bmatrix} \right) \mathbf{\Pi}^\top \mathbf{V}^\top \right) \tilde{\mathbf{w}}_0 \right],$$

where the first inequality is because that Eqn. C.26 can be applied for the stages $\ell$ where $\mathbf{T}'_{\ell,j}$ has only complex eigenvalues and Eqn. C.24 can be applied to the matrix product for all stages $\ell$ that $\mathbf{T}_{\ell,j}$ has only real eigenvalues. And the second inequality holds as

$$\sqrt{(1+\beta-\eta'_n\lambda_j)^2 - 4\beta} = \sqrt{(1-\sqrt{\beta})^2 - \eta'_n\lambda_j} \cdot \sqrt{(1+\sqrt{\beta})^2 - \eta'_n\lambda_j}$$

$$\geq \sqrt{(1-\sqrt{\beta})^2 - \eta'_n \lambda_j} = \sqrt{(1-\sqrt{\beta})^2 - C^{-n_\ell+1} \cdot \eta'_1 \lambda_j}$$

$$\overset{(C.31)}{\geq} \sqrt{(1-\sqrt{\beta})^2 - \frac{3}{4}(1-\sqrt{\beta})^2} = \frac{1}{2}(1-\sqrt{\beta}) = \frac{1}{2}\sqrt{\eta'_1 \mu}.$$

Notice that

$$\mathbf{V}\boldsymbol{\Pi}\left(\boldsymbol{\Lambda} \otimes \begin{bmatrix} 1 & 0 \\ 0 & 1 \end{bmatrix}\right)\boldsymbol{\Pi}^\top \mathbf{V}^\top = \begin{bmatrix} \mathbf{H} & \mathbf{O} \\ \mathbf{O} & \mathbf{H} \end{bmatrix}$$

since

$$\boldsymbol{\Pi}^\top \mathbf{V}^\top \begin{bmatrix} \mathbf{H} & \mathbf{O} \\ \mathbf{O} & \mathbf{H} \end{bmatrix} \mathbf{V}\boldsymbol{\Pi}$$

$$\overset{(C.11)}{=} \begin{bmatrix} \mathbf{e}_1^\top & \mathbf{0}^\top \\ \mathbf{0}^\top & \mathbf{e}_1^\top \\ \vdots \\ \mathbf{e}_d^\top & \mathbf{0}^\top \\ \mathbf{0}^\top & \mathbf{e}_d^\top \end{bmatrix} \begin{bmatrix} \mathbf{U}^\top & \mathbf{O} \\ \mathbf{O} & \mathbf{U}^\top \end{bmatrix} \begin{bmatrix} \mathbf{H} & \mathbf{O} \\ \mathbf{O} & \mathbf{H} \end{bmatrix} \begin{bmatrix} \mathbf{U} & \mathbf{O} \\ \mathbf{O} & \mathbf{U} \end{bmatrix} \begin{bmatrix} \mathbf{e}_1 & \mathbf{0} & \mathbf{e}_2 & \mathbf{0} & \dots & \mathbf{e}_d & \mathbf{0} \\ \mathbf{0} & \mathbf{e}_1 & \mathbf{0} & \mathbf{e}_2 & \dots & \mathbf{0} & \mathbf{e}_d \end{bmatrix}$$

$$\overset{(C.12)}{=} \begin{bmatrix} \mathbf{e}_1^\top & \mathbf{0}^\top \\ \mathbf{0}^\top & \mathbf{e}_1^\top \\ \vdots \\ \mathbf{e}_d^\top & \mathbf{0}^\top \\ \mathbf{0}^\top & \mathbf{e}_d^\top \end{bmatrix} \begin{bmatrix} \boldsymbol{\Lambda} & \mathbf{O} \\ \mathbf{O} & \boldsymbol{\Lambda} \end{bmatrix} \begin{bmatrix} \mathbf{e}_1 & \mathbf{0} & \mathbf{e}_2 & \mathbf{0} & \dots & \mathbf{e}_d & \mathbf{0} \\ \mathbf{0} & \mathbf{e}_1 & \mathbf{0} & \mathbf{e}_2 & \dots & \mathbf{0} & \mathbf{e}_d \end{bmatrix}$$

$$= \begin{bmatrix} \lambda_1 & & & & & & \\ & \lambda_1 & & & & & \\ & & \lambda_2 & & & & \\ & & & \lambda_2 & & & \\ & & & & \ddots & & \\ & & & & & \lambda_d & \\ & & & & & & \lambda_d \end{bmatrix} = \boldsymbol{\Lambda} \otimes \begin{bmatrix} 1 & 0 \\ 0 & 1 \end{bmatrix}$$

and $\boldsymbol{\Pi}, \mathbf{V}$ are orthogonal matrices (Eqn. (C.14)). We then further have

$$B \leq \frac{256}{\eta'_1 \mu}\left(8k_1 \rho\left(\mathbf{T}'_{1,d}\right)^{k_1}\right)^2 \cdot \mathbb{E}\left[\tilde{\mathbf{w}}_0^\top \begin{bmatrix} \mathbf{H} & \mathbf{O} \\ \mathbf{O} & \mathbf{H} \end{bmatrix} \tilde{\mathbf{w}}_0\right]$$

$$\overset{(C.4)}{=} \frac{256}{\eta'_1 \mu}\left(8k_1 \rho\left(\mathbf{T}'_{1,d}\right)^{k_1}\right)^2 \cdot \mathbb{E}\left[\begin{bmatrix} \mathbf{w}_0 - \mathbf{w}_* \\ \mathbf{w}_{-1} - \mathbf{w}_* \end{bmatrix}^\top \begin{bmatrix} \mathbf{H} & \mathbf{O} \\ \mathbf{O} & \mathbf{H} \end{bmatrix} \begin{bmatrix} \mathbf{w}_0 - \mathbf{w}_* \\ \mathbf{w}_{-1} - \mathbf{w}_* \end{bmatrix}\right]$$

$$= \frac{256}{\eta'_1 \mu}\left(8k_1 \rho\left(\mathbf{T}'_{1,d}\right)^{k_1}\right)^2 \cdot \mathbb{E}\left[(\mathbf{w}_0 - \mathbf{w}_*)^\top \mathbf{H}(\mathbf{w}_0 - \mathbf{w}_*) + (\mathbf{w}_{-1} - \mathbf{w}_*)^\top \mathbf{H}(\mathbf{w}_{-1} - \mathbf{w}_*)\right]$$

$$\overset{(C.1)}{=} \frac{256}{\eta'_1 \mu}\left(8k_1 \rho\left(\mathbf{T}'_{1,d}\right)^{k_1}\right)^2 \cdot \mathbb{E}\left[f(\mathbf{w}_{-1}) + f(\mathbf{w}_0) - 2f(\mathbf{w}_*)\right]$$

$$= \frac{256}{\eta'_1 \mu}\left(8k_1\left(\sqrt{\beta}\right)^{k_1}\right)^2 \cdot \mathbb{E}\left[f(\mathbf{w}_{-1}) + f(\mathbf{w}_0) - 2f(\mathbf{w}_*)\right]$$

$$= \frac{256}{\eta'_1 \mu}\left(8k_1\left(1 - \sqrt{\eta'_1 \mu}\right)^{k_1}\right)^2 \cdot \mathbb{E}\left[f(\mathbf{w}_{-1}) + f(\mathbf{w}_0) - 2f(\mathbf{w}_*)\right]$$

$$\leq \exp\left(14\ln 2 + 2\ln k_1 - 2\ln(\eta'_1 \mu) - 2k_1\sqrt{\eta'_1 \mu}\right) \cdot \mathbb{E}\left[f(\mathbf{w}_{-1}) + f(\mathbf{w}_0) - 2f(\mathbf{w}_*)\right]$$

$$= \mathbb{E}\left[f(\mathbf{w}_{-1}) + f(\mathbf{w}_0) - 2f(\mathbf{w}_*)\right] \cdot \exp\left(14\ln 2 + 2\ln k_1 - 2\ln\left(\frac{1}{\kappa}\right) - \frac{2T}{\sqrt{\kappa}\log_C(T\sqrt{\kappa})}\right)$$

$$\leq \mathbb{E}\left[f(\mathbf{w}_{-1}) + f(\mathbf{w}_0) - 2f(\mathbf{w}_*)\right] \cdot \exp\left(14\ln 2 + 2\ln T + 2\ln \kappa - \frac{2T}{\sqrt{\kappa}\log_C(T\sqrt{\kappa})}\right)$$

$$= \mathbb{E}\left[f(\mathbf{w}_0) - f(\mathbf{w}_*)\right] \cdot \exp\left(15\ln 2 + 2\ln T + 2\ln\kappa - \frac{2T}{\sqrt{\kappa}\log_C\left(T\sqrt{\kappa}\right)}\right),$$

where the second last inequality is because of the fact that $1 - x \leq \exp(-x)$ for $x \geq 0$ and the last equality is because of the setting of Algorithm 1 that $\mathbf{v}_0 = \mathbf{0}$ and thus $\mathbf{w}_{-1} = \mathbf{w}_0$.

**(2) Bounding variance term:**

We denote

$$V_{t,j} \triangleq \eta_t^2\lambda_j^2 \left\|\mathbf{T}_{T-1,j}\mathbf{T}_{T-2,j}...\mathbf{T}_{t+1,j}\right\|^2$$

in the following analysis. We first assume that the batch size $M = 1$ in the main analysis and we will transfer the result to the general $M \geq 1$ case.

We make use of Lemma 9 and 10 and divide the analysis of $V_{t,j}$ into 4 cases with respect to $\eta_t\lambda_j$ and equivalently the corresponding eigenvalues. The division of the 4 cases is due to two major boarders: $b_1 = (1 - \sqrt{\beta})^2$ and $b_2 = h/(T\sqrt{\kappa})$.

- For momentum matrix $\mathbf{T}_{t,j}$ with $\eta_t\lambda_j > b_1$, $\mathbf{T}$ has complex eigenvalues and a large spectral radius that supports sufficient geometric decay of $V_{t,j}$, the variance generated at iteration $t$ on $\lambda_j$. Case 2.1 and Case 2.2 discuss this case.

- For momentum matrix $\mathbf{T}_{t,j}$ with $b_2 < \eta_t\lambda_j \leq b_1$, $\mathbf{T}_{t,j}$ has real eigenvalues but its spectral radius is still large enough to support sufficient geometric decay of $V_{t,j}$. Case 1.2 discusses this case.

- For momentum matrix $\mathbf{T}_{t,j}$ with $\eta_t\lambda_j \leq b_2$, the corresponding step size $\eta_t\lambda_j$ is small enough to ensure the generated variance $V_{t,j}$ is small. Case 1.1 discusses this case.

Among this cases, we use requirement (C.31) to ensure that Case 1.1 exists and thus the variance can be small enough. Let's discuss $V_{t,j}$ case by case then.

**Case 1.1: Real eigenvalues with small step size.** We first consider the case that $\mathbf{T}_{t,j}$ only has real eigenvalues. This case is equivalent to that

$$(1 + \beta - \eta_t\lambda_j)^2 - 4\beta \geq 0.$$

It is also equivalent to that

$$\eta_t\lambda_j \leq (1 - \sqrt{\beta})^2 = \eta_1'\mu. \tag{C.35}$$

And in this case we further assume that $\eta_t\lambda_j$ is small enough that

$$\eta_t\lambda_j \leq \frac{Ch}{T\sqrt{\kappa}}, \tag{C.36}$$

where $h$ is defined in (C.28). We also discuss From requirement (C.30), this case exists. Then from the fact that Frobenius norm of a matrix is always larger than $\ell_2$ norm and using Lemma 8 we can obtain that

$$\begin{aligned}
\left\|\mathbf{T}_{T-1,j}\mathbf{T}_{T-2,j}...\mathbf{T}_{\tau+1,j}\right\| &\leq \left\|\mathbf{T}_{T-1,j}\mathbf{T}_{T-2,j}...\mathbf{T}_{\tau+1,j}\right\|_F \\
&\overset{(C.24)}{\leq} \min\left(8(T-1-\tau), \frac{8}{\sqrt{(1+\beta-\eta_{T-1}\lambda_j)^2 - 4\beta}}\right)\rho(\mathbf{T}_{T-1,j})^{T-1-\tau} \\
&\leq \frac{8}{\sqrt{(1+\beta-\eta_{T-1}\lambda_j)^2 - 4\beta}}\rho(\mathbf{T}_{T-1,j})^{T-1-\tau} \\
&= \frac{8}{\sqrt{(1+\beta-\eta_n'\lambda_j)^2 - 4\beta}}\rho(\mathbf{T}_{T-1,j})^{T-1-\tau}.
\end{aligned}$$

$$\tag{C.37}$$

Then we analyze the right hand side term by term. First it holds that

$$\sqrt{(1 + \beta - \eta'_n \lambda_j)^2 - 4\beta} = \sqrt{(1 - \sqrt{\beta})^2 - \eta'_n \lambda_j} \cdot \sqrt{(1 + \sqrt{\beta})^2 - \eta'_n \lambda_j}$$

$$\geq \sqrt{(1 - \sqrt{\beta})^2 - \eta'_n \lambda_j} = \sqrt{(1 - \sqrt{\beta})^2 - C^{-n_\ell+1} \cdot \eta'_1 \lambda_j} \qquad \text{(C.38)}$$

$$\overset{(C.31)}{\geq} \sqrt{(1 - \sqrt{\beta})^2 - \frac{3}{4}(1 - \sqrt{\beta})^2} = \frac{1}{2}(1 - \sqrt{\beta}) = \frac{1}{2}\sqrt{\eta'_1 \mu}.$$

From Lemma 9, $\rho(\mathbf{T}_{t,j}) < 1$ holds for all $t, j$. Combining all above we can obtain that for $t, j$ satisfying $\eta_t \lambda_j \leq h/(T\sqrt{\kappa})$, it holds that

$$V_{t,j} = \eta_t^2 \lambda_j^2 \|\mathbf{T}_{T-1,j}\mathbf{T}_{T-2,j}...\mathbf{T}_{t+1,j}\|^2$$

$$\overset{(C.36), (C.37)}{\leq} \left(\frac{Ch}{T\sqrt{\kappa}}\right)^2 \left(\frac{8}{\sqrt{(1 + \beta - \eta'_n \lambda_j)^2 - 4\beta}}\right)^2$$

$$\overset{(C.38)}{\leq} \frac{C^2 h^2}{T^2 \kappa} \frac{64}{\left(\frac{1}{2}\sqrt{\eta'_1 \mu}\right)^2} = \frac{256 C^2 h^2}{T^2 \kappa \eta'_1 \mu} = \frac{256 C^2 h^2}{T^2},$$

where the last equality holds as $\eta'_1 = 1/L$.

**Case 1.2: Real eigenvalues with large step size.** Then we continue to consider the case that $\mathbf{T}_{t,j}$ only has real eigenvalues. Similar to case 1.1, this case is equivalent to that

$$(1 + \beta - \eta_t \lambda_j)^2 - 4\beta \geq 0.$$

It is also equivalent to that

$$\eta_t \lambda_j \leq (1 - \sqrt{\beta})^2 = \eta'_1 \mu. \qquad \text{(C.39)}$$

But in this case we further assume that $\eta_t \lambda_j$ is large enough that

$$\eta_t \lambda_j \geq \frac{Ch}{T\sqrt{\kappa}}$$

as the case that $\eta_t \lambda_j \leq Ch/(T\sqrt{\kappa})$ has been discussed in case 1.1. From requirement (3.16) we know that such stage exists. Then according to Lemma 10, with $\eta_t \lambda_j \geq h/(T\sqrt{\kappa})$, it holds that

$$\rho(\mathbf{T}_{t,j}) = \frac{1}{2}\left[1 + \beta - \eta_t \lambda_j + \sqrt{(1 + \beta - \eta_t \lambda_j)^2 - 4\beta}\right]$$

$$\overset{(C.25)}{\leq} 1 - \frac{\eta_t \lambda_j}{4(1 - \sqrt{\beta})} = 1 - \frac{\eta_t \lambda_j}{4\sqrt{\eta'_1 \mu}}$$

$$\leq 1 - \frac{h}{4T}.$$

Thus if we denote $t_*$ to be the first iteration that $\eta_{t_*} \lambda_j \leq h/(T\sqrt{\kappa})$, it holds that

$$V_{t,j} = \eta_t^2 \lambda_j^2 \|\mathbf{T}_{T-1,j}\mathbf{T}_{T-2,j}...\mathbf{T}_{t+1,j}\|^2$$

$$\leq \eta_t^2 \lambda_j^2 \|\mathbf{T}_{T-1,j}\mathbf{T}_{T-2,j}...\mathbf{T}_{t_*,j}\|^2 \|\mathbf{T}_{t_*-1,j}\mathbf{T}_{t_*-2,j}...\mathbf{T}_{t+1,j}\|^2$$

$$\overset{(C.23)}{\leq} \eta_t^2 \lambda_j^2 \|\mathbf{T}_{T-1,j}\mathbf{T}_{T-2,j}...\mathbf{T}_{t_*,j}\|^2 \|\mathbf{T}_{t_*,j}^{t_*-t-1}\|^2$$

$$\overset{(C.38), (C.22)}{\leq} \eta_t^2 \lambda_j^2 \left(\frac{8}{\sqrt{(1 + \beta - \eta'_n \lambda_j)^2 - 4\beta}}\right)^2 \cdot (8(t_* - t - 1))^2 \left(1 - \frac{h}{4T}\right)^{2(t_*-t-1)}$$

$$\overset{(C.38)}{\leq} \eta_t^2 \lambda_j^2 \frac{2^{12} T^2}{\left(\frac{1}{2}\sqrt{\eta'_1 \mu}\right)^2} \left(1 - \frac{h}{4T}\right)^{2(t_*-t-1)} \leq \eta_t^2 \lambda_j^2 \frac{2^{12} T^2}{\left(\frac{1}{2}\sqrt{\eta'_1 \mu}\right)^2} \left(1 - \frac{h}{4T}\right)^{2K}$$

$$\overset{(C.32), (C.39)}{\leq} (\eta'_1 \mu)^2 \frac{2^{12} T^2}{\left(\frac{1}{2}\sqrt{\eta'_1 \mu}\right)^2} \left(1 - \frac{h}{4T}\right)^{\frac{4T}{h} \cdot 2\ln(2^6 T^4)} \overset{(D.4)}{\leq} \frac{256}{T^2 \kappa},$$

where the third last inequality holds as $\eta_t \lambda_j \geq Ch/(T\sqrt{\kappa})$, which suggests that there is at least one stage between $t$ and $t_*$.

**Case 2.1: Complex eigenvalues with small step size.** Then we consider the case that $\mathbf{T}'_{\ell,j}$ has complex eigenvalues but $\mathbf{T}'_{\ell+1,j}$ has real eigenvalues, which is equivalent to that

$$C(1-\sqrt{\beta})^2 \geq \eta_t \lambda_j > (1-\sqrt{\beta})^2 = \eta'_1 \mu. \tag{C.40}$$

Thus if we denote $t_*$ to be the first iteration in the $\ell+1$ stage, consider the spectral radius of the $\ell+1$ stage, it holds that

$$
\begin{aligned}
\rho(\mathbf{T}_{t_*,j}) = \rho(\mathbf{T}'_{\ell+1,j}) = & \frac{1}{2}\left[1 + \beta - \eta'_{\ell+1}\lambda_j + \sqrt{(1+\beta-\eta'_{\ell+1}\lambda_j)^2 - 4\beta}\right] \\
& \overset{(C.25)}{\leq} 1 - \frac{\eta'_{\ell+1}\lambda_j}{4(1-\sqrt{\beta})} = 1 - \frac{\eta'_{\ell+1}\lambda_j}{4\sqrt{\eta'_1 \mu}} \\
& \overset{(C.40)}{\leq} 1 - \frac{\eta'_1 \mu}{4C\sqrt{\eta'_1 \mu}} = 1 - \frac{\sqrt{\eta'_1 \mu}}{4C}.
\end{aligned} \tag{C.41}
$$

Thus we have

$$
\begin{aligned}
V_{t,j} = & \eta_t^2 \lambda_j^2 \left\|\mathbf{T}_{T-1,j}\mathbf{T}_{T-2,j}...\mathbf{T}_{t+1,j}\right\|^2 \\
\leq & (C\eta'_{\ell+1}\lambda_j)^2 \left\|\mathbf{T}_{T-1,j}\mathbf{T}_{T-2,j}...\mathbf{T}_{t+1,j}\right\|^2 \overset{(C.35)}{\leq} C^2(\eta'_1\mu)^2 \left\|\mathbf{T}_{T-1,j}\mathbf{T}_{T-2,j}...\mathbf{T}_{t+1,j}\right\|^2 \\
\leq & C^2(\eta'_1\mu)^2 \left\|\mathbf{T}_{T-1,j}\mathbf{T}_{T-2,j}...\mathbf{T}_{t_*+K,j}\right\|^2 \left\|\mathbf{T}_{t_*+K-1,j}\mathbf{T}_{T-2,j}...\mathbf{T}_{t_*,j}\right\|^2 \left\|\mathbf{T}_{t_*-1,j}\mathbf{T}_{t_*-2,j}...\mathbf{T}_{t+1,j}\right\|^2 \\
\overset{(C.38)}{\leq} & C^2(\eta'_1\mu)^2 \cdot \frac{256}{\eta'_1\mu} \left\|\mathbf{T}_{t_*+K-1,j}\mathbf{T}_{T-2,j}...\mathbf{T}_{t_*,j}\right\|^2 \left\|\mathbf{T}_{t_*-1,j}\mathbf{T}_{t_*-2,j}...\mathbf{T}_{t+1,j}\right\|^2 \\
= & 256C^2\eta'_1\mu \cdot \left\|\left(\mathbf{T}'_{\ell+1,j}\right)\right\|^2 \left\|\mathbf{T}_{t_*-1,j}\mathbf{T}_{t_*-2,j}...\mathbf{T}_{t+1,j}\right\|^2 \\
\overset{(C.41)}{\leq} & 256C^2\eta'_1\mu \cdot \left(8T\left(1-\frac{\sqrt{\eta'_1\mu}}{4C}\right)^{K+1}\right)^2 \left\|\mathbf{T}_{t_*-1,j}\mathbf{T}_{t_*-2,j}...\mathbf{T}_{t+1,j}\right\|^2 \\
\leq & 256C^2\eta'_1\mu \cdot \left(8T\left(1-\frac{\sqrt{\eta'_1\mu}}{4C}\right)^{K}\right)^2 \cdot \max_{0\leq i \leq K}\left\|\left(\mathbf{T}'_{\ell,j}\right)^{K-i}\right\|^2 \\
\overset{(C.22)}{\leq} & 256C^2\eta'_1\mu \cdot \left(8T\left(1-\frac{\sqrt{\eta'_1\mu}}{4C}\right)^{K}\right)^2 \cdot (8K)^2 \leq 2^{20}C^2T^4\eta'_1\mu\left(1-\frac{\sqrt{\eta'_1\mu}}{4C}\right)^{2K} \\
\overset{(C.33),\ (C.40)}{\leq} & 2^{20}C^2T^4\eta'_1\mu\left(1-\frac{\sqrt{\eta'_1\mu}}{4C}\right)^{4C\sqrt{\kappa}\cdot\ln(2^{12}T^6)} \overset{(D.4)}{\leq} \frac{256C^2}{T^2},
\end{aligned}
$$

where in the fifth inequality we introduce $\max_i$ because the leading stage $\ell$ may be incomplete.

**Case 2.2: Complex eigenvalues with large step size.** Finally we consider the case that $\mathbf{T}_{t,j}$ has complex eigenvalues which is equivalent to Equation (C.40) and also $\mathbf{T}_{t,j}$ is far away from the boundary, namely, the step size of the next stage can still satisfy Equation (C.40). In this stage, it holds that

$$\eta_t \lambda_j > C\eta'_1 \mu. \tag{C.42}$$

Thus if we denote $t_*$ the first iteration of the $\ell_* + 1$ stage where $\ell_*$ is the last stage that $\mathbf{T}'_{\ell,j}$ has complex eigenvalues, namely, $\eta'_{\ell_*}\lambda_j \in (\eta'_1\mu, C\eta'_1\mu]$. We denote $\ell_t$ the stage where iteration $t$ is in. Thus it holds that $\ell_t \leq \ell_* - 1$. Then we consider that

$$
\begin{aligned}
V_{t,j} = & \eta_t^2 \lambda_j^2 \left\|\mathbf{T}_{T-1,j}\mathbf{T}_{T-2,j}...\mathbf{T}_{t+1,j}\right\|^2 \\
\overset{(3.14)}{\leq} & \left\|\mathbf{T}_{T-1,j}...\mathbf{T}_{t_*,j}\right\|^2 \left\|\left(\mathbf{T}'_{\ell_*,j}\right)^K\right\|^2 \left\|\left(\mathbf{T}'_{\ell_*-1,j}\right)^K\right\|^2 ... \left\|\left(\mathbf{T}'_{\ell_t+1,j}\right)^K\right\|^2
\end{aligned}
$$

$$\cdot \max_{0 \leq i \leq K} \left\| \left(\mathbf{T}'_{\ell_t,j}\right)^{K-i} \right\|^2$$

$$\overset{(\text{C.37})}{\leq} \left( \frac{8}{\sqrt{(1+\beta-\eta'_n\lambda_j)^2 - 4\beta}} \right)^2 \left\| \left(\mathbf{T}'_{\ell_*,j}\right)^K \right\|^2 \left\| \left(\mathbf{T}'_{\ell_*-1,j}\right)^K \right\|^2 \cdots \left\| \left(\mathbf{T}'_{\ell_t+1,j}\right)^K \right\|^2$$

$$\cdot \max_{0 \leq i \leq K} \left\| \left(\mathbf{T}'_{\ell_t,j}\right)^{K-i} \right\|^2$$

$$\overset{(\text{C.38})}{\leq} \frac{256}{\eta'_1\mu} \left\| \left(\mathbf{T}'_{\ell_*,j}\right)^K \right\|^2 \left\| \left(\mathbf{T}'_{\ell_*-1,j}\right)^K \right\|^2 \cdots \left\| \left(\mathbf{T}'_{\ell_t+1,j}\right)^K \right\|^2 \cdot \max_{0 \leq i \leq K} \left\| \left(\mathbf{T}'_{\ell_t,j}\right)^{K-i} \right\|^2$$

$$\overset{(\text{C.26})}{\leq} \frac{256}{\eta'_1\mu} \left\| \left(\mathbf{T}'_{\ell_*,j}\right)^K \right\|^2 \cdot \max_{0 \leq i \leq K} \left\| \left(\mathbf{T}'_{\ell_t,j}\right)^{K-i} \right\|^2$$

$$\overset{(\text{C.22})}{\leq} \frac{256}{\eta'_1\mu} \left( 8K \left(\sqrt{\beta}\right)^K \right)^2 (8K)^2 \leq \frac{2^{20}K^4}{\eta'_1\mu} \left( 1 - \sqrt{\eta'_1\mu} \right)^{2K}$$

$$\overset{(\text{C.34}),\ (\text{C.42})}{\leq} \frac{2^{20}T^4}{\eta'_1\mu} \left( 1 - \sqrt{\eta'_1\mu} \right)^{\sqrt{\kappa}\ln(2^{14}T^6\kappa)} \overset{(\text{D.4})}{\leq} \frac{256}{T^2},$$

where in the first inequality we introduce $\max_i$ because the leading stage $\ell_t$ may be incomplete.

Therefore, combining the four cases, we have the result

$$V = \sigma^2 \sum_{j=1}^d \sum_{t=0}^{T-1} V_{t,j} \leq \sigma^2 \sum_{j=1}^d \sum_{t=0}^{T-1} \frac{256C^2h^2}{T^2} \overset{(\text{C.28})}{\leq} \frac{4096C^2d\sigma^2}{T} \ln^2 \left(2^6 T^4\right) \cdot \log_C^2 \left(T\sqrt{\kappa}\right),$$

which concludes the proof in the case that batch size $M = 1$.

For general batch size $M \geq 1$, the gradient noise

$$\mathbf{n}'_t = \nabla_{\mathbf{w}} f(\mathbf{w}_t) - \frac{1}{|\mathcal{B}_t|} \sum_{\xi \in \mathcal{B}_t} \nabla_{\mathbf{w}} f(\mathbf{w}_t, \xi)$$

$$= \frac{1}{M} \sum_{\xi \in \mathcal{B}_t} \left( \nabla_{\mathbf{w}} f(\mathbf{w}_t) - \nabla_{\mathbf{w}} f(\mathbf{w}_t, \xi) \right)$$

$$= \frac{1}{M} \sum_{i=0}^{M-1} \mathbf{n}_{tM+i}$$

satisfies

$$\mathbb{E}\left[ \mathbf{n}'_t \left(\mathbf{n}'_t\right)^\top \right]$$

$$= \mathbb{E}\left[ \left( \frac{1}{M} \sum_{i=0}^{M-1} \mathbf{n}_{tM+i} \right) \left( \frac{1}{M} \sum_{i'=0}^{M-1} \mathbf{n}_{tM+i'}^\top \right) \right]$$

$$= \frac{1}{M^2} \cdot \mathbb{E}\left[ \sum_{i=0}^{M-1} \mathbf{n}_{tM+i} \mathbf{n}_{tM+i}^\top + \sum_{i \neq i'} \mathbf{n}_{tM+i} \mathbf{n}_{tM+i'}^\top \right]$$

$$= \frac{1}{M^2} \cdot \left( \sum_{i=0}^{M-1} \mathbb{E}\left[ \mathbf{n}_{tM+i} \mathbf{n}_{tM+i}^\top \right] + \sum_{i \neq i'} \mathbb{E}\left[ \mathbf{n}_{tM+i} \mathbf{n}_{tM+i'}^\top \right] \right)$$

$$= \frac{1}{M^2} \cdot \left( \sum_{i=0}^{M-1} \mathbb{E}\left[ \mathbf{n}_{tM+i} \mathbf{n}_{tM+i}^\top \right] + \sum_{i \neq i'} \mathbb{E}\left[ \mathbf{n}_{tM+i} \right] \mathbb{E}\left[ \mathbf{n}_{tM+i'}^\top \right] \right) \qquad \triangleright \text{Assumption 1}$$

$$= \frac{1}{M^2} \cdot \sum_{i=0}^{M-1} \mathbb{E}\left[ \mathbf{n}_{tM+i} \mathbf{n}_{tM+i}^\top \right] \qquad \triangleright \text{Assumption 2}$$

$$\preceq \frac{1}{M^2} \cdot M \cdot \sigma^2 \mathbf{H} \qquad \triangleright \text{Assumption 3}$$

$$= \frac{\sigma^2 \mathbf{H}}{M}.$$

So for general $M \geq 1$, it is equivalent to replacing the noise term $\sigma^2$ with $\sigma^2/M$. Thus it holds that

$$V \leq \frac{4096 C^2 d \sigma^2}{MT} \ln^2 \left( 2^6 T^4 \right) \cdot \log_C^2 \left( T \sqrt{\kappa} \right).$$

Combining the bias and variance term, we can verify that Theorem 2 holds.

$\square$

## D PRELIMINARY: USEFUL LEMMAS

We provide some basic mathematical tools relevant to our proof. It serves as a manual section and can be skipped if one is already familiar with those tools.

### D.1 RANDOM VARIABLES

**Lemma 12.** *If each pair of entries of matrix $\mathbf{X}$ and matrix $\mathbf{Y}$ are independent from each other, then*

$$\mathbb{E}\left[\mathbf{XY}\right] = \mathbb{E}\left[\mathbf{X}\right] \mathbb{E}\left[\mathbf{Y}\right] \tag{D.1}$$

*Proof.* For $\forall i, j$,

$$\mathbb{E}\left[\mathbf{XY}\right]_{i,j} = \mathbb{E}\left[\left(\mathbf{XY}\right)_{i,j}\right] = \mathbb{E}\left[\sum_k \mathbf{X}_{i,k} \mathbf{Y}_{k,j}\right] = \sum_k \mathbb{E}\left[\mathbf{X}_{i,k} \mathbf{Y}_{k,j}\right] = \sum_k \mathbb{E}\left[\mathbf{X}_{i,k}\right] \mathbb{E}\left[\mathbf{Y}_{k,j}\right]$$

$$= \sum_k \mathbb{E}\left[\mathbf{X}\right]_{i,k} \mathbb{E}\left[\mathbf{Y}\right]_{k,j} = \left(\mathbb{E}\left[\mathbf{X}\right] \mathbb{E}\left[\mathbf{Y}\right]\right)_{i,j}$$

Thus $\mathbb{E}[\mathbf{XY}] = \mathbb{E}[\mathbf{X}]\mathbb{E}[\mathbf{Y}]$. $\square$

### D.2 LOEWNER ORDER

In Loewner order, $\mathbf{X} \preceq \mathbf{Y}$ if and only if $\forall \mathbf{z}, \mathbf{z}^\top (\mathbf{Y} - \mathbf{X}) \mathbf{z} \geq 0$.

**Lemma 13.** *Given two $2 \times 2$ block matrices $\mathbf{X}$ and $\mathbf{Y}$ where*

$$\mathbf{X}_{11} \preceq \mathbf{Y}_{11}, \quad \mathbf{X}_{12} = \mathbf{Y}_{12}, \quad \mathbf{X}_{21} = \mathbf{Y}_{21}, \quad \mathbf{X}_{22} = \mathbf{Y}_{22},$$

*then $\mathbf{X} \preceq \mathbf{Y}$.*

*Proof.*

For $\forall \mathbf{z} = \begin{bmatrix} \mathbf{z}_1 \\ \mathbf{z}_2 \end{bmatrix}$, $\quad \mathbf{z}^\top (\mathbf{Y} - \mathbf{X}) \mathbf{z} = \begin{bmatrix} \mathbf{z}_1^\top & \mathbf{z}_2^\top \end{bmatrix} \begin{bmatrix} \mathbf{Y}_{11} - \mathbf{X}_{11} & \mathbf{O} \\ \mathbf{O} & \mathbf{O} \end{bmatrix} \begin{bmatrix} \mathbf{z}_1 \\ \mathbf{z}_2 \end{bmatrix} = \mathbf{z}_1^\top (\mathbf{Y}_{11} - \mathbf{X}_{11}) \mathbf{z}_1 \geq 0,$

where the inequality comes from $\mathbf{X}_{11} \preceq \mathbf{Y}_{11}$. Therefore $\mathbf{X} \preceq \mathbf{Y}$ according to the definition of Loewner order. $\square$

**Lemma 14.** *If $\mathbf{X} \preceq \mathbf{Y} \in \mathbb{R}^{n \times n}$, then for $\forall \mathbf{C} \in \mathbb{R}^{n \times m}$,*

$$\mathbf{C}^\top \mathbf{XC} \preceq \mathbf{C}^\top \mathbf{YC}$$

*Proof.* For $\forall \mathbf{z} \in \mathbb{R}^m$,

$$\mathbf{z}^\top \left( \mathbf{C}^\top \mathbf{YC} - \mathbf{C}^\top \mathbf{XC} \right) \mathbf{z} = (\mathbf{Cz})^\top (\mathbf{Y} - \mathbf{X})(\mathbf{Cz}) \geq 0 \quad \Rightarrow \quad \mathbf{C}^\top \mathbf{XC} \preceq \mathbf{C}^\top \mathbf{YC}$$

$\square$

**Lemma 15.** *If $\mathbf{X} \preceq \mathbf{Y}$, then $\mathrm{tr}(\mathbf{X}) \leq \mathrm{tr}(\mathbf{Y})$.*

*Proof.* Denote

$$\mathbf{e}_i = [0 \ldots 0 \underbrace{1}_{i-th} 0 \ldots 0]^\top \in \mathbb{R}^d,$$

$$\mathbf{Y}_{ii} - \mathbf{X}_{ii} = \mathbf{e}_i^\top (\mathbf{Y} - \mathbf{X}) \mathbf{e}_i \geq 0 \quad \text{for } \forall i \quad \Rightarrow \quad \mathrm{tr}(\mathbf{Y}) - \mathrm{tr}(\mathbf{X}) = \sum_{i=1}^d \left( \mathbf{Y}_{ii} - \mathbf{X}_{ii} \right) \geq 0$$

$\square$

### D.3 QUADRATIC EQUATIONS

**Lemma 16.** *(Roots of quadratic equations) If $x_1, x_2 \in \mathbb{C}$ are roots of equation $x^2 + Bx + C = 0$, where $B, C \in \mathbb{R}$, then*

$$
\begin{aligned}
&(1) \quad x_1 + x_2 = -B, \\
&(2) \quad x_1 x_2 = C \\
&(3) \quad x_1\overline{x_2} + \overline{x_1}x_2 = \begin{cases} 2C, & \text{if } x_1, x_2 \text{ are real} \\ B^2 - 2C, & \text{if } x_1, x_2 \text{ are imaginary} \end{cases} \\
&(4) \quad |x_1|^2 + |x_2|^2 = \begin{cases} B^2 - 2C, & \text{if } x_1, x_2 \text{ are real} \\ 2C, & \text{if } x_1, x_2 \text{ are imaginary} \end{cases}
\end{aligned}
\tag{D.2}
$$

*Proof.* (1) and (2) are a special case of the famous Vieta's formulas, which can be directly obtained from

$$x^2 + Bx + C = (x - x_1)(x - x_2) = x^2 - (x_1 + x_2)x + (x_1 x_2).$$

(3) can be derived from the form of $x_1$ and $x_2$. The roots of $x^2 + Bx + C = 0$ is

$$x = \frac{1}{2}\left(-B \pm \sqrt{B^2 - 4C}\right),$$

If $x_1$ and $x_2$ are both real, then

$$x_1\overline{x_2} + \overline{x_1}x_2 = x_1 x_2 + x_1 x_2 = 2x_1 x_2 = 2C.$$

Otherwise $x_1$ and $x_2$ are both imaginary, then $x_1 = \overline{x_2}$, $x_2 = \overline{x_1}$, which follows

$$x_1\overline{x_2} + \overline{x_1}x_2 = x_1^2 + x_2^2 = (x_1 + x_2)^2 - 2x_1 x_2 = B^2 - 2C.$$

(4) can be obtained from (1) and (3) by

$$
\begin{aligned}
|x_1|^2 + |x_2|^2 &= x_1\overline{x_1} + x_2\overline{x_2} = (x_1 + x_2)(\overline{x_1} + \overline{x_2}) - (x_1\overline{x_2} + \overline{x_1}x_2) = B^2 - (x_1\overline{x_2} + \overline{x_1}x_2) \\
&= \begin{cases} B^2 - 2C, & \text{if } x_1, x_2 \text{ are real} \\ 2C, & \text{if } x_1, x_2 \text{ are imaginary} \end{cases}
\end{aligned}
$$

$\square$

### D.4 BOUNDING SPECIAL FUNCTIONS

**Lemma 17.**

$$\sqrt{1 - x} \le 1 - \frac{x}{2} \quad \text{holds for } \forall x \in [0, 1]. \tag{D.3}$$

*Proof.* Only in this lemma, denote

$$f(x) \triangleq \sqrt{1 - x} - \left(1 - \frac{x}{2}\right),$$

then for $\forall x \in [0, 1)$,

$$
\begin{aligned}
&f(0) = 0, \quad \frac{d}{dx}f(x) = -\frac{1}{2} \cdot \frac{1}{\sqrt{1-x}} + \frac{1}{2} = \frac{1}{2}\left(1 - \frac{1}{\sqrt{1-x}}\right) \le 0 \\
&\Rightarrow \quad f(x) \le f(0) = 0 \\
&\Rightarrow \quad \sqrt{1 - x} \le 1 - \frac{x}{2}
\end{aligned}
$$

For $x = 1$, we also have $\sqrt{1 - x} = 0 \le 1/2 = 1 - x/2$. $\square$

**Lemma 18.** *For $x \in [1, +\infty)$, it holds that*

$$f(x) = \left(1 - \frac{1}{x}\right)^x \le \frac{1}{e}. \tag{D.4}$$

*Proof.* The lemma is equivalent to that

$$x \ln \left(1 - \frac{1}{x}\right) \leq -1$$

Denote $g(t) = \ln(1 - t) + t, \quad t \in (0, 1)$, then $g(t)$ is monotonically decreasing as

$$g'(t) = -\frac{1}{1 - t} + 1 < 0.$$

Therefore, $g(t) \leq g(0) = 0$ when $t \in (0, 1)$. Thus substituting $t = 1/x$, it holds that

$$\ln \left(1 - \frac{1}{x}\right) + \frac{1}{x} \leq 0.$$

After rearrangement, we can obtain the result. $\qquad\square$

