# OpenReview forum: "Accelerated Convergence of Stochastic Heavy Ball Method under Anisotropic Gradient Noise"
_ICLR.cc/2024/Conference — ICLR 2024 poster_

### Official Review · Reviewer_JMNK · 2023-10-21

**Soundness:** 3 good
**Presentation:** 3 good
**Contribution:** 3 good
**Rating:** 8
**Confidence:** 4

**Summary:**

Motivated by empirical successes of stochastic heavy ball (SHB) method and the prevalence of the anisotropic gradient noise in training neural networks, the paper presented last iterate convergence rate for the SHB method on quadratic objectives under the anisotropic gradient noise assumption. The results implies SHB with polynomially decaying learning rate provides $\tilde{O}(\sqrt{\kappa})$ acceleration rate with respect to stochastic gradient descent (SGD) when batch size is large, where $\kappa$ denotes the conditional number.

**Strengths:**

The paper is well-written, and proofs are neat and easy to follow.

The paper formally established a lower bound for SGD tailored for the scope of investigation, which served as a concrete comparison benchmark for SHB method.

The paper presents novel theoretical results of SHB for quadratic loss. The result implies $\tilde{O}(\sqrt{\kappa})$ acceleration in comparison to SGD when batch size is big. The results is also nearly optimal in comparison to heavy ball method in deterministic setting.

The experiment results matches with the theoretical bound: acceleration guarantee with large batch size. The advantage of SHB with large batch size is also well-motivated in practice.

The analysis follows the classical bias variance decomposition paradigm. The novelty is to quantifying bias and variance with some linear operator, and then analyze the property of those linear operators.

Although the scope of theoretical investigation is limited at quadratic loss with anisotropic gradient noise, the author justified the broad implication of such assumptions to real applications.

**Weaknesses:**

Potential minor typos:

pg5: eqn 3.11: missing - sign before $\beta$. (Same typo appears at pg 21 under the proof of Lemma 13 at two places)

pg6: Algorithm 1 line 6, batch size $M$ instead of $m$ being consistent

pg27: the equality applies (C.18) and (C.21): the last row of the matrix $2d$.


Everything is rigorous, but the notation might be a bit overly complicated for what was actually being used in order to establish Theorem 2. For example: number of stages $n_{\ell}$, the stage lengths $\set{ k_1, \cdots,  k_{n_{\ell}} }$. Theorem 2 actually set all stages the same length, and each stage has the same learning rate.

**Questions:**

No

---

> ### Author Response · Authors · 2023-11-17
> **Response to reviewer JMNK, thanks!**
>
> We would like to extend our sincere appreciation to the reviewer for all the constructive and positive feedbacks on our contribution, which definitely helps us improve the paper concretely and motivates us to do our best. Thanks!
>
> **Overall:**
>
> We have made corresponding changes in our revised paper to address the mentioned potential issues.
>
> **Weaknesses**
>
> Thanks very much for pointing out those typos and organization issues! We have uploaded a revised version of our paper to reflect the changes.

---

### Official Review · Reviewer_Vpgj · 2023-10-30

**Soundness:** 3 good
**Presentation:** 3 good
**Contribution:** 3 good
**Rating:** 6
**Confidence:** 3

**Summary:**

The paper investigates effect of Polyak-type momentum in SGD with large batch size. The authors show that momentum, combined with a stage-wise geometric decay stepsize schedule, improves the linear convergence part of SGD. Although the analysis is done on convex quadratic functions, it takes a step towards better understanding of SGD with momentum.

**Strengths:**

The paper is well-written and easy to follow. The motivation and main results are clearly presented, with enough intuition given after each result. The technical results seem solid and are supported by empirical evidence.

**Weaknesses:**

There is still a gap between real-life SGD applications and the analysis. Most SGD applications are for nonconvex (even nonsmooth) problems, where Hessian may even not exist and the notion of condition number is uncommon.

**Questions:**

1. Assumption 3 seems to suggest "noise is small in the coordinates that result in ill-conditioning". What do you think is the difficulty in extending it to commonly used assumptions (e.g., bounded variance)?
2. In your first experiment, It seems that SGD performs worse when $M$ gets large. To me it feels that it is because your range of grid search for $\eta_t$ is invariant for all batchsizes. I'm curious whether SGD performs better when you multiply the stepsize by $\sqrt{M}$.
3. Nonconvex models are actually used in the second experiment. Although the authors claim that the strongly convex quadratic optimization model approximates the landscape near local optimum, Figure 1 actually suggests that benefits of momentum are significant at the beginning of the training procedure. Could you elaborate more on this?
4. Some recent researches show that even without decay of learning rate [1], momentum SGD also outperforms SGD using large batch size. What do you think might contribute to this phenomenon?
5. I would suggest the authors remove (or delay to the end of the appendix) proofs for auxiliary results, most of which are well-known results. Some notations should be properly defined (such as $\mathbf{X} \succeq \mathbf{Y}$ and Hessian norm $\\|\cdot\\|_\mathbf{H}$)

**Minor typos and stylistic issues**

1. Algorithm 1. Line 6

   $m$ => $M$

2. Experiment

   initialize $\mathbf{w}^0$ from $(-1,1) \Rightarrow (-1,1)^d$.

3. Proof of Lemma 3

   $\succ$ should be $\succeq$

4. Page 29

   genarality => generality

**References**

[1] Wang, R., Malladi, S., Wang, T., Lyu, K., & Li, Z. (2023). The marginal value of momentum for small learning rate sgd. *arXiv preprint arXiv:2307.15196*.

---

> ### Author Response · Authors · 2023-11-17
> **Response to reviewer Vpgj (part 1/2)**
>
> We would like to offer our sincere thanks for the reviewer's constructive comments and recognition on our contributions.
>
> **Overall**:
>
> Regarding the raised questions, We have provided additional experimental results and relevant discussions to address the concerns. The details are as follows.
>
>
> **Q1: What do you think is the difficulty in extending Assumption 3 to commonly used assumptions (e.g., bounded variance)?**
>
> That's an interesting question. I think one major difficulty arises from bias-variance decomposition.
>
> First, for SGD, a more general assumption $E[n_t n_t^\top] \preceq \sigma^2 D$ can be well handled, where the rest standard proofs for quadratics lead to $\sigma^2 D_{jj} / (\lambda_j T) \le \sigma^2 D_{jj} / (\mu T)$ for each dimension, summing them up across dimensions and we obtain the common upper bound $O(\sigma^2 / (\mu T))$ given $\sum_j D_{jj} = 1$.
>
> For SHB in our proof, one thing is that bias-variance decomposition requires the noise and the hessian to be simultaneously diagonalizable, as shown in (C.19). If this part is resolved, we shall obtain similar results as SGD and recover the $~O(\sigma^2 / (\mu T))$ bound, given the rest part of the proof in our paper providing similar guarantees.
>
> Also, it is worth noticing that generally our setting is technically harder than normal settings of bounded variance. For bounded variance, standard techniques such as [1] can be considered to apply, which we believe to lead to a results similar to existing literatures. We choose Assumption 3 because the corresponding result is still unknown and the setting is technically more difficult.
>
> **Q2: In your first experiment, ... I'm curious whether SGD performs better when you multiply the stepsize by $\sqrt{M}$.**
>
> Thanks for the question! Actually we tried larger learning rates before, such as initial learning rate $\eta_0 = 10.0$, but the loss soon exploded. This is no surprise given it matches the theoretical insight that learning rate $\eta_t$ must be no greater than $2/L$, no matter how large the batch size is.
>
> We also conducted additional experiments of multiplying the stepsize by $\sqrt{M}$. The loss also exploded for $M \ge 32$. According to our experience, increasing the stepsize generally helps SGD to converge faster, but there is an upper limit for this increased stepsize.
>
>
> **Q3: Nonconvex models are actually used in the second experiment. Although the authors claim that the strongly convex quadratic optimization model approximates the landscape near local optimum, Figure 1 actually suggests that benefits of momentum are significant at the beginning of the training procedure. Could you elaborate more on this?**
>
> That's a very good question. We provide further experimental results on cifar10 to verify SGD and SHB's behavior near optimum, where we first run SHB to for 50 epochs (batch size 2048, constant learning rate $\eta = 0.1$) to approach the optimum. We then conduct the exactly same hyperparameters search process as our original experimental settings, with batch size 2048 and number of epochs 100.
>
> As shown in following table, both SGD and SHB has a much lower loss as it is near the optimum, but SHB's loss is an order of magnitude lower than SGD, along with a higher test accuracy.
>
> || ResNet-18 || MobilenetV2 || DenseNet-121 ||
> | -- | -- | -- | -- | -- | -- | -- |
> | Method |  Training loss | Test acc (%) | Training loss | Test acc (%) | Training loss | Test acc (%) |
> | SGD | $4.13 \times 10^{-2}$  | $90.14$ | $3.40 \times 10^{-2}$ | $90.29$ | $3.93 \times 10^{-3}$ | $92.91$ |
> | SHB  | $2.08 \times 10^{-3}$ | $91.10$ | $6.32 \times 10^{-3}$ | $91.07$ | $2.51 \times 10^{-4}$  | $93.58$ |
>
> In addition, our intention of the original set of experiments is to demonstrate the effectiveness of momentum in common non-convex settings, whose assumption may not be perfectly aligned with our theory as the reviewer implied.
>
>
> [1]: Aybat, Necdet Serhat, et al. "A universally optimal multistage accelerated stochastic gradient method." Advances in neural information processing systems 32 (2019).

---

> > ### Author Response · Authors · 2023-11-17
> > **Response to reviewer Vpgj (part 2/2)**
> >
> > **Q4: Some recent researches show that even without decay of learning rate [1], momentum SGD also outperforms SGD using large batch size. What do you think might contribute to this phenomenon?**
> >
> > Thanks for the suggestion! We have included the mentioned paper in our revised paper to discuss the similarity and difference.
> >
> > Briefly speaking, the conclusion of this paper does not conflict with our paper. When the noise level is high and batch size is not sufficiently large, variance term becomes dominant, thus the acceleration benefit brought by momentum in the bias term becomes subtle. Also, when the number of iterations becomes large, the benefit also diminishes, as the bias term will no longer be the bottlenck of SGD.
> >
> > On the other hand, momentum can provide acceleration with constant learning rates. This normally happens when the number of iterations is small or the batch size is sufficiently large, where bias term becomes dominant. Our work further analyze the effect of step-decay learning rate schedule, which generally serves as a variance-reduction technique to reduce the variance term, while still retaining the benefits of momentum in the bias term.
> >
> >
> > **Q5: I would suggest the authors remove (or delay to the end of the appendix) proofs for auxiliary results, most of which are well-known results. Some notations should be properly defined (such as $\mathbf{X} \succeq \mathbf{Y}$ and Hessian norm $\|\|\cdot\|\|_{\mathbf{H}}$)**
> >
> > Thanks for the suggestion! We have made corresponding changes in our revisions.
> >
> >
> > **Minor typos and stylistic issues**
> >
> > Thank you very much for pointing out these issues! We have fixed them accordingly in the revision.

---

> > > ### Comment · Reviewer_Vpgj · 2023-11-19
> > > **Thanks for your response**
> > >
> > > Thank you for the response. And based on the response, I keep my current evaluation of the paper.

---

> > > > ### Author Response · Authors · 2023-11-19
> > > > **Thanks for the positive feedback!**
> > > >
> > > > Thanks very much for recognizing the contribution of our paper! We really appreciate it.

---

### Official Review · Reviewer_dqiw · 2023-10-31

**Soundness:** 3 good
**Presentation:** 3 good
**Contribution:** 2 fair
**Rating:** 6
**Confidence:** 4

**Summary:**

This study presents an analysis of the convergence properties of the stochastic HB method when applied to quadratic objective functions. Specifically, when we make an assumption about the presence of anisotropic noise, it demonstrates that the stochastic HB method exhibits a significantly faster rate of convergence. Earlier research has already established that this anisotropic noise model can manifest in neural networks when their parameters are in proximity to the optimal values.

**Strengths:**

This paper for the first time shows that in the anistorpic setting for the quadratic objective function, the stochastic HB method can achieve accelerated rate. This is a novel contribution.

**Weaknesses:**

The paper should delve further into the circumstances under which the anisotropic setting is applicable to machine learning models. The content of Theorem 1 may seem redundant as its proof is not contingent on stochasticity; it essentially establishes a lower bound for gradient descent (GD), a well-known result. Therefore, its inclusion in the paper is somewhat unclear. It would be more appropriate to allocate space to discuss the technical innovations of the main theorem's proof within the main body of the paper. Additionally, there are several typographical errors in the proofs, including indexing, which should be rectified.

**Questions:**

In the paper you mentioned that this rate achieved for the objective in the large enough batch regime. However the acceleration is oblivion to the mini-batch size. So I was wondering why you emphasise on this in the paper.

---

> ### Author Response · Authors · 2023-11-17
> **Response to reviewer dqiw**
>
> Thank you very much for all the constructive comments and recognition on our contributions. We really appreciate it.
>
> **Overall**
>
> Regarding the mentioned concerns, we have provided relevant discussions and a revised version of our paper to address the issues.
>
> **Weakness**
>
> Thanks for all the constructive suggestions! We have made corresponding revisions based on the mentioned valuable advices, including
>
> 1) Shorten the paragraph for Theorem 1.
> 2) Add two short paragraphs after Corollary 3 to discuss about the intuition of this paper's technical innovation in proofs.
> 3) Rectify the typos pointed out by the reviewer.
>
> Any further comments would be greatly appreciated.
>
>
> **Q: In the paper you mentioned that this rate achieved for the objective in the large enough batch regime. However the acceleration is oblivion to the mini-batch size. So I was wondering why you emphasise on this in the paper.**
>
> That's a very good question. It is because without a sufficiently large batch size, the acceleration effect in the bias term will become much less obvious, and may even be completed covered by the large variance term. This phenomenon is similar to the one in [1] mentioned by reviewer Vpgj, where under small or medium batch size settings, momentum's empircal benefit is not evident.
>
> [1] Wang, R., Malladi, S., Wang, T., Lyu, K., & Li, Z. (2023). The marginal value of momentum for small learning rate sgd. arXiv preprint arXiv:2307.15196

---

### Official Review · Reviewer_Uq7v · 2023-11-01

**Soundness:** 3 good
**Presentation:** 4 excellent
**Contribution:** 2 fair
**Rating:** 5
**Confidence:** 4

**Summary:**

This paper analyzes the performance of SGD and SHB (Stochastic Heavy Ball) in stochastic optimization with a quadratic objective. A lower bound of the convergence rate of SGD is given, which argued to be strictly worse than the upper bound of convergence rate of SHB also proved in the paper. More precisely, SGD is shown to take at least $O(\kappa)$ iterations to converge, while SHB requires only $\tilde{O}(\sqrt{\kappa})$ iterations with proper learning rate schedule.

**Strengths:**

The contrast between SGD and SHB does seem plausible. I checked the proof briefly and found no obvious mistakes.

**Weaknesses:**

1. The upper bound of SHB seems very similar (and worse by a logarithmic factor) than the quadratic setting in [Can et al. (2019)](https://arxiv.org/pdf/1901.07445.pdf). I would suggest adding a detailed comparison. Also the convergence rate is sublinear, which is not very satisfactory.
2. The batch size is required to be of order $\Omega(1/\epsilon)$, which does not seem realistic, as $\epsilon$ is usually exponentially small.

**Questions:**

My foremost concern is the comparison with previous literature, which I discussed in the `Weaknesses` part.

---

> ### Author Response · Authors · 2023-11-17
> **Response to reviewer Uq7v**
>
> We would like to appreciate for all the time and effort the reviewer has spent on checking our proofs, along with recognition on our writing. Thanks!
>
> **Overall**:
>
> According to the reviewer's suggestion, we have included the mentioned paper in related works, together with a corresponding comparison and relevant discussion.
>
> On the other hand, we have noticed that there maybe some misunderstandings or incorrect points in the review, so we would like to provide further explanations to clarify our contributions.
>
>
> **Weakness 1.1: The upper bound of SHB seems very similar (and worse by a logarithmic factor) than the quadratic setting in Can et al. (2019). I would suggest adding a detailed comparison.**
>
> Thanks for the suggestion! We've added corresponding discussions and comparisons with the aforementioned paper in the "Related Work" section.
>
> To make things clearer, we also summarize the comparisons here. Briefly speaking, the proofs in [Can et al. (2019)](https://arxiv.org/pdf/1901.07445.pdf) has **different settings** as ours. Among all the results related to quadratic objectives:
>
>   - Theorem 3, 5, 9 focus on the **deterministic version** of Stochastic Heavy Ball (SHB) and Nesterov's Accelerated Gradient (NAG), which has no stochastic gradient noise and thus can achieve accelerated linear convergence rate with no surprise.
>   - Theorem 4, 7, 11 measure the **distributional convergence**, i.e. convergence speed of parameter distributions towards a stable distribution. Notice that this does NOT imply parameter convergence to optimum or expected excess risk $E[f(w)] - f(w_*)$ convergence to 0.
>   - Theorem 8, 12 provides an upper bound in terms of the expected loss/expected excess risk, which has an extra non-convergent term thus is **much worse** than our bounds.
>
>
> **Weakness 1.2: Also the convergence rate is sublinear, which is not very satisfactory.**
>
> We would love to obtain linear convergence if possible. Regrettably, it is theoretically non-achievable under stochastic gradient settings, as pointed by lower bounds mentioned in [1] and [2]. This lower bound is called statistical minimax rate, which not only applies to SHB and NAG, but also **all** possible optimization methods based on stochastic gradient oracles. Briefly speaking, all of them have a $\Omega(1/T)$ lower bound for convergence rate in terms of expected excess risk $E[f(w)] - f(w_*)$.
>
> It is worth noticing that the provided convergence rate in our paper is already near-optimal when compared with the aforementioned lower bounds (up to log factors).
>
> The mentioned paper [Can et al. (2019)](https://arxiv.org/pdf/1901.07445.pdf) may give readers a false impression that linear convergence is achievable for SHB or NAG, where they are actually only achievable in distributional convergence or deterministic settings, but not for stochastic optimization, as discussed in our responses for Weakness 1.1.
>
>
> **Weakness 2: The batch size is required to be of order $\Omega(1/\epsilon)$, which does not seem realistic, as $\epsilon$ is usually exponentially small.**
>
> We would like to kindly remind the reviewer that there is no such requirement in our paper. Notice that in Corollary 3, we only need the number of training samples $MT$ to be $\Omega(1/\epsilon)$, where $M$ is the batch size and $T$ is the number of iterations. This is achievable since we adopt decaying learning rates instead of constant learning rates, where the generated variance decreases when learning rates gradually diminish during training.
>
>
>
> [1]: Ge, Rong, et al. "The step decay schedule: A near optimal, geometrically decaying learning rate procedure for least squares." Advances in neural information processing systems 32 (2019).
>
> [2]: Wu, Jingfeng, et al. "Last iterate risk bounds of sgd with decaying stepsize for overparameterized linear regression." International Conference on Machine Learning. PMLR, 2022.

---

> ### Comment · Reviewer_Uq7v · 2023-11-18
>
> I would like to express my appreciation to the authors for their very patient response to my review, and my apologize for a few comments that are unclear or incorrect. The authors did make an impressive effort to address my concerns, but some questions persist.
>
> 1. I appreciate the efforts of the authors to provide a very careful comparison with Can et al. (2019). This helps to clarify the situations, but aside from the obvious differences the authors mentioned, there are some deeper connections that need to be addressed. For example, the point that there is an extra non-convergent term in Can et al. (2019), which uses a constant learning rate, seems easily remedied by using a decaying learning rate (hinted in Remark 23 therein).
>
> 2. By sublinear convergence I meant the first term $\mathbb{E}(f(w_0)-f(w_*))\cdot\exp(2\log(T)-\Omega(T/\log T))$ rather than the variance term (which, of course, cannot be linear). Usually something like $\exp(-\Omega(T))$ is more welcomed here, so that the algorithm converges linearly before reaching the variance-dominated regime. Anyway, this point is only a minor complaint which does not affect my assessment of this paper. But I greatly appreciate the authors' patience in answering this point which I did not word properly in detail.
>
> 3. I apologize for the careless mistake claiming that $M=\Omega(1/\epsilon)$. I appreciate again the authors' patience for forgiving my carelessness.
>
> My previous evaluation is based on the thought that Theorem 1 is novel, which, as the other reviewers pointed out, does not depend on stochasticity and follows from the corresponding (well-known) result for GD. Considering that the contribution of Theorem 2 is fair but not significant enough compared with Can et al. (2019), I will keep my score unless there is strong evidence to refute this point.

---

> > ### Author Response · Authors · 2023-11-18
> > **Response to reviewer Uq7v (R2), thanks for your prompt reply!**
> >
> > Thanks for the prompt response! We appreciate the reviewer's further comment and would like to provide more explanations about the nontriviality of our results and techniques.
> >
> > **Q1: ... there are some deeper connections that need to be addressed. For example, the point that there is an extra non-convergent term in Can et al. (2019), which uses a constant learning rate, seems easily remedied by using a decaying learning rate (hinted in Remark 23 therein), yielding completely similar results as this paper. I understand that it would be too harsh to ask the authors to discuss possible variants of the proofs in another paper in a short period of time, but it is also hard for me to make a positive recommendation if the contribution of this paper could be summarized, were I a very malicious reviewer, as a straightforward variant of Can et al. (2019) using decaying learning rate without significant modification in the proof.**
> >
> >
> > First, there is absolutely no evidence that [Can et al. (2019)](https://arxiv.org/pdf/1901.07445.pdf) implies $\mathbb{E}[f(w_T)] - f(w_*)]$ matching stochastic lower bound. Specifically, we checked Remark 23 in page 34 of the above paper, which only mentions the **distributional convergence rate** for different learning rate $\alpha$. The reviewer made claims on results not proved in that paper, and claims like "seems easily remedied by using a decaying learning rate (hinted in Remark 23 therein), yielding completely similar results as this paper" is totally unsubstantiated.
> >
> > Second, matching the statistical lower bound is not a simple problem of A+B in terms of combining existing techniques. Under the isotropic noise assumption of [Can et al. (2019)](https://arxiv.org/pdf/1901.07445.pdf), which is (H2) on page 2, no known results exist that match the statistical lower bounds of this paper, no matter what learning rate schedule you use. We would greatly appreciate if the reviewer could provide the full proof of the unsubstantiated claim, as we see no way that this can be proved from Can et al. (2019).
> >
> > Furthermore, regarding significance and technical contributions of our paper, we are the first work that achieves near-optimality for SHB with step decay schedules on quadratics. Almost all previous works analyzed SHB were either based on constant learning rates, or failed to achieve near-optimality, as pointed out in our "Related Work" section. Technically, we invented a **new proof technique** that overcomes the non-commutative obstacle introduced by SHB + decaying learning rate, via utilizing special properties of the $2 \times 2$ HB update matrix. This could be useful for future researchs of SHB + decaying learning rates. The main idea is summarized in paragraphs after Corollary 3 in the revised version of our paper.
> >
> >
> > **Q2: By sublinear convergence I meant the first term $E(f(w_0) - f(w_*)) \cdot \exp(2 \log(T) - \Omega(T / \log T))$ rather than the variance term (which, of course, cannot be linear). Usually something like $\exp(-\Omega(T))$ is more welcomed here (though this is not always possible for decaying learning rate), so that the algorithm converges linearly before reaching the variance-dominated regime...**
> >
> > By elongating the first interval in step decay, this rate is achievable. Specifically, in page 33 of our paper, right hand side of the second inequality, the convergence rate for bias term solely depends of the first interval's length $k_1$. By replacing it with $T/2$, while retaining the rest of interval lengths to be $\tilde{O}(T/\log T)$, this term can be improved to $\exp(-{\Omega}(T/\sqrt{\kappa}))$. We adopt an simpler scheduler so that the proof and notations can be more concise and easy to read.
> >
> > Notice that elongating the first interval is a common practice to improve convergence rates in bias.
> >
> >
> > **Q3: I apologize for the careless mistake claiming that $M = \Omega(1/\epsilon)$. I appreciate again the authors' patience for forgiving my carelessness.**
> >
> > No need to apologize (smileface).
> >
> >
> > **Summary: My previous evaluation is based on ... Considering that Theorem 2 also seems incremental compared with Can et al. (2019), I will keep my score unless there is strong evidence to refute this point.**
> >
> > Theorem 2 is not incremental at all. It is the first work that achieves near-optimality statistical lower bound for SHB with step decay schedules on quadratic objective functions. To achieve this, we have invented new proof techniques that can be beneficial for future researches in this direction. It also clearly illustrated SHB's superiority over SGD under large batch settings.
> >
> > The reviewer's claim of incremental is based on unsubstantiated claim that [Can et al. (2019)](https://arxiv.org/pdf/1901.07445.pdf) covers our result. In fact, that paper has no convergence result for stochastic optimization, not to mention matching statistical lower bound as we do. Moreover, we do not see any easy adapation of their results to derive our result proved in this work.

---

> ### Comment · Reviewer_Uq7v · 2023-11-18
>
> Thanks to the authors for their detailed response. Concerning the interpretation of Can et al. (2019), I would like to remind that distributional convergence is **stronger** than convergence in expectation, as $\mathbb{E}f(w) - f(w_*)\le \frac{L}{2}\mathbb{E}\|w-w^*\|^2\le \frac{L}{2}W_2(\nu(w), \delta_{w_*})^2$.

---

> ### Author Response · Authors · 2023-11-18
> **Response  to reviewer Uq7v R3, false claim and two mistakes**
>
> Thanks for your prompt reply. Regrettably, this claim is simply wrong.
>
> Convergence in distribution generally does not imply convergence in optimization and the two objectives measuring the convergence are not the same. If the reviewer insist on claiming one is stronger, we would encourage the reviewer to write down a step-by-step proof.
>
> There are two mistakes the reviewer is making here.
>
> First, in advanced probability theory, convergence in distribution is indeed stronger than convergence in expectation. However, in the context of optimization, this is totally different. The tool is simply applied in a wrong way here. In Can et al. (2019), it proved that the initial distribution converges to a stable distribution in terms of Wasserstain Distance. Let's omit the details and say that if the tool could be applied here, then we get:
> - The expectation of the distribution, i.e. $\mathbb{E}[w_t]$, converges to the expectation of the stable distribution $\hat{w}$.
>
> Whether $\hat{w}$ is $w_*$ remains unknown. Also, even $\hat{w} = w_*$ and $\mathbb{E}[w_t] \to w_*$, it doesn't imply $\mathbb{E}[f(w_t)] - f(w_*) \to 0$, just consider the simple case of $w_t \in U(-1,1)$ (uniform distribution) and $f(w) = w^2$.
>
> The second mistake is the right hand side of the second inequlity $W_2(\nu(w), \delta_{w_*})^2$. Notice that in Can et al. (2019), it only says that the distribution converges to a stable distribution, whether this stable distribution is Dirac delta distribution at optimum $w_*$ still remains unknown.

---

> ### Comment · Reviewer_Uq7v · 2023-11-18
>
> I wish I could have the privilege of omitting most of the details, since most of them are contained in Can et al. (2019). An argument that is completely similar to my previous comments will show $\mathbb{E} f(w_t) - f(w_*) \lesssim W_2^2(\nu(w_t), \nu_\infty) + \mathbb{E} \|w_\infty - w_* \|^2$, where $w_\infty\sim\nu_\infty$ is the stable distribution. The MSE $\mathbb{E}\|w_\infty-w_*\|^2$ is computed explicitly in (87), which can be adapted to (stagewise) decaying learning rate with only notational change. The authors may also notice that the proof in Appendix C.2 of  Can et al. (2019) is actually very similar to this paper.

---

> > ### Author Response · Authors · 2023-11-18
> > **Response to reviewer Uq7v R4**
> >
> > We really appreciate the prompt reply. The issue here is, $\mathbb{E}|w_\infty-w_*|^2$ is a non-convergent constant in (87) given $\alpha_{HB}$ and $\beta_{HB}$ being constants in (12). Replacing them to decaying learning rates is not straightforward at all, as the whole proof in Appendix C.2 relies on the assumption of constant learning rates, whose result only requires properties of $\|\|T_i^k\|\|$ instead of $\|\|T_{i_1} T_{i_2} \cdots T_{i_n}\|\|$ with different $T_i$'s.
> >
> > Regarding the similarity, it is no surprise that our paper covers some techniques in Can et al. (2019), given step decay is a general version of constant learning rates. But we provide much more non-trivial techniques, especially the **new proof technique** that overcomes the non-commutative obstacle introduced by SHB + decaying learning rate. To the best of our knowledge, no such technique was available before for SHB + decaying learning rates on quadratics.

---

> ### Author Response · Authors · 2023-11-18
> **Response to reviewer Uq7v R5**
>
> Thank you for the prompt reply. We encourage the reviewer to write down the full proof rigorously and step-by-step.
>
> Among various issues in the above proof idea, one is that the reviewer failed to take into account the effect of small learning rates, where the convergence is a logarithmic number of iterations only when the learning rate is large, such as constant learning rate $\alpha_{HB}$ defined in (12) of Can et al. (2019).

---

> ### Author Response · Authors · 2023-11-18
> **Response to Uq7v R6**
>
> We appreciate your prompt reply.
>
> Again, as we pointed out in Response 2 (R2) to the reviewer, there is absolutely no evidence that Can et al. (2019) implies $E[f(w_T)] - f(w_*)$ matching stochastic lower bound. The reviewer's claim is completely unsubstantiated.

---

> > ### Comment · Reviewer_Uq7v · 2023-11-18
> >
> > All my previous comments are clear evidence for this implication, as I have already written a proof showing that $\mathbb{E}f(w_t)-f(w_*)$ converges to $\mathbb{E}\|w_\infty - w_*\|^2$, and Remark 23 argued that it's possible to make $\mathbb{E}\|w_\infty - w_*\|^2=O(1/k)$ with decaying learning rate (to be absolutely rigorous, though it was worded like $\operatorname{Var}(w_\infty)=O(1/k)$, the literature it pointed and the techniques therein actually controlled $\mathbb{E}\|w_\infty - w_*\|^2$). I will stop here as the discussions have become less constructive, but will come back if there is strong evidence otherwise.

---

> ### Author Response · Authors · 2023-11-18
> **Response to Uq7v R7**
>
> The prompt reply is greatly appreciated. As the reviewer repetitively referred to Remark 23 in [Can et al. (2019)](https://arxiv.org/pdf/1901.07445.pdf), we would like to point out some key aspects that the review missed in the remark.
>
> - First, this is for Nesterov's Accelerated Gradient (AG), not Stochastic Heavy Ball (SHB).
>
> - Also, we highly recommend the reviewer to double-check the proof for the remark, is $\alpha = \log^2(k) / (\mu k^2)$ a decaying learning rate?
>
> As we provide techniques for SHB + decaying learning rates, differences in those key aspects matter a lot.

---

> > ### Comment · Reviewer_Uq7v · 2023-11-18
> >
> > When I was typing a full proof here, I realized that there were several lemmas doing the same modification I was thinking of, much of which is already present in the paper. This greatly helps me understand the authors' points. I see that we have disagreements on what kind of technical changes should be considered significant, so let's find a more constructive way to reach a consent. I will avoid saying this paper is a "simple" (which seems too subjective) variant of Can et al. (2019) and will revise all my previous comments accordingly. Instead, I think we can agree on the following points:
> >
> > 1. The bias term was already shown to converge linearly in $O(\sqrt{\kappa})$ iterations in Can et al. (2019), and the main contribution of this paper is to show that the variance term can be made $O(1/T)$ using decaying learning rate.
> >
> > 2. Technically, the main contribution is Lemma 19, and the anisotropic noise assumption (which made it possible to have a $\kappa$-free variance term).
> >
> > I think these points are worth better emphasizing in the paper to clearly position the contributions with the literature. That being said, I retain my opinion that these contributions are fair but fall marginally below the threshold for ICLR's high standard.

---

> ### Author Response · Authors · 2023-11-19
> **Response to Uq7v R8**
>
> We thank for the reviewer's active participation during the discussion session.
>
> ### **Modifications**
>
> To reach a consensus, we have made corresponding modifications in a new revised version of our paper (version 3/V3) asked by the reviewer. Specifically,
>
>   - In the "Related Work" section, we provide the explicit form of accelerated distributional convergence for Can et al. (2019).
>   - In our contribution (Section 1.1), we emphasize more on the variance term $\tilde{O}(d \sigma^2/T)$.
>   - In paragraphs after Corollary 3, which explains the intuition behind one of our key technical contributions, we add a specific reference to the Lemma that provides the contribution.
>
> ### **Consensus**
>
> Regarding the reviewer's proposed two points, we think the reviewer now understands most parts correctly. However, there are still some missing points.
>
> To avoid confusion, we would like to provide more clarification about the background and context:
>   - **Lemma 19 in V1 and Lemma 8 in V2/V3**: For Lemma 19, we assume the reviewer is referring to the version before rebuttal. This lemma corresponds to Lemma 8 in our revised versions during the rebuttal.
>   - **Accelerating bias is easy**: Showing the bias term to be converged linearly in $O(\sqrt{\kappa})$ for constant learning rates is straightforward, as it is essentially the same for deterministic HB, which is covered in almost every textbook about optimization. The main contribution of Can et al. (2019) is that it proves accelerated linear convergence rates for distributional convergence.
>   - **Balancing bias and variance is hard**: It is highly non-trivial to obtain accelerated linear convergence in bias while still achieving near-optimal rates in variance for SHB + decaying learning rates on quadratics, as to the best of our knowledge, no such work existed before.
>   - **Other technical contributions**: The techniques of using schedulers are also adopted in our proofs to balance bias and variance. Combining it with SHB is non-trivial as well.
>
> ### **About Top Conferences**
>
> We would also like to provide the reviewer with a list of past papers published in top conferences for reference, serving as a background for this subfield and top conferences' attitude towards it:
>
>   - **COLT 2018**: Modified SHB + averaging + least square regression (implies anisotropic noise on quadratics), achieving optimal rate and acceleration in certain types of noises [1]
>   - **NeurIPS 2019**: SGD + step decay + least square regression (implies anisotropic noise on quadratics), achieving near-optimal rate [2]
>   - **ICLR 2022**: SGD + step decay/improved scheduler + strongly convex least square regression & anistropic noise on quadratic, achieving near-optimal rate generally and optimal-rate for certain types of instances [3]
>   - **ICML 2022 (Oral)**: SGD + step decay/polynomial decay + overparameterized linear regression (implies anisotropic noise on quadratics), a detailed analysis of lower bounds and upper bounds [4]
>
> As SHB covers SGD, our result is technically much harder. Also to the best of our knowledge, this is the first time that an accelerated convergence rate is obtained while still achieving near-optimal variance for SHB + decaying learning rates on quadratics. It also implies SHB benefits in practice, which is mostly negative results in past literature.
>
> We really appreciate the reviewer's prompt replies during the discussion and hope this can clarify the significance of our contributions.
>
> [1]: Jain, Prateek, et al. "Accelerating stochastic gradient descent for least squares regression." Conference On Learning Theory. PMLR, 2018.
>
> [2]: Ge, Rong, et al. "The step decay schedule: A near optimal, geometrically decaying learning rate procedure for least squares." Advances in neural information processing systems 32 (2019).
>
> [3]: Pan, Rui, Haishan Ye, and Tong Zhang. "Eigencurve: Optimal Learning Rate Schedule for SGD on Quadratic Objectives with Skewed Hessian Spectrums." International Conference on Learning Representationsk(2022).
>
> [4]: Wu, Jingfeng, et al. "Last iterate risk bounds of sgd with decaying stepsize for overparameterized linear regression." International Conference on Machine Learning. PMLR, 2022.

---

### Public Comment · ~Matrix_sun1 · 2023-11-20
**reviewer Uq7v's comments are quite strange**

I have been watching this discussion for 2 days. Seems like reviewer Uq7v is unfamilar with the field of optimization and the mentioned paper Can et al. (2019). Uq7v made a lot of mistakes during the rebuttal. Why does the reviewer have confidence 4?

---

### Meta-Review · Area_Chair_yUFz · 2023-12-13

**Metareview:**

Authors study Stochastic Heavy Ball on a quadratic objective. They provide a last iterate convergence guarantee under the anisotropic gradient noise assumption. Authors are able show acceleration in condition number under decaying step size.

This paper was reviewed by 4 reviewers and received the following Rating/Confidence scores: 8 / 4, 6 / 3, 6 / 4, 5 / 4.

I think the paper is overall interesting and should be included in ICLR. The authors should carefully go over and address all reviewers' suggestions.

**Justification For Why Not Higher Score:**

Quadratic objective function.

**Justification For Why Not Lower Score:**

n/a

---

### Decision · Program_Chairs · 2024-01-16

Accept (poster)